# WEBGEN-AGENT: ENHANCING INTERACTIVE WEBSITE GENERATION WITH MULTI-LEVEL FEEDBACK AND STEP-LEVEL REINFORCEMENT LEARNING

**Zimu Lu**[1*], **Houxing Ren**[1*], **Yunqiao Yang**[1], **Ke Wang**[1], **Zhuofan Zong**[1], **Junting Pan**[1]
**Mingjie Zhan**[2†], **Hongsheng Li**[1,3,4,5†]
[1]Multimedia Laboratory (MMLab), The Chinese University of Hong Kong,
[2]SenseTime, [3]CPII under InnoHK, [4]Shanghai AI Laboratory, [5]Ace Robotics
luzimu@link.cuhk.edu.hk   zhanmingjie@sensetime.com   hsli@ee.cuhk.edu.hk

## ABSTRACT

Agent systems powered by large language models (LLMs) have demonstrated impressive performance on repository-level code-generation tasks. However, for tasks such as website codebase generation, which depend heavily on visual effects and user-interaction feedback, current code agents rely only on simple code execution for feedback and verification. This approach fails to capture the actual quality of the generated code. In this paper, we propose *WebGen-Agent*, a novel website-generation agent that leverages comprehensive and multi-level visual feedback to iteratively generate and refine the website codebase. Detailed and expressive text descriptions and suggestions regarding the screenshots and GUI-agent testing of the websites are generated by a visual language model (VLM), together with scores that quantify their quality. The screenshot and GUI-agent scores are further integrated with a backtracking and select-best mechanism, enhancing the performance of the agent. Utilizing the accurate visual scores inherent in the WebGen-Agent workflow, we further introduce *Step-GRPO with Screenshot and GUI-agent Feedback* to improve the ability of LLMs to act as the reasoning engine of WebGen-Agent. By using the screenshot and GUI-agent scores at each step as the reward in Step-GRPO, we provide a dense and reliable process supervision signal, which effectively improves the model's website-generation ability. On the WebGen-Bench dataset, WebGen-Agent increases the accuracy of Claude-3.5-Sonnet from 26.4% to 51.9% and its appearance score from 3.0 to 3.9, outperforming the previous state-of-the-art agent system. Additionally, our Step-GRPO training approach increases the accuracy of Qwen2.5-Coder-7B-Instruct from 38.9% to 45.4% and raises the appearance score from 3.4 to 3.7.

## 1 INTRODUCTION

Recent studies on code agents have shown great advancements in repository-level code-generation tasks, such as fixing GitHub issues (Yang et al., 2024b) and implementing new features (Miserendino et al., 2025). However, for tasks like website code generation, which depend heavily on visual aesthetics and the fluency of user interactions, current code-agent systems fail to fully capture the actual quality of the generated codebase, because they mostly rely on simple code-execution feedback. This limitation can lead to various rendering and functional problems in the generated web applications, such as misaligned components, disharmonious coloring, unresponsive buttons, and broken links.

To enable the code agent to effectively handle such tasks, we introduce **WebGen-Agent**, a code-generation system that generates websites from natural-language instructions that specify appearance and functional requirements, thus offering a highly automated website-development process. To ensure that the generated websites meet both functional requirements and aesthetic standards, we

---

*Equal contribution.
†Corresponding author.

leverage both execution feedback and visual feedback to refine the project. Specifically, we leverage a visual language model (VLM) to assess the visual appeal and aesthetic quality of the current website, and a graphical user interface (GUI) agent to evaluate the correctness and intended functionality of the website's codebase, thereby gathering accurate information and providing targeted suggestions. By iteratively applying this feedback and editing the codebase, WebGen-Agent builds websites with appealing designs and smooth interactive functionality.

As shown in Fig. 1, WebGen-Agent adopts an iterative, multi-step paradigm in which each step consists of three actions: code generation, code execution, and feedback gathering. The agent begins each step by creating and editing files in the codebase in a manner similar to Bolt.diy (stackblitz labs, 2024). During code execution, dependencies are installed, and the website service is started. If execution emits errors, the errors are returned to the agent, which starts the next step to fix them. If five consecutive erroneous steps occur, the agent backtracks to a previous non-erroneous step.

In the feedback-gathering process, a screenshot of the landing page of the website is captured first. A VLM then provides a description and an appearance score based on the screenshot. If the screenshot has room for improvement, the model supplies suggestions, which are implemented in the subsequent step to explicitly refine the website's visual aesthetics. Otherwise, a GUI-agent session is initiated to explore the website, which evaluates the functional requirements and generates corresponding feedback. If the testing is successful, the task is complete; otherwise, suggestions for fixing the website are generated, and the agent can edit the codebase in the next step. At the end of the task trajectory, the best step is selected on the basis of the screenshot and GUI-agent scores, and the codebase is restored to the state of that step. Based on the pipeline, various models achieve better performance on WebGen-Bench (Lu et al., 2025b), consistently outperforming other code agents. Remarkably, Claude-3.5-Sonnet improves its accuracy from 26.4% to 51.9% and its appearance score from 3.0 to 3.9, outperforming Bolt.diy.

To equip code agents with enhanced reasoning abilities, we further propose **Step-GRPO with Screenshot and GUI-agent Feedback**. Given an instruction, multiple WebGen-Agent trajectories are generated. Each step in an agent trajectory is accompanied by a screenshot score and a GUI-agent testing score, and an accurate and reliable step-level reward can be computed by summing these two scores. This dual supervision of website appearance and functionality effectively optimizes the model to generate high-quality website codebases, providing stepwise, process-level guidance for the agent trajectory. This process transforms information obtained from the VLM's visual ability into the coding LLM's programming skills. Training a Qwen2.5-Coder-7B-Instruct model with this Step-GRPO approach increases the accuracy from 38.9% to 45.4% and raises the appearance score from 3.4 to 3.7 on WebGen-Bench, greatly improving both the functionality and the appearance of the generated websites. We name the trained family of models **WebGenAgent-LM**.

Our contributions include:

- We propose WebGen-Agent, a code-agent system that leverages screenshots and GUI-agent testing to provide feedback signals and iteratively improve the quality of generated websites.
- We introduce Step-GRPO with Screenshot and GUI-agent Feedback, which uses screenshots and GUI-agent scores as step-level supervision in the GRPO training process, significantly improving the performance of smaller open-source models.
- Extensive experiments demonstrate the effectiveness of the proposed system. The system increases the accuracy of Claude-3.5-Sonnet from 26.4% to 51.9% and its appearance score from 3.0 to 3.9, outperforming Bolt.diy. Our training approach also increases the accuracy of Qwen2.5-Coder-7B-Instruct from 38.9% to 45.4% and raises the appearance score from 3.4 to 3.7.

## 2 METHOD

In this section, we first introduce WebGen-Agent, a novel website generation system that leverages screenshots and GUI-agent testing as reliable feedback to iteratively refine both the appearance and functionalities of the generated website with a coding LLM. Building on the reliable visual scores produced by WebGen-Agent, we then propose Step-GRPO with Screenshot and GUI-agent Feedback, a method that uses these scores to provide process supervision during GRPO training. This system significantly enhances language models' ability to generate high-quality websites.

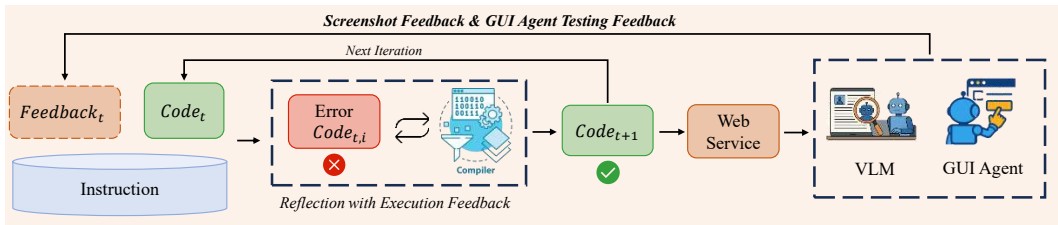

Figure 1: Iterative website generation with screenshot- and GUI-agent-based feedback. A backtracking and best-step-selection mechanism is applied on the basis of the screenshot and GUI-agent testing scores.

## 2.1 WEBGEN-AGENT WORKFLOW

The WebGen-Agent workflow consists of multiple steps, with each step including code generation, code execution, and feedback gathering. As shown in Fig. 1, the agent trajectory starts from a website generation instruction ($\mathcal{I}$), denoted as $\mathcal{T} = [\mathcal{I}]$, and an empty codebase $\mathcal{C}_0$. The instruction $\mathcal{I}$ is created by concatenating a system prompt similar to that of Bolt.diy (stackblitz labs, 2024) with the user-provided website-generation request. A coding LLM acting as the engine of the agent generates code $\Delta\mathcal{C}_1$ to edit the codebase, resulting in $\mathcal{C}_1$. Then, the dependencies of the codebase are installed, and the website service is started. The code execution output is denoted as $\mathcal{O}_1$, which contains both stdout and stderr. If the dependency installation or service initialization fails, the output message $\mathcal{O}_1$ is returned to the agent as feedback, so that the agent can fix the error in the next step. If no error occurs, a screenshot of the website is captured and presented to a VLM (e.g., Qwen2.5-VL-32B; Bai et al. (2025)), which is requested to provide a description of the screenshot and, if needed, suggestions to improve the website's appearance. The prompt for acquiring screenshot feedback is provided in Fig. 4 of Appendix C. A score of the website appearance based on the screenshot is also generated and, together with the description and suggestions, composes the screenshot feedback. The feedback can be denoted as:

$$\mathcal{F}_{\text{shot}} = \langle Description, Score_{\text{shot}}, Suggestions_{\text{shot}} \rangle \quad (1)$$

$\mathcal{F}_{\text{shot}}$ is used to reflect the integrity and aesthetics of the website's appearance. Here, a separate VLM is used besides the coding LLM to make the system more cost-effective, as we observe that a relatively small open-source VLM is sufficient for the task, while the code generation requires an LLM with strong coding abilities. We use Qwen2.5-VL-32B-Instruct as the VLM in our experiments unless stated otherwise. $\mathcal{F}_{\text{shot}}$ focuses on page-level appearance, which is where VLMs currently demonstrate the highest reliability. Extending the task to multi-page inputs would introduce a qualitatively different problem formulation, as cross-page coherence is inherently subjective and highly context-dependent. The code execution and screenshot feedback are appended to the agent trajectory, resulting in $\mathcal{T} = [\mathcal{I}, \Delta\mathcal{C}_1, \mathcal{O}_1, \mathcal{F}_{\text{shot},1}]$. Then, the agent judges whether the website's appearance is satisfactory based on the trajectory. If it is unsatisfactory, the agent continues to generate code $\Delta\mathcal{C}_2$ to improve the website's appearance. Otherwise, the agent initiates a GUI-agent testing session, generating an instruction for the GUI-agent to explore various website functionalities specified in the instruction $\mathcal{I}$, resulting in a GUI-agent testing trajectory. The prompt used to generate the GUI-agent instructions is shown in Fig. 6 of Appendix C. It instructs the model to produce a GUI-agent instruction that comprehensively checks all website-development requirements and includes a one-shot example. As shown in Tab. 9 of Appendix G, a manual inspection indicates that 98.3% of the sampled instructions achieve high coverage of the requirements. Based on the GUI-agent testing result, the LLM acting as the engine of the agent judges whether the testing is successful and provides a score, denoted as $Score_{\text{gui}}$. The prompt for acquiring the GUI-agent testing feedback is provided in Fig. 7 of Appendix C. If the testing result is unsatisfactory, suggestions are also made to improve the functionality. Thus, the GUI-agent testing feedback can be denoted as:

$$\mathcal{F}_{\text{gui}} = \langle Score_{\text{gui}}, Suggestions_{\text{gui}} \rangle \quad (2)$$

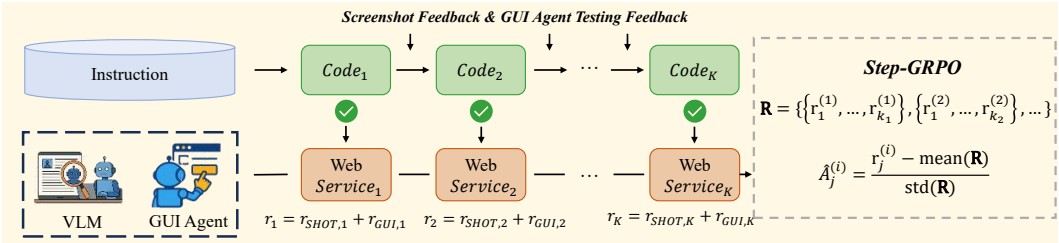

Figure 2: Step-GRPO with Screenshot and GUI-agent Feedback. Multiple WebGen-Agent trajectories are produced, and the reward for each step is computed by summing the screenshot score and the GUI-agent score.

$\mathcal{F}_{\text{gui}}$ is also appended to the trajectory, resulting in $\mathcal{T} = [\mathcal{I}, \Delta\mathcal{C}_1, \mathcal{O}_1, \mathcal{F}_{\text{shot},1}, \mathcal{F}_{\text{gui},1}] = [\mathcal{I}, \Delta\mathcal{C}_1, \mathcal{O}_1, \mathcal{F}_1]$. Here, $\mathcal{F}_1$ denotes $[\mathcal{F}_{\text{shot},1}, \mathcal{F}_{\text{gui},1}]$. In this way, WebGen-Agent continues to improve the appearance and functionality of the website, resulting in a trajectory $\mathcal{T}$, denoted as $\mathcal{T} = [\mathcal{I}, \Delta\mathcal{C}_1, \mathcal{O}_1, \mathcal{F}_1, \Delta\mathcal{C}_2, \mathcal{O}_2, \mathcal{F}_2, \ldots, \Delta\mathcal{C}_K, \mathcal{O}_K, \mathcal{F}_K]$.

The process ends when the website passes the GUI-agent testing, or the maximum iteration number is reached. During the iterations, at step $i \in \{1, 2, \ldots\}$, the codebase state $\mathcal{C}_i$, the edit $\Delta\mathcal{C}_i$, together with the $Score_{\text{shot},i}$ and $Score_{\text{gui},i}$, are stored in a memory list. If five consecutive steps contain code execution errors, a backtracking mechanism is triggered, and the agent trajectory and the codebase are returned to the state at the best previous step. The best previous step is selected by first choosing the steps with the highest $Score_{\text{gui}}$, and then among these steps, the ones with the highest $Score_{\text{shot}}$ are chosen. If there is still more than one chosen step, then the latest one among them is selected. Considering that later code edits might not always improve the previous codebase, at the end of the agent workflow, the best step among all the steps is selected in the same way as mentioned above. A more detailed algorithmic presentation can be found in Appendix B, and example trajectories are presented in Appendix D.

## 2.2 STEP-GRPO WITH SCREENSHOT AND GUI-AGENT FEEDBACK

While using strong proprietary models as the engine LLM in WebGen-Agent can produce high performance, the agent workflow would be more cost-efficient if smaller open-source models of 7B-8B parameters can be used instead. However, current small open-source language models still lag behind proprietary models in website code generation. Therefore, we introduce Step-GRPO with Screenshot and GUI-agent Feedback, leveraging the $Score_{\text{shot}}$ and $Score_{\text{gui}}$ inherently produced in the WebGen-Agent workflow to train them with step-level process supervision in GRPO training.

Before the GRPO-based training, we first perform a light supervised fine-tuning (SFT) using approximately 700 WebGen-Agent trajectories generated by DeepSeek-V3, training for one epoch to serve as a warm start. Then, Step-GRPO is performed on the fine-tuned model. The Step-GRPO training objective can be written as:

$$\mathcal{J}_{GRPO}(\theta) = \mathbb{E}_{[q \sim P(Q), \{o_i\}_{i=1}^{G} \sim \pi_{\theta_{old}}(O|q)]}$$
$$\frac{1}{G}\sum_{i=1}^{G}\frac{1}{|o_i|}\sum_{t=1}^{|o_i|}\left\{\min\left[\frac{\pi_\theta(o_{i,t}|q, o_{i,<t})}{\pi_{\theta_{old}}(o_{i,t}|q, o_{i,<t})}\hat{A}_{i,t}, \text{clip}\left(\frac{\pi_\theta(o_{i,t}|q, o_{i,<t})}{\pi_{\theta_{old}}(o_{i,t}|q, o_{i,<t})}, 1-\epsilon, 1+\epsilon\right)\hat{A}_{i,t}\right]\right\}, \quad (3)$$

Here, $q$ denotes the website generation instruction, and $\{o_i\}_{i=1}^{G}$ denotes the group of trajectories generated from the instruction $q$. We remove the KL loss to encourage the model to more freely adapt its behavior to the reward signals (Qian et al., 2025). $o_i$ can be denoted as $[\Delta\mathcal{C}_1, \mathcal{O}_1, \mathcal{F}_1, \ldots, \Delta\mathcal{C}_{K_i}, \mathcal{O}_{K_i}, \mathcal{F}_{K_i}]$. Different from the naive GRPO, which sets the advantages on all tokens in a trajectory to the same value, the Step-GRPO sets advantages on tokens in different steps to different values. In our work, the GRPO loss is only applied to the model outputs $\Delta\mathcal{C}_1, \Delta\mathcal{C}_2, \ldots, \Delta\mathcal{C}_K$. We denote the reward of all tokens in the $j$-th step of $o^{(i)}$ as $r_j^{(i)}$, which is computed by summing the $Score_{\text{shot}}$ and $Score_{\text{gui}}$ of that step, generated in the WebGen-Agent workflow:

$$r_j^{(i)} = Score_{\text{shot},j}^{(i)} + Score_{\text{gui},j}^{(i)} \quad (4)$$

The rewards for all steps in the trajectories sampled from $q$ can be written as $\mathbf{R} = \{\{r_1^{(1)}, \cdots, r_{K_1}^{(1)}\}, \ldots, \{r_1^{(G)}, \cdots, r_{K_G}^{(G)}\}\}$. The advantage for step $j$ of the $i$-th trajectory is computed by standardizing its immediate reward: $\hat{A}_j^{(i)} = \dfrac{r_j^{(i)} - \text{mean}(\mathbf{R})}{\text{std}(\mathbf{R})}$. $\hat{A}_j^{(i)}$ denotes the advantage of $o^{(i)}$ at the $j$-th step. We do not accumulate normalized rewards from future steps as in Shao et al. (2024), because in the website-generation task $Score_{\text{shot}}$ and $Score_{\text{gui}}$ directly reflect the quality of the website at the current step, which is more appropriate for representing the desirability of the current code. The Step-GRPO training process is illustrated in Fig. 2. This Step-GPPO method, with screenshot and GUI-agent feedback, incorporates accurate step-level supervision and effectively helps the model learn to generate websites with an appealing appearance and smooth functionality.

## 3 EXPERIMENTS

In this section, we first present the performance of WebGen-Agent on WebGen-Bench using a variety of proprietary and open-source LLMs, as well as models trained using Step-GRPO with Screenshot and GUI-agent Feedback. Then, we conduct comprehensive ablation studies on the design choices in the WebGen-Agent workflow and the Step-GRPO training process.

### 3.1 MAIN RESULTS

**Benchmark Dataset and Baselines.** We evaluate WebGen-Agent using WebGen-Bench (Lu et al., 2025b), a benchmark containing 101 website-generation instructions in natural language and 647 GUI-agent test cases, covering a wide range of web applications. Following Lu et al. (2025b), we use Qwen2.5-VL-32B-Instruct (Bai et al., 2025) in functional testing and GPT-4o (Hurst et al., 2024) in appearance evaluation. We compare WebGen-Agent with three other popular code agents: OpenHands (Wang et al., 2024), Aider (Aider-AI, 2024), and Bolt.diy (stackblitz labs, 2024). We present the results of OpenHands and Aider in combination with DeepSeek-V3 (Liu et al., 2024), Claude-3.5-Sonnet (Anthropic, 2024), and DeepSeek-R1 (Guo et al., 2025a), as well as the results of Bolt.diy with DeepSeek-V3 (Liu et al., 2024), Claude-3.5-Sonnet (Anthropic, 2024), DeepSeek-R1 (Guo et al., 2025a), GPT-4o (Hurst et al., 2024), o3-mini (OpenAI, 2025b), Qwen2.5-Coder-32B (Hui et al., 2024), Qwen2.5-72B-Instruct (Yang et al., 2024a), WebGen-LM-7B, WebGen-LM-14B, and WebGen-LM-32B (Lu et al., 2025b). The values are taken from Lu et al. (2025b). We also present results on the Web Application subset of the ArtifactsBench (Zhang et al., 2025a) dataset in Appendix M.

**Models and WebGen-Agent Inference Settings.** We evaluate WebGen-Agent using a wide range of proprietary and open-source models as coding LLMs. The proprietary models we tested include Claude-3.5-Sonnet (Anthropic, 2024), DeepSeek-R1 (Guo et al., 2025a), DeepSeek-V3 (Liu et al., 2024), o3 (OpenAI, 2025a), Claude-4-Sonnet (Anthropic, 2025), Gemini-2.5-Pro (Comanici et al., 2025), and Qwen3-Coder-480B-A35B-Instruct (Yang et al., 2025a). The smaller open-source models we tested include Qwen2.5-Coder-32B-Instruct (Hui et al., 2024), Qwen3-Coder-30B-A3B-Instruct (Yang et al., 2025a), Qwen2.5-72B-Instruct (Yang et al., 2024a), Qwen2.5-Coder-7B-Instruct (Hui et al., 2024), and Qwen3-8B (Yang et al., 2025a), as well as 7B and 8B WebGenAgent-LM models trained with SFT and Step-GRPO. The maximum number of iterations is set to 20, and the model temperature is set to 0.5. We use Qwen2.5-VL-32B-Instruct as the feedback VLM for screenshot and GUI-agent testing in all the experiments. Analysis of the maximum iteration number is presented in Appendix I.

**Training Settings.** We first fine-tune Qwen2.5-Coder-7B-Instruct and Qwen3-8B on approximately seven hundred WebGen-Agent trajectories generated from website generation instructions taken from WebGen-Instruct (Lu et al., 2025b) with DeepSeek-V3 for one epoch with a learning rate of 4e-5 and a batch size of 32. This results in the models WebGenAgent-LM-7B-SFT and WebGenAgent-LM-8B-SFT, which serve as a warm start for Step-GRPO. We then train these SFT models using Step-GRPO on five hundred website generation instructions randomly sampled from WebGen-Instruct for one epoch, resulting in the final models WebGenAgent-LM-7B-Step-GRPO and WebGenAgent-LM-8B-Step-GRPO. The learning rate is set to 1e-6 with a batch size of 16. For each instruction, we sample 5 outputs. Ambiguous or underspecified instructions are manually

Table 1: The performance of WebGen-Agent with various proprietary and open-source models on WebGen-Bench (Lu et al., 2025b), compared with other code-agent systems. Results of OpenHands, Aider, and Bolt.diy are taken from Lu et al. (2025b). Accuracy is computed using a weighted score, where YES samples are weighted by 1 and PARTIAL samples by 0.5; the total score is then divided by the number of test cases. The highest Accuracy and Appearance Score are highlighted in **bold**.

| Test Name | Yes | Partial | No | Start Failed | Accuracy | Appearance Score |
|---|---|---|---|---|---|---|
| **OpenHands** | | | | | | |
| Claude-3.5-Sonnet | 18.1 | 8.3 | 58.6 | 15.0 | 22.3 | 2.6 |
| DeepSeek-R1 | 8.5 | 3.4 | 60.4 | 27.7 | 10.2 | 1.4 |
| DeepSeek-V3 | 7.4 | 3.2 | 73.9 | 15.5 | 9.0 | 1.5 |
| **Aider** | | | | | | |
| Claude-3.5-Sonnet | 19.9 | 5.9 | 42.0 | 32.1 | 22.9 | 1.9 |
| DeepSeek-R1 | 23.3 | 8.7 | 44.5 | 23.5 | 27.7 | 2.7 |
| DeepSeek-V3 | 12.5 | 3.1 | 54.3 | 30.1 | 14.1 | 1.3 |
| **Bolt.diy** | | | | | | |
| Claude-3.5-Sonnet | 22.6 | 7.6 | 64.1 | 5.7 | 26.4 | 3.0 |
| DeepSeek-R1 | 24.7 | 6.2 | 64.3 | 4.8 | 27.8 | 2.5 |
| DeepSeek-V3 | 18.5 | 4.5 | 73.9 | 3.1 | 20.8 | 2.0 |
| GPT-4o | 10.4 | 4.8 | 64.5 | 20.4 | 12.8 | 1.5 |
| o3-mini | 17.9 | 3.4 | 40.0 | 38.6 | 19.6 | 1.6 |
| Qwen2.5-Coder-32B-Inst. | 8.2 | 2.6 | 81.8 | 7.4 | 9.5 | 1.1 |
| Qwen2.5-72B-Inst. | 12.1 | 3.6 | 80.7 | 3.7 | 13.8 | 1.4 |
| WebGen-LM-7B | 24.9 | 7.1 | 68.0 | 0.0 | 28.4 | 2.5 |
| WebGen-LM-14B | 25.0 | 8.7 | 66.3 | 0.0 | 29.4 | 2.5 |
| WebGen-LM-32B | 34.2 | 8.0 | 57.8 | 0.0 | 38.2 | 2.8 |
| **WebGen-Agent** | | | | | | |
| **Proprietary Models** | | | | | | |
| Claude-3.5-Sonnet | 45.6 | 12.7 | 40.6 | 1.1 | 51.9 | 3.9 |
| DeepSeek-R1 | 40.2 | 12.4 | 45.9 | 1.5 | 46.4 | 3.8 |
| DeepSeek-V3 | 46.1 | 13.1 | 40.6 | 0.2 | 52.6 | 3.8 |
| o3 | 45.7 | 11.9 | 41.6 | 0.8 | 51.7 | 3.5 |
| Gemini-2.5-Pro | 44.5 | 12.7 | 39.4 | 3.4 | 50.9 | 3.8 |
| Claude-4-Sonnet | 48.8 | 15.3 | 33.4 | 2.5 | 56.5 | 4.1 |
| Qwen3-Coder-480B-A35B-Inst. | 50.5 | 15.3 | 34.2 | 0.0 | **58.2** | **4.3** |
| **Open-Source Models (10B+)** | | | | | | |
| Qwen2.5-Coder-32B-Inst. | 26.7 | 10.5 | 60.3 | 2.5 | 32.0 | 3.3 |
| Qwen3-Coder-30B-A3B-Inst. | 45.7 | 14.1 | 40.2 | 0.0 | **52.8** | **4.0** |
| Qwen2.5-72B-Inst. | 29.1 | 13.8 | 57.2 | 0.0 | 35.9 | 3.4 |
| WebGen-LM-14B | 32.9 | 10.2 | 38.9 | 17.9 | 38 | 3.4 |
| WebGen-LM-32B | 35.4 | 9.9 | 54.6 | 0.2 | 40.3 | 3.6 |
| **Open-Source Models (10B-)** | | | | | | |
| Qwen2.5-Coder-7B-Inst. | 10.0 | 4.8 | 60.9 | 24.3 | 12.4 | 1.6 |
| WebGen-LM-7B | 27.8 | 9.9 | 51.2 | 11.1 | 32.8 | 3.3 |
| WebGenAgent-LM-7B-SFT | 33.8 | 10.2 | 56.0 | 0.0 | 38.9 | 3.4 |
| WebGenAgent-LM-7B-Step-GRPO | 40.2 | 10.5 | 49.3 | 0.0 | **45.4** | **3.7** |
| Qwen3-8B | 29.5 | 9.1 | 61.4 | 0.0 | 34.1 | 3.2 |
| WebGenAgent-LM-8B-SFT | 32.8 | 11.6 | 55.6 | 0.0 | 38.6 | 3.4 |
| WebGenAgent-LM-8B-Step-GRPO | 37.4 | 12.1 | 50.5 | 0.0 | **43.4** | **3.6** |

filtered out. We observe that this relatively small number of high-quality instructions is sufficient for Step-GRPO training, likely due to the reliable step-level feedback from screenshots and the GUI agent. Training on more samples is costly and does not yield noticeable gains.

Table 2: Ablation study on the WebGen-Agent workflow. The configuration starts from execution-only and incrementally adds capabilities. Accuracy is computed using a weighted score, where YES samples are weighted by 1 and PARTIAL samples are weighted by 0.5; the total score is then divided by the number of test cases. The coding LLM used is DeepSeek-V3.

| Test Name | Yes | Partial | No | Start Failed | Accuracy | Appearance Score |
|---|---|---|---|---|---|---|
| Execution-only | 39.7 | 12.4 | 43.3 | 4.6 | 45.9 | 3.0 |
| Screenshot | 41.3 | 10.7 | 45.9 | 2.2 | 46.6 | 3.6 |
| Screenshot+GUI-agent | 43.0 | 13.9 | 41.3 | 1.9 | 49.9 | 3.4 |
| Screenshot+GUI-agent+Backtrack | 45.6 | 11.1 | 43.1 | 0.2 | 51.2 | 3.7 |
| **Screenshot+GUI-agent+Backtrack+Select-best** | 46.1 | 13.1 | 40.6 | 0.2 | **52.6** | **3.8** |

**Results.** The WebGen-Agent test results are presented in Tab. 1. Based on the results, we make the following observations: (1) WebGen-Agent demonstrates superior performance across various proprietary models compared to other code agent systems. On Claude-3.5-Sonnet, DeepSeek-R1, and DeepSeek-V3, WebGen-Agent significantly outperforms OpenHands, Aider, and Bolt.diy when using the same model. Across all seven proprietary models from five different providers, WebGen-Agent achieves consistently high performance, demonstrating the generalizability of the method. Qwen3-Coder-480B-A35B-Instruct achieves the highest accuracy of 58.2% and an appearance score of 4.3. (2) With 30B–72B sized open-source models, WebGen-Agent also achieves high performance. On Qwen2.5-Coder-32B-Instruct and Qwen2.5-72B-Instruct, WebGen-Agent outperforms the previous state-of-the-art, Bolt.diy, by 22.5% and 22.1% in accuracy, and by 2.2 and 2.0 in appearance scores, respectively. Qwen3-Coder-30B-A3B-Instruct achieves the best performance among 30B–72B models, with 52.8% accuracy and an appearance score of 4.0. Also, when using the WebGen-LM models, our WebGen-Agent pipeline consistently outperforms the previous SOTA, Bolt.diy, across all three model sizes. (3) Step-GRPO with Screenshot and GUI-agent Feedback significantly improves the performance of Qwen2.5-Coder-7B-Instruct and Qwen3-8B. For Qwen2.5-Coder-7B-Instruct, SFT improves accuracy from 12.4% to 38.9% and the appearance score from 1.6 to 3.4; Step-GRPO further improves accuracy from 38.9% to 45.4% and the appearance score from 3.4 to 3.7. For Qwen3-8B, SFT improves accuracy from 34.1% to 38.6% and the appearance score from 3.2 to 3.4; Step-GRPO further improves accuracy from 38.6% to 43.4% and the appearance score from 3.4 to 3.6. Additionally, when we compare WebGenAgent-LM-7B-Step-GRPO with WebGen-LM-7B, which shares the same base model, our WebGenAgent-7B model outperforms WebGen-LM-7B by 12.6% and 0.4 in accuracy and appearance score, respectively, under the WebGen-Agent pipeline. This further demonstrates the effectiveness of our Step-GRPO with Screenshot and GUI-agent Feedback training method. Qualitative analysis of SFT and Step-GRPO's effect in improving the performance is presented in Appendix J. These results demonstrate the effectiveness of our training method in improving both the functionality and appearance of the generated websites. Categorical results are presented in Tab. 10 of Appendix H.

## 3.2 ABLATION STUDIES

**Analysis of the WebGen-Agent Workflow.** We analyze various design choices in the WebGen-Agent workflow in Tab. 2. We incrementally add the designs, starting from using only the code execution response messages $\mathcal{O}$ ("Execution-only"), then gradually adding screenshot feedback $\mathcal{F}_{shot}$ ("Screenshot"), GUI-agent testing feedback $\mathcal{F}_{gui}$ ("Screenshot+GUI-agent"), the backtracking mechanism ("Screenshot+GUI-agent+Backtrack"), and finally the select-best mechanism ("Screenshot+GUI-agent+Backtrack+Select-best"), which makes up the full WebGen-Agent workflow. As shown in Tab. 2, each of these designs yields notable gains in accuracy and appearance. The GUI-agent testing contributes the largest accuracy gain of 3.3%, showing its effectiveness in guiding the functionality of the generated websites. The addition of screenshot feedback greatly improves the appearance score, raising it from 3.0 to 3.6, demonstrating its effect in enhancing website appearance. Adding GUI-agent testing slightly impairs the appearance score, likely because modifying the codebase for functional fulfillment sometimes damages the website appearance or causes errors. This negative effect is mitigated by the addition of the backtracking and select-best mechanisms. Qualitative analysis of the effect of screenshot and GUI-agent feedback is provided in

Table 3: Training strategy ablation on the Qwen2.5-Coder-7B-Instruct model. The configuration starts from the raw model and successively introduces supervised fine-tuning (SFT) and various reinforcement-learning variants. Accuracy is computed using a weighted score, where YES samples are weighted by 1 and PARTIAL samples are weighted by 0.5; the total score is then divided by the number of test cases.

| Test Name | Yes | Partial | No | Start Failed | Accuracy | Appearance Score |
|---|---|---|---|---|---|---|
| No Additional Training | 10.0 | 4.8 | 60.9 | 24.3 | 12.4 | 1.6 |
| SFT for 1 Epoch | 33.8 | 10.2 | 56.0 | 0.0 | 38.9 | 3.4 |
| SFT for 2 Epochs | 32.1 | 14.2 | 53.5 | 0.2 | 39.3 | 3.4 |
| SFT with 1.4K Samples for 1 Epoch | 33.5 | 11.6 | 54.9 | 0.0 | 39.3 | 3.4 |
| Naive Outcome GRPO | 38.0 | 9.0 | 53.0 | 0.0 | 42.5 | 3.5 |
| Step-GRPO w/ Cumulative Advantage | 32.6 | 12.2 | 55.2 | 0.0 | 38.7 | 3.5 |
| Step-GRPO w/ Screenshot Reward Only | 34.9 | 10.5 | 53.9 | 0.6 | 40.2 | 3.5 |
| Step-GRPO w/ GUI-agent Reward Only | 34.8 | 11.3 | 53.6 | 0.3 | 40.4 | 3.4 |
| **Step-GRPO w/ Screenshot+GUI-agent (ours)** | 40.2 | 10.5 | 49.3 | 0.0 | **45.4** | **3.7** |

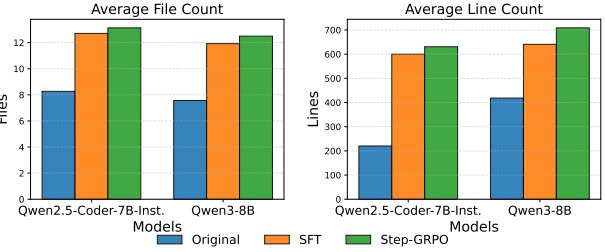

Figure 3: Comparison of the average file count and average line count among the original, SFT, and Step-GRPO models for Qwen2.5-Coder-7B-Instruct and Qwen3-8B.

Appendix K. Also, as shown in Tab. 9 of Appendix G, a manual inspection indicates that 98.3% of the sampled instructions achieve high coverage of the requirements.

**Analysis of Step-GRPO with Screenshot and GUI-agent Feedback.** We analyze the design choices in the Step-GRPO with Screenshot and GUI-agent Feedback training process in Tab. 3. The first line shows the result of the Qwen2.5-Coder-7B-Instruct model with no additional training. The analysis based on Tab. 3 is as follows: (1) The second and third lines present SFT training for one epoch and two epochs, showing that training for two epochs does not notably improve performance compared to training for only one epoch. Therefore, we train for only one epoch in the SFT stage. (2) The fifth and sixth lines show the results of using naive outcome GRPO and Step-GRPO with cumulative advantage. The rewards in these two variants are the same as in our final design ($Score_{shot} + Score_{gui}$); only the advantage computation method differs. Naive outcome GRPO uses the maximum value of the step-level rewards in a trajectory as the outcome reward, setting the advantages to the normalized outcome rewards. Step-GRPO with cumulative advantage calculates the advantage of each token as the sum of the normalized rewards from the subsequent steps, as introduced in Shao et al. (2024). Both GRPO advantage computation variants perform notably worse than our final Step-GRPO setting. (3) The seventh and eighth lines present the results of using only the screenshot scores ($Score_{shot}$) or only the GUI-agent testing scores ($Score_{gui}$) as the rewards. Both are lower than using $Score_{shot} + Score_{gui}$, demonstrating the necessity of incorporating both types of feedback. We also gather statistics on the average file count and average line count for the Original, SFT, and Step-GRPO models, as shown in Fig. 3. For both Qwen2.5-Coder-7B-Instruct and Qwen3-8B, the average file count and average line count consistently increase after SFT and Step-GRPO training. This shows that both the SFT and Step-GRPO stages increase the complexity of the generated websites, which is consistent with their improved performance. (4) We also generate seven hundred new trajectories from instructions in WebGen-Instruct with DeepSeek-V3. Com-

Table 4: Impact of the feedback VLM on the performance of WebGen-Agent on WebGen-Bench. The highest Accuracy and Appearance Score are highlighted in **bold**.

| Coding LLM | Feedback VLM | Yes | Partial | No | Start Failed | Accuracy | Appearance Score |
|---|---|---|---|---|---|---|---|
| Qwen2.5-VL-32B-Inst. | Qwen2.5-VL-32B-Inst. | 4.5 | 2.2 | 78.8 | 14.5 | 5.6 | 1.3 |
| DeepSeek-V3 | GPT-4o | 46.4 | 11.4 | 42.0 | 0.2 | 52.1 | 3.6 |
| **DeepSeek-V3** | **Qwen2.5-VL-32B-Inst.** | 46.1 | 13.1 | 40.6 | 0.2 | **52.6** | **3.8** |

Table 5: Comparison of using Qwen2.5-VL-7B-Instruct and Qwen2.5-VL-32B-Instruct as the feedback VLM in the Step-GRPO with Screenshot and GUI-agent Feedback training process.

| Training Method | Feedback VLM | Yes | Partial | No | Start Failed | Accuracy | Appearance Score |
|---|---|---|---|---|---|---|---|
| SFT for 1 Epoch | – | 33.8 | 10.2 | 56 | 0 | 38.9 | 3.4 |
| Step-GRPO | Qwen2.5-VL-32B-Instruct | 40.2 | 10.5 | 49.3 | 0 | 45.4$_{(+6.5)}$ | 3.7$_{(+0.3)}$ |
| Step-GRPO | Qwen2.5-VL-7B-Instruct | 33.7 | 10.7 | 54.7 | 0.9 | 39.0$_{(+0.1)}$ | 3.3$_{(-0.1)}$ |

bining them with the original trajectories, we created a scaled training set of approximately 1.4K trajectories. As shown in the fourth line in Tab. 3, increasing the SFT data to twice the original size does not significantly improve the accuracy and appearance score of Qwen2.5-Coder-7B-Instruct. This demonstrates that simple distillation has a clear upper bound and cannot achieve the same gain in accuracy and appearance score as our Step-GRPO with Screenshot and GUI-agent Feedback training method.

**Analysis of the Coding LLM and Feedback VLM.** We analyze the choice of the coding LLM and feedback VLM in Tab. 4. In our experiments, we use a relatively small and inexpensive VLM, Qwen2.5-VL-32B-Instruct, to provide screenshot and GUI-agent testing feedback, while employing a strong LLM capable of generating high-quality code, such as DeepSeek-V3, as the coding LLM. As shown in the second row of Tab. 4, replacing Qwen2.5-VL-32B-Instruct with a proprietary VLM, GPT-4o, as the feedback VLM does not notably improve the accuracy or the appearance score. This is likely because that there is little gap between Qwen2.5-VL-32B-Instruct and GPT-4o in providing appearance and functional judgements, as shown in Tab. 7 and Tab. 8 of Appendix F. This demonstrates that Qwen2.5-VL-32B-Instruct is already sufficient for providing accurate screenshot and GUI-agent testing feedback, while being more cost-effective than proprietary VLMs. As shown in the first row of Tab. 4, replacing DeepSeek-V3 with Qwen2.5-VL-32B-Instruct results in significantly worse performance, indicating that the coding LLM cannot be replaced by smaller open-source VLMs. The design choice of decoupling the coding LLM and feedback VLM ensures that code is generated by a strong LLM to maintain quality, while screenshot and GUI-agent testing feedback is handled by a smaller open-source VLM for cost efficiency. As shown in Tab. 5, using a weaker VLM greatly reduces the effectiveness of the GRPO training, as it introduces noise and inaccuracies in the RL supervision signal. Using Qwen2.5-VL-7B-Instruct as the feedback VLM provides almost no gain in the GRPO training process, demonstrating the major role the VLM plays in our WebGen-Agent pipeline and Step-GRPO with Screenshot and GUI-agent Feedback training approach. Further analysis of the accuracy of the screenshot and GUI-agent scores provided by the feedback VLM is included in Tab. 6 of Appendix E, demonstrating the reliability of the scores, and quantitative evaluation results of Qwen2.5-VL-32B-Instruct as a visual judge for images and GUI agent trajectories are provided in Appendix F.

## 4 RELATED WORK

**Visual Code Generation.** Code generation that is associated with visual effects exists in a wide range of application scenarios, such as web page development (Lu et al., 2025b; Xu et al., 2025) and GitHub issue fixing (Yang et al., 2024d; Guo et al., 2025b). Previous work has proposed various ways to treat visual elements in code generation and other reasoning-intensive tasks (Su et al., 2025), such as generating code to represent images in problem statements (Huang et al., 2025; Wang et al.,

2025b) and using natural language to describe images (Zhang et al., 2024b). We also apply natural language descriptions when providing screenshot feedback. More related to our work, a line of studies (Guo et al., 2024; Si et al., 2025; Yun et al., 2024; Beltramelli, 2017; Sun et al., 2025; Gui et al., 2025; Laurençon et al., 2024; Wan et al., 2024) explores MLLMs' ability to reconstruct single-file HTML code from webpage screenshots. Other studies benchmark MLLMs' performance in implementing interactive elements in existing web projects (Xiao et al., 2025a) or performing web development tasks in a pre-defined sequential manner with detailed technical settings (Xiao et al., 2025b; Xu et al., 2025). The web development tasks in these works are often solved in a single HTML file (Zhang et al., 2025a) or contain rigid pipelines (Xu et al., 2025), which are more suitable for testing MLLMs rather than code agents for end-to-end, repository-level website development, as proposed in our work. Therefore, we evaluate our agent workflow with WebGen-Bench (Lu et al., 2025b), which measures a code agent's ability to create multi-file website codebases from scratch and includes diverse website generation instructions.

**Code Agents.** Equipped with various tools and powered by LLMs (Soni et al., 2025; Yao et al., 2023; Zhang et al., 2024a), code agents can perform a variety of tasks, such as developing websites (Lu et al., 2025b) and fixing GitHub issues (Jimenez et al., 2024; Yang et al., 2024c). Some code agents specialize in a specific field, such as bug fixing (Zhang et al., 2024c) or machine learning (Jiang et al., 2025). Others, such as OpenHands (Wang et al., 2024) and Aider (Aider-AI, 2024), are general-purpose code agents that are not limited to a single field, though their performance on a specific task might not match that of specialist code agents (Lu et al., 2025b). Their relatively low performance in website development is because their tools are not suited to website development, and their general system prompts. Similar to our work, Bolt.diy (stackblitz labs, 2024) specializes in multi-file website generation. While it executes the generated codebase, the executed results are, not returned to the model. In contrast, our WebGen-Agent is a code agent specializing in end-to-end website generation, with screenshot feedback and GUI-agent testing features specifically designed for this task, achieving state-of-the-art performance.

**Fine-tuning and Reinforcement Learning for Agents.** Supervised fine-tuning (Pan et al., 2025; Yang et al., 2025b) and reinforcement learning (Dong et al., 2025; Qian et al., 2025) are two methods widely used to improve the agentic and tool-calling abilities of LLMs. In the field of code agents, various works (Pan et al., 2025; Yang et al., 2025b; Zhang et al., 2025b; Wang et al., 2025a; Ma et al., 2024; Xie et al., 2025; Jain et al., 2025; Guo et al., 2025c; Ma et al., 2025a) leverage supervised fine-tuning combined with software engineering data synthesis and rejection sampling to improve the performance of open-source models. Similar to these works, we also use rejection sampling and supervised fine-tuning in the warm-up stage before the GRPO training. Other works use reinforcement learning with rewards acquired through comparison with the ground truth (Wei et al., 2025a; Ma et al., 2025c; Zhuang et al., 2025), determined by the code execution output (Gehring et al., 2025; Ma et al., 2025b; Golubev et al., 2025), or dependent on task success (Wei et al., 2025b; Lu et al., 2025a; Chen et al., 2025). These works either use outcome supervision, which provides sparse training signals, or require detailed ground truth to provide step supervision, which is rigid and difficult to obtain. In contrast to these methods, our work leverages screenshot and GUI-agent testing scores at each step—which are inherent in the WebGen-Agent pipeline—to provide accurate step-level supervision in GRPO training.

## 5 CONCLUSION

In this paper, we introduce WebGen-Agent, a code agent that leverages screenshot and GUI-agent testing feedback, combined with backtracking and select-best mechanisms, to iteratively generate websites with appealing appearance and smooth functionality. We also propose Step-GRPO with Screenshot and GUI-agent Feedback, which leverages inherent screenshot and GUI-agent testing scores to provide step-level supervision in the GRPO training process. Testing WebGen-Agent on WebGen-Bench shows significant improvements across a wide range of proprietary and open-source LLMs compared to other code agent systems. WebGen-Agent with Qwen3-Coder-480B-A35B-Instruct achieves the best performance, with an accuracy of 58.2% and an appearance score of 4.3. Training Qwen2.5-Coder-7B-Instruct and Qwen3-8B first with supervised fine-tuning and then with Step-GRPO with Screenshot and GUI-agent Feedback notably improves accuracies and appearance scores, demonstrating the effectiveness of our training approach.

## 6 REPRODUCIBILITY STATEMENT

To ensure reproducibility, we release the WebGen-Agent workflow code, along with the training code and data for Step-GRPO with Screenshot and GUI-Agent Feedback, as well as the weights of the WebGenAgent-LM models. The complete code base and datasets are provided in the supplementary material accompanying this paper. Details of the agent workflow and of all prompts used to deliver multi-level feedback are presented in Section 2.1, Appendix B, and Appendix C. The training procedure for Step-GRPO with Screenshot and GUI-Agent Feedback is described in Section 2.2 and Section 3.1. We also report a manual inspection of the screenshot and GUI-agent scores in Appendix E, and we assess the comprehensiveness of the GUI-Agent testing instructions in Appendix G. Collectively, these resources ensure that our findings are transparent, robust, and independently verifiable.

## 7 ACKNOWLEDGEMENTS

This study was supported in part by National Key R&D Program of China Project 2022ZD0161100, in part by the Centre for Perceptual and Interactive Intelligence, a CUHK-led InnoCentre under the InnoHK initiative of the Innovation and Technology Commission of the Hong Kong Special Administrative Region Government, in part by NSFC-RGC Project N_CUHK498/24, and in part by Guangdong Basic and Applied Basic Research Foundation (No. 2023B1515130008, XW).

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

## A    LIMITATIONS AND FUTURE WORK

WebGen-Agent is specifically designed to generate websites based on natural language instructions from non-expert users. We do not consider website response speed or complex network conditions when generating and evaluating the websites; these are interesting questions for future work. In the supervised fine-tuning and Step-GRPO experiments, we train only 7B- and 8B-parameter models due to limited computing power and GPU memory, as Step-GRPO training would take more than 24 hours on 16 NVIDIA A800 GPUs, and we currently do not have enough GPUs to train larger models. The results on the 7B and 8B models show great potential for our method, and we plan to apply our training approach to 30B–72B models in the future. Moreover, the functionality reward depends on the capability of the GUI agent. While our experiments show that even with the current WebVoyager agent, the functionality reward provides meaningful and consistent feedback and learning signals, we also note that our design is agent-agnostic: as GUI agents improve, the functionality reward naturally becomes more accurate and more informative. This means that the performance of our method can continue to improve with the advancement of GUI agents, which can be explored in future work.

## B    WEBGEN-AGENT ALGORITHM

Algorithm 1 demonstrates the WebGen-Agent inference workflow in detail. Algorithms 2 and 3 are two helper functions for Algorithm 1, presented separately for clarity.

## C    WEBGEN-AGENT PROMPTS

The prompts for acquiring screenshot and GUI-agent testing feedback are presented in Fig. 4, Fig. 5, Fig. 6, and Fig. 7.

## D    EXAMPLES OF WEBGEN-AGENT TRAJECTORIES

To demonstrate the WebGen-Agent workflow in a straightforward way, we present three example trajectories in Fig. 8, Fig. 9, and Fig. 10. As shown in these examples, WebGen-Agent iteratively improves the appearance and functionality of the generated website based on screenshot and GUI-agent testing feedback.

## E    ACCURACY OF SCREENSHOT AND GUI-AGENT TESTING SCORES

To analyze the accuracy of the screenshot and GUI-agent testing scores given by the feedback VLM in the WebGen-Agent workflow, we evaluated the results of Claude-4-Sonnet, Qwen3-Coder-30B-A3B-Instruct, Qwen3-Coder-480B-A35B-Instruct, and DeepSeek-V3 as coding LLMs, with Qwen2.5-VL-32B-Instruct as the feedback VLM, as well as DeepSeek-V3 as the coding LLM and GPT-4o as the feedback VLM. We manually verified the accuracy of the screenshot and GUI-agent testing scores. Human annotators were provided with the score and the screenshot or GUI-agent trajectory at each step and asked to judge whether the score was accurate. If the score was inaccurate, they provided the correct score. The results are presented in Table 6. The manual testing results serve as the ground truth and require precision; therefore, three human testers independently annotated the results, and we assessed the consistency of their annotations. If the annotations of a test case are inconsistent, a fourth human tester is tasked with re-examining the test case and the inconsistent annotations to decide on a final annotation. The annotation guidelines are presented in Fig. 12 and Fig. 14, while the annotation interface screenshots are presented in Fig. 11 and Fig. 13. The testers are student volunteers with bachelor's degrees in computer science-related majors who are studying for a master's degree. We also compute Cohen's $\kappa$ between the three human annotators to evaluate inter-annotator agreement, as shown in Table A and Table B below (Cohen's $\kappa$ 1–2 denotes Cohen's $\kappa$ between annotator 1 and annotator 2; the others are similar). All of Cohen's $\kappa$ values are above 0.8, with most above 0.9, demonstrating strong inter-annotator agreement.

The accuracies of the screenshot scores across all experiments ranged from 93% to 96%, while the accuracies of the GUI-agent scores ranged from 89% to 93%. The standard errors of the screenshot

**Prompt:**
You are given a single website screenshot as input.
**Task**

1. Examine the screenshot closely for any rendering or runtime errors (e.g., "404 Not Found", stack traces, missing styles, blank areas).

2. Decide whether the screenshot *shows a rendering or runtime error*.

   - If **yes**, set "is_error": true, extract or paraphrase the visible error message(s) into "error_message", and leave "screenshot_description" empty.

   - If **no**, set "is_error": false, leave "error_message" as an empty string (""), and write a concise but thorough "screenshot_description" that covers:

     – Overall layout (e.g., header/sidebar/footer, grid, flex, single-column).
     – Key UI components (navigation bar, buttons, forms, images, cards, tables, modals, etc.).
     – Color scheme and visual style (dominant colors, light/dark theme, gradients, shadows).
     – Visible content and text (headings, labels, sample data).
     – Notable design details (icons, spacing, font style) that help someone understand what the page looks like).

3. Suggest ways to improve the appearance of the website, for example:

   - Separate incorrectly overlapping components.

   - Adjust layout to avoid large blank areas.

   - Adjust text or background color to avoid text color being too similar to the background color.

   - If no improvement is necessary, leave "suggestions" as an empty string (""); otherwise, briefly list the suggestion(s) in "suggestions".

4. Grade the response.

**Output format (valid JSON)**

```json
{
  "is_error": <boolean>,
  "error_message": "<string>",
  "screenshot_description": "<string>",
  "suggestions": "<string>"
}
```

Return **only** this JSON object—no additional commentary, markdown, or code fences.

Figure 4: The prompt for generating the description and suggestions based on the website screenshot.

**Prompt:**

You are tasked with evaluating the design of a webpage. Grade the webpage's appearance on a scale of 0 to 5 (5 being highest), considering the following criteria:

- **Successful Rendering:** Are there any components in the page or is it completely blank? Does the webpage render correctly without visual errors? Are colors, fonts, and components displayed as specified?

- **Content Relevance:** Does the design align with the website's purpose and user requirements? Are elements (e.g., search bars, report formats) logically placed and functional?

- **Layout Harmony:** Is the arrangement of components (text, images, buttons) balanced, intuitive, and clutter-free?

- **Modernness & Beauty:** Does the design follow contemporary trends (e.g., minimalism, responsive layouts)? Are colors, typography, and visual hierarchy aesthetically pleasing?

**Grading Scale:**

- **0 (Blank Page):** The screenshot is completely blank or does not contain any visible content. It may only have a background color or display an error message.

- **1 (Poor):** Major rendering issues (e.g., broken layouts, incorrect colors). Content is irrelevant or missing. Layout is chaotic. Design is outdated or visually unappealing.

- **2 (Below Average):** Partial rendering with noticeable errors. Content is partially relevant but poorly organized. Layout lacks consistency. Design is basic or uninspired.

- **3 (Average):** Mostly rendered correctly with minor flaws. Content is relevant but lacks polish. Layout is functional but unremarkable. Design is clean but lacks modern flair.

- **4 (Good):** Rendered well with no major errors. Content is relevant and logically organized. Layout is harmonious and user-friendly. Design is modern and visually appealing.

- **5 (Excellent):** Flawless rendering. Content is highly relevant, intuitive, and tailored to user needs. Layout is polished, responsive, and innovative. Design is cutting-edge, beautiful, and memorable.

**Task:**

Review the provided screenshot(s) of the webpage. Provide a concise analysis of a few sentences and then assign a grade (0–5) based on your analysis. Highlight strengths, weaknesses, and how well the design adheres to the specifications.

**Your Response Format**

```json
{
  "analysis": "<string>",
  "grade": <int>
}
```

**Your Response:**

Figure 5: Prompt for evaluating the visual quality of a webpage and generating an appearance score.

---

**Algorithm 1** WebGen-Agent

---

**Require:** Initial instruction $\mathcal{I}$, maximum steps $T$
**Ensure:** Final codebase $\mathcal{C}^\star$
1: $\mathcal{T} \leftarrow [\mathcal{I}]$        ▷ trajectory: instruction, **edit**, feedback, . . .
2: $Steps \leftarrow \emptyset$        ▷ archive of step snapshots
3: $\mathcal{C} \leftarrow \emptyset$        ▷ current *codebase*
4: $t \leftarrow 1, \; consecErr \leftarrow 0$
5: **while** $t \leq T$ **do**
6:     $\Delta\mathcal{C}_t \leftarrow \text{GENERATEEDIT}(\mathcal{T})$
7:     $\mathcal{T} \mathrel{+}= \Delta\mathcal{C}_t$
8:     $\mathcal{C} \leftarrow \text{APPLYEDIT}(\mathcal{C}, \Delta\mathcal{C}_t)$
9:     $\mathcal{O} \leftarrow \text{EXECUTE}(\mathcal{C})$
10:     **if** $\mathcal{O} = error$ **then**
11:         $\mathcal{T} \mathrel{+}= \mathcal{O}$
12:         $consecErr \leftarrow consecErr + 1$
13:         **if** $consecErr = 5$ **then**
14:             $\langle t^\star, \mathcal{C}^\star, *, * \rangle \leftarrow \text{SELECTBESTSTEP}(Steps)$
15:             $\mathcal{C} \leftarrow \mathcal{C}^\star$        ▷ restore codebase
16:             $\mathcal{T} \leftarrow \text{TRUNCATE}(\mathcal{T}, t^\star)$
17:             $t \leftarrow t^\star + 1, \; consecErr \leftarrow 0$
18:         **else**
19:             $t \leftarrow t + 1$
20:         **end if**
21:         **continue**
22:     **else**
23:         $consecErr \leftarrow 0$
24:     **end if**
25:     $img \leftarrow \text{SCREENSHOT}(\mathcal{C})$
26:     $\langle desc, sugg_{\text{shot}}, score_{\text{shot}} \rangle \leftarrow \text{VLM\_JUDGE}(img)$
27:     $\mathcal{T} \mathrel{+}= \langle desc, sugg_{\text{shot}} \rangle$
28:     $goNext \leftarrow \text{AGENTDECISION}(\mathcal{T})$
29:     **if not** $goNext$ **then**
30:         $t \leftarrow t + 1;$ **continue**
31:     **end if**
32:     $\langle pass, sugg_{\text{gui}}, score_{\text{gui}} \rangle \leftarrow \text{GUI\_AGENT}(\mathcal{C})$
33:     $\mathcal{T} \mathrel{+}= \langle pass, sugg_{\text{gui}} \rangle$
34:     $Steps \mathrel{+}= \langle t, \mathcal{C}, score_{\text{shot}}, score_{\text{gui}} \rangle$
35:     **if** $pass$ **then**
36:         **break**
37:     **else**
38:         $t \leftarrow t + 1$
39:     **end if**
40: **end while**
41: $\langle *, \mathcal{C}^\star, *, * \rangle \leftarrow \text{SELECTBESTSTEP}(Steps)$
42: **return** $\mathcal{C}^\star$

---

**Algorithm 2** SELECTBESTSTEP

---

**Require:** $Steps = \{\langle t, \mathcal{C}, score_{\text{shot}}, score_{\text{gui}} \rangle\}$
1: $g_{\max} \leftarrow \max_{s \in Steps} score_{\text{gui}}$
2: $\mathcal{S}_g \leftarrow \{s \mid score_{\text{gui}} = g_{\max}\}$
3: **return** $\arg\max\limits_{s \in \mathcal{S}_g} score_{\text{shot}}$

---

scores range from 0.20 to 0.26, while the standard errors of the GUI-agent scores range from 0.31 to 0.44. This demonstrates that the scores are highly accurate, supporting the effectiveness of the WebGen-Agent workflow and the Step-GRPO training process. Compared with using Qwen2.5-VL-32B-Instruct, using GPT-4o as the feedback VLM only marginally improved the screenshot score accuracy from 94.8% to 95.5% and the GUI-agent score accuracy from 91.2% to 92.2%. This shows that Qwen2.5-VL-32B-Instruct is sufficient for the task while being significantly more cost-effective.

---

**Algorithm 3** TRUNCATE

---

**Require:** Trajectory $\mathcal{T}$, step id $t^\star$
 1: **return** prefix of $\mathcal{T}$ ending just after the edit and feedback of step $t^\star$

---

> Based on the original website development instruction, you should identify **all the requirements** of the website generation and create a comprehensive instruction for a web-navigation GUI agent to test the generated website. The following is an example of triggering the GUI agent testing based on the original instruction:
>
> *Example*
>
> Original instruction:
> Please implement a self-driving tour website that provides self-driving tour products and services. The website should have functionalities for browsing self-driving tour routes, booking self-driving tour hotels, and self-help self-driving tour packages. Users should be able to browse different types of self-driving tour routes, book hotels and packages, and query self-driving club information. The website should also provide search and filtering functions to help users quickly find the self-driving tour products they need. Define background as cream; define components with dark teal.
>
> ```
> <boltAction type="gui_agent_test">
> ```
> Verify cream background and dark-teal buttons. Browse different types of self-driving tour routes, book hotels and packages, and query self-driving club information. Search and filter for self-driving tour products.
> ```
> </boltAction>
> ```
>
> The following is the original website development instruction:
>
> ```
> <instruction>{instruction}</instruction>
> ```
>
> Trigger the GUI agent testing based on the original instruction in a way similar to the example. **Do not generate additional comments.**

Figure 6: Prompt for generating a GUI-agent testing instruction from the original website specification.

## F    RELIABILITY OF THE CHOSEN VLM

To evaluate the reliability of the screenshot and GUI agent feedback, we test the performance of our chosen VLM, Qwen2.5-VL-32B-Instruct, on PairBench (Feizi et al., 2025) and AgentReward-Bench (Lù et al., 2025), and present the results in Tab. 7 and Tab. 8 below. Results of the closed-source models are taken from Feizi et al. (2025) and Lù et al. (2025). Tab. 7 shows the aggregated MMScore, 1-RelaxSym, Smoothness, and Controllability over all four data splits in PairBench. Tab. 8 shows the VLM judge performance for predicting success, mainly measured with precision (Lù et al., 2025).

As shown in Tab. 7, Qwen2.5-VL-32B-Instruct outperforms closed-source models such as GPT-4o and Gemini-1.5-Pro on three out of four metrics. Notably, on MMScore, which is the main metric in the PairBench paper (Feizi et al., 2025), Qwen2.5-VL-32B-Instruct outperforms GPT-4o by 20.03%. This shows that Qwen2.5-VL-32B-Instruct is good at comparing similar images, making it a suitable choice for providing screenshot feedback and appearance scores in the WebGen-Agent pipeline.

Additionally, as demonstrated in Tab. 8, Qwen2.5-VL-32B-Instruct outperforms GPT-4o, GPT-4o-mini, and Qwen2.5-VL-72B-Instruct in Overall Precision, which is the main metric used in Lù et al. (2025) to judge a model's ability to predict the success of web agent trajectories. Its performance is only 1.1% lower than Claude-3.7-Sonnet, while being much more cost-effective than the closed-source models. On the five data splits, Qwen2.5-VL-32B-Instruct achieves the highest performance on VWA (VisualWebArena) and Wk++ (WorkArena++), and is only lower than the highest precision on Work (WorkArena) by 0.5%. This demonstrates that Qwen2.5-VL-32B-Instruct is a reliable

**Prompt:** You are given a GUI-agent testing trajectory.

**The GUI agent testing trajectory:**

**GUI-agent Testing Instruction:**
{gui_instruction}

**Trajectory:**
{result}

**Task**

1. Examine the trajectory for any failed actions that indicate a problem in the website design.

2. Decide whether the GUI-agent testing trajectory reveals any flaw in the website implementation.
   - If **yes**, set `"test_passed": true`, and leave `"improvement_suggestions"` empty.
   - If **no**, set `"test_passed": false`, and write a concise but thorough `"improvement_suggestions"` that covers the suggested improvements targeting the problems revealed by the testing result.

3. Evaluate the results of the GUI-agent test run and assign **one integer grade from 1 to 5**:
   - 1: The vast majority of tested functions fail or behave incorrectly.
   - 2: Many functions fail; only a few behave as expected.
   - 3: About half of the functions work as expected; success is mixed.
   - 4: Most functions work as expected; only minor issues remain.
   - 5: All tested functions work exactly as expected; no issues observed.

   Assign the grade to `"grade"`.

**Output format (valid JSON)**

```json
{
  "test_passed": <boolean>,
  "improvement_suggestions": "<string>",
  "grade": <int>
}
```

You can first make a short analysis of two or three sentences, then output this JSON object.

Figure 7: Prompt for evaluating GUI-agent testing trajectories and providing improvement suggestions.

Table 6: Accuracy of the screenshot and GUI-agent scores using human annotation as ground truth. For every experiment we report the accuracy together with the standard error and inter-rater agreement (Cohen's $\kappa$). $\kappa_{1-2}$ denotes Cohen's $\kappa$ between annotator 1 and annotator 2; the others are similar.

| Score Type | Coding LLM | Feedback VLM | Acc. (%) | Std. Err. | $\kappa_{1-2}$ | $\kappa_{1-3}$ | $\kappa_{2-3}$ | Avg. $\kappa$ |
|---|---|---|---|---|---|---|---|---|
| Screenshot | Claude-4-Sonnet | Qwen2.5-VL-32B-Inst. | 93.6 | 0.25 | 0.959 | 0.964 | 0.933 | 0.952 |
| Screenshot | Qwen3-Coder-30B-A3B-Inst. | Qwen2.5-VL-32B-Inst. | 93.9 | 0.26 | 0.964 | 0.984 | 0.978 | 0.975 |
| Screenshot | Qwen3-Coder-480B-A35B-Inst. | Qwen2.5-VL-32B-Inst. | 95.6 | 0.20 | 0.948 | 0.953 | 0.912 | 0.938 |
| Screenshot | DeepSeek-V3 | Qwen2.5-VL-32B-Inst. | 94.8 | 0.22 | 0.968 | 0.982 | 0.985 | 0.979 |
| Screenshot | DeepSeek-V3 | GPT-4o | 95.5 | 0.20 | 0.980 | 0.986 | 0.970 | 0.979 |
| GUI agent | Claude-4-Sonnet | Qwen2.5-VL-32B-Inst. | 90.1 | 0.31 | 0.973 | 0.946 | 0.932 | 0.950 |
| GUI agent | Qwen3-Coder-30B-A3B-Inst. | Qwen2.5-VL-32B-Inst. | 91.4 | 0.44 | 0.933 | 0.885 | 0.941 | 0.920 |
| GUI agent | Qwen3-Coder-480B-A35B-Inst. | Qwen2.5-VL-32B-Inst. | 89.6 | 0.41 | 0.967 | 0.971 | 0.943 | 0.960 |
| GUI agent | DeepSeek-V3 | Qwen2.5-VL-32B-Inst. | 91.2 | 0.36 | 0.947 | 0.960 | 0.911 | 0.939 |
| GUI agent | DeepSeek-V3 | GPT-4o | 92.2 | 0.33 | 0.928 | 0.977 | 0.948 | 0.951 |

judge of the GUI agent trajectories and can provide accurate functional feedback for our WebGen-Agent pipeline.

Table 7: Comparison of the aggregated MMScore, 1-RelaxSym, Smoothness, and Controllability over all four data splits in PairBench (Feizi et al., 2025). Results of the closed-source models are taken from Feizi et al. (2025).

| Model | MMScore (%) | 1-RelaxSym (%) | Smoothness | Controllability (%) |
|---|---|---|---|---|
| GPT-4o-mini | 48.28 | 89.07 | 1.59 | 85.48 |
| GPT-4o | 53.95 | 91.53 | 1.54 | 72.77 |
| Gemini-1.5-Flash | 56.55 | **93.19** | 1.34 | 74.54 |
| Gemini-1.5-Pro | 52.60 | 88.72 | 1.17 | 88.09 |
| Qwen2.5-VL-32B-Instruct | **73.98** | 91.30 | **2.86** | **93.37** |

# G   ANALYSIS OF THE COMPREHENSIVENESS AND ACCURACY OF GUI-AGENT TESTING INSTRUCTIONS

To analyze the comprehensiveness of the GUI-agent testing instructions generated by the agent, we manually evaluated the instructions from the experiment runs using Claude-4-Sonnet, Qwen3-Coder-30B-A3B-Instruct, Qwen3-Coder-480B-A35B-Instruct, and DeepSeek-V3. We graded each GUI-agent instruction on a 1–5 scale, determined by how completely the instruction translates each website requirement into concrete GUI-agent checks. The grading guidelines are presented in Fig. 16, and the annotator interface screenshot is presented in Fig. 15.

As shown in Tab. 9, 77.2% of the GUI-agent testing instructions across the four models receive a score of 5 (Complete, $\approx$ 100% of requirements). Instructions with a score of 4 or higher (High, 75–90%) account for 98.3% of the total, while only 1.7% receive a score of 3 (Moderate, 50–75%); none score below 3. These results indicate that the GUI-agent instructions comprehensively cover most of the website requirements.

To analyze the accuracy of the generated test cases, we also randomly sampled 100 test cases, and inspected them manually. We judge the test case of a website development instruction accuracy if it fulfills the following two conditions: (1) The test case covers all the requirements in the instruction.(2) Each part of the test case corresponds to a requirement in the instruction. The accuracy of the 100 test cases is 87%. Among the 100 test cases, only 13 show slight mismatches with the website development instructions, further demonstrating the effectiveness of the generated test cases.

Table 8: Detailed precision, recall and F1 comparison across models.

| Model | Overall Precision | Overall Recall | Overall F1 | AB Precision | VWA Precision | WA Precision | Work Precision | Wk++ Precision |
|---|---|---|---|---|---|---|---|---|
| Claude-3.7-Sonnet | **69.4** | 76.3 | 72.7 | 71.4 | 64.8 | 69.3 | 85.3 | **66.7** |
| GPT-4o | 68.1 | 80.3 | **73.7** | 77.8 | 60.7 | **69.9** | **93.8** | 59.6 |
| GPT-4o-mini | 64.5 | 78.3 | 70.8 | **80.0** | 57.4 | 66.9 | 90.3 | 54.8 |
| Qwen2.5-VL-72B-Instruct | 64.5 | **86.1** | 73.7 | 70.0 | 58.5 | 62.9 | **93.8** | 64.4 |
| Qwen2.5-VL-32B-Instruct | 68.3 | 79.7 | 73.5 | 70.0 | **68.2** | 61.8 | 93.3 | **66.7** |

Table 9: Distribution (%) of human scores regarding the comprehensiveness of the GUI-agent testing instructions and the resulting average score, and inter-rater agreement (Cohen's $\kappa$). The definition of the scores is presented in Fig. 16. The scores range from 1 to 5. $\kappa_{1\text{-}2}$ denotes Cohen's $\kappa$ between annotator 1 and annotator 2, the others are similar.

| Model | 5 | 4 | 3 | 2 | 1 | Avg. Score | $\kappa_{1\text{-}2}$ | $\kappa_{1\text{-}3}$ | $\kappa_{2\text{-}3}$ | Avg. $\kappa$ |
|---|---|---|---|---|---|---|---|---|---|---|
| Claude-4-Sonnet | 84.2 | 13.9 | 2.0 | 0.0 | 0.0 | 4.82 | 0.873 | 0.807 | 0.930 | 0.870 |
| DeepSeek-V3 | 73.3 | 24.8 | 2.0 | 0.0 | 0.0 | 4.71 | 0.940 | 0.922 | 0.864 | 0.908 |
| Qwen3-Coder-30B-A3B-Inst. | 75.2 | 23.8 | 1.0 | 0.0 | 0.0 | 4.74 | 0.914 | 0.914 | 0.880 | 0.903 |
| Qwen3-Coder-480B-A35B-Inst. | 76.2 | 21.8 | 2.0 | 0.0 | 0.0 | 4.74 | 0.868 | 0.822 | 0.860 | 0.850 |
| **Total** | **77.2** | **21.0** | **1.7** | **0.0** | **0.0** | **4.75** | **0.898** | **0.881** | **0.871** | **0.883** |

## H    CATEGORICAL RESULTS

Tab. 10 shows the categorical results of WebGen-Agent with various proprietary and open-source models on WebGen-Bench. As shown in the table, WebGen-Agent consistently achieves superior performance across all instruction and test-case categories compared to other code agent systems. For both the 7B and 8B models, Step-GRPO improves performance in most categories compared to the original instruct model and the SFT model. This demonstrates the effectiveness of the WebGen-Agent workflow and the Step-GRPO training process, which incorporates screenshots and GUI-agent feedback.

## I    ANALYSIS OF MAXIMUM ITERATION NUMBERS

To analyze the effect of the maximum iteration number parameter on the performance of WebGen-Agent, we test the accuracy, appearance score, and the percentage of samples that exceed the maximum iteration limit (exceed rate) at different maximum iteration numbers. The coding LLM used is DeepSeek-V3.

As shown in Fig. 17 and Tab. 11, the accuracy and appearance score show a rising trend as the maximum iteration number increases, while the exceed rate continuously decreases. When the maximum iteration number is between 14 and 20, the accuracy, appearance score, and exceed rate all begin to converge. This is because most samples finish before reaching the iteration limit, as reflected by the exceed rate, and the impact of the maximum iteration number on performance diminishes.

## J    QUALITATIVE ANALYSIS OF SUPERVISED FINETUNING AND STEP-GRPO

To provide a qualitative analysis of the effects of supervised fine-tuning and Step-GRPO with screenshot and GUI-agent feedback, we present examples of websites generated by Qwen2.5-Coder-7B-Instruct, WebGenAgent-LM-7B-SFT, and WebGenAgent-LM-7B-Step-GRPO in Figs. 18 and 19. We also include examples of websites generated by Qwen3-8B, WebGenAgent-LM-8B-SFT, and WebGenAgent-LM-8B-Step-GRPO in Figs. 20 and 21. As demonstrated in the examples, supervised fine-tuning greatly reduces the models' tendency to generate erroneous or malformed web-

Table 10: Categorical results of WebGen-Agent with various proprietary and open-source models on WebGen-Bench (Lu et al., 2025b), compared with other code agent systems. The highest score of each column is marked in **bold**.

| Test Name | Instruction Categories | | | Test-case Categories | | |
|---|---|---|---|---|---|---|
| | Content Presentation | User Interaction | Data Management | Functional Testing | Data-Display Testing | Design-Validation |
| **OpenHands** | | | | | | |
| Claude-3.5-Sonnet | 32.8 | 18.4 | 18.4 | 12.4 | 33.9 | 32.0 |
| DeepSeek-R1 | 16.4 | 8.9 | 5.9 | 5.0 | 9.9 | 25.0 |
| DeepSeek-V3 | 12.6 | 7.3 | 8.4 | 3.8 | 8.1 | 25.0 |
| **Aider** | | | | | | |
| Claude-3.5-Sonnet | 31.9 | 21.1 | 16.6 | 14.9 | 30.1 | 34.0 |
| DeepSeek-R1 | 39.1 | 28.6 | 13.4 | 17.6 | 35.2 | 44.3 |
| DeepSeek-V3 | 17.8 | 12.8 | 12.5 | 9.7 | 19.1 | 18.4 |
| **Bolt.diy** | | | | | | |
| Claude-3.5-Sonnet | 35.6 | 21.2 | 26.2 | 17.1 | 26.3 | 52.0 |
| DeepSeek-R1 | 43.7 | 20.6 | 24.7 | 21.1 | 29.3 | 44.3 |
| DeepSeek-V3 | 37.1 | 16.6 | 11.2 | 10.5 | 28.2 | 38.1 |
| GPT-4o | 26.4 | 5.9 | 11.2 | 4.7 | 19.6 | 24.6 |
| o3-mini | 28.7 | 17.7 | 13.4 | 11.4 | 25.5 | 33.6 |
| Qwen2.5-Coder-32B | 17.5 | 6.9 | 5.9 | 1.9 | 14.5 | 23.0 |
| Qwen2.5-72B-Inst. | 28.2 | 10.1 | 5.6 | 5.8 | 21.0 | 25.4 |
| WebGen-LM-7B | 27.9 | 23.8 | 38.1 | 22.0 | 27.7 | 47.5 |
| WebGen-LM-14B | 30.2 | 27.8 | 31.6 | 23.6 | 26.9 | 49.2 |
| WebGen-LM-32B | 46.6 | 33.2 | 38.8 | 29.1 | 43.0 | 56.1 |
| **WebGen-Agent** | | | | | | |
| **Proprietary Models** | | | | | | |
| Claude-3.5-Sonnet | 57.8 | 48.7 | 51.9 | 38.5 | 60.5 | 76.2 |
| DeepSeek-R1 | 57.8 | 44.2 | 38.1 | 35.0 | 53.8 | 66.8 |
| DeepSeek-V3 | 58.0 | 53.2 | 45.6 | 40.9 | 61.0 | 72.5 |
| o3 | 59.2 | 46.6 | 53.4 | 43.7 | 55.1 | 68.9 |
| Claude-4-Sonnet | **68.7** | 51.8 | 52.5 | **44.0** | 69.4 | 71.7 |
| Gemini-2.5-Pro | 60.3 | 48.2 | 45.6 | 37.9 | 60.2 | 72.5 |
| Qwen3-Coder-480B-A35B-Inst. | 64.7 | **55.8** | **55.9** | 43.2 | **71.2** | **79.9** |
| **Open-Source Models (10B+)** | | | | | | |
| Qwen2.5-Coder-32B-Inst. | 35.6 | 28.8 | 34.4 | 20.9 | 32.3 | 62.3 |
| Qwen3-Coder-30B-A3B-Inst. | **55.2** | **54.3** | **47.2** | **39.1** | **62.1** | **76.6** |
| Qwen2.5-72B-Instruct | 43.4 | 30.4 | 38.8 | 23.0 | 39.8 | 66.0 |
| WebGen-LM-14B | 44.3 | 31.3 | 44.4 | 28.6 | 40.1 | 61.1 |
| WebGen-LM-32B | 47.1 | 37.2 | 39.1 | 24.6 | 45.7 | 75.8 |
| **Open-Source Models (10B-)** | | | | | | |
| Qwen2.5-Coder-7B-Inst. | 20.7 | 8.6 | 10.9 | 7.4 | 15.9 | 21.3 |
| WebGen-LM-7B | 33.9 | 27.5 | 41.9 | 22.0 | 34.7 | 59.8 |
| WebGenAgent-LM-7B-SFT | **53.4** | 33.5 | 33.8 | 23.5 | 48.4 | 67.6 |
| WebGenAgent-LM-7B-Step-GRPO | 51.1 | **41.1** | **47.8** | **30.7** | **56.7** | **69.3** |
| Qwen3-8B | 37.4 | 34.3 | 30.0 | 26.8 | 34.1 | 54.1 |
| WebGenAgent-LM-8B-SFT | 41.7 | 34.2 | **43.8** | 26.8 | 43.8 | 63.1 |
| WebGenAgent-LM-8B-Step-GRPO | **52.0** | **38.8** | 43.1 | **30.2** | **51.1** | **68.4** |

sites and improves their ability to follow the appearance requirements specified in the instructions. Step-GRPO further refines the aesthetics and harmony of the generated websites.

Table 11: Influence of the maximum number of iterations on WebGen-Agent performance. The coding LLM used is DeepSeek-V3.

| Metric | 2 | 4 | 6 | 8 | 10 | 12 | 14 | 16 | 18 | 20 |
|---|---|---|---|---|---|---|---|---|---|---|
| Accuracy (%) | 42.4 | 47.9 | 50.2 | 50.4 | 52.0 | 51.9 | 51.2 | 53.3 | 52.6 | 52.6 |
| Appearance Score | 3.2 | 3.6 | 3.7 | 3.6 | 3.7 | 3.8 | 3.8 | 3.7 | 3.8 | 3.8 |
| Exceed Rate (%) | 100.0 | 57.4 | 34.7 | 22.8 | 15.8 | 13.9 | 10.9 | 8.9 | 8.9 | 7.9 |

## K  QUALITATIVE ANALYSIS OF THE WEBGEN-AGENT WORKFLOW

To demonstrate how the WebGen-Agent workflow functions, we provide examples of steps in WebGen-Agent trajectories where the agent improves the website's appearance based on screenshot or GUI-agent feedback. As shown in Fig. 22, Fig. 23, Fig. 24, Fig. 25, and Fig. 26, the agent enhances the website's visual appeal by incorporating suggested improvements. Similarly, Fig. 27, Fig. 28, Fig. 29, Fig. 30, and Fig. 31 illustrate how the agent refines the website's functionality based on feedback from the GUI-agent testing process. The steps are simplified due to space constraints.

## L  DATA LEAKAGE EVALUATION

The 700 trajectories used in SFT training are generated from website generation instructions taken from WebGen-Instruct, which is introduced in the WebGen-Bench (Lu et al., 2025b) paper. As explained in Lu et al. (2025b), meticulous decontamination steps have been taken in the construction of WebGen-Instruct. They first employ 5-gram Jaccard similarity, removing the instructions in the training set with a similarity score higher than 0.6 with one of the instructions in the testing set. Then, to remove the instructions that are semantically similar to those in the testing set, they compute the sentence embeddings of the instructions using the all-MiniLM-L6-v2 model of Sentence-Transformer (Reimers & Gurevych, 2019), removing the training instructions with a cosine similarity of larger than 0.55.

To further ensure there is no data leakage, we also apply the n-gram testing, as demonstrated in the table below. The overlap percentages for various n-grams are quite low, and the overlap becomes 0.00% when n-grams are 13.

Table 12: Overlap ratios between the instructions in the training set and the WebGen-Bench test set for various $n$-gram sizes.

| n-grams | 7 | 9 | 11 | 13 |
|---|---|---|---|---|
| Overlap Ratio (%) | 5.59 | 1.47 | 0.22 | 0.00 |

## M  GENERALIZABILITY OF WEBGEN-AGENT

To demonstrate the generalizability of our method, we test WebGen-Agent on the Web Application subset of the ArtifactsBench (Zhang et al., 2025a) dataset, which also requires generating a website based on user instructions, though their requirements are less complicated than the instructions in WebGen-Bench. The baseline results of the closed-source models and DeepSeek-V3 are taken from Zhang et al. (2025a), while the others we tested using their open-sourced codebase. As shown in the Tab. 13, WebGen-Agent with Qwen3-Coder-480B-A35B-Instruct as the coding LLM achieves the best performance of 63.86, outperforming the baseline results of various closed-source models and the baseline result of Qwen3-Coder-480B-A35B-Instruct. WebGen-Agent with DeepSeek-V3 also outperforms the baseline result of DeepSeek-V3. This demonstrates the effectiveness and generalizability of WebGen-Agent. We have added this to the revised paper.

Table 13: Performance of WebGen-Agent on the Web Application subset of the Artifacts-Bench (Zhang et al., 2025a) dataset. The baseline results of the closed-source models and DeepSeek-V3 are taken from Zhang et al. (2025a), while the others are tested using their open-sourced codebase.

| Test Name | Average MLLM score |
|---|---|
| GPT-4o | 39.27 |
| o1 | 39.01 |
| Claude 3.7-Sonnet | 53.64 |
| Claude 4.0-Sonnet | 55.79 |
| Gemini-2.5-Pro | 58.30 |
| DeepSeek-V3 | 47.47 |
| **DeepSeek-V3 + WebGen-Agent** | **55.74** |
| Qwen3-Coder-480B-A35B-Instruct | 55.42 |
| **Qwen3-Coder-480B-A35B-Instruct + WebGen-Agent** | **63.86** |

Table 14: Runtime, cost, compute, and qualitative scores for each model/agent configuration. Numbers in parentheses denote the gain of WebGen-Agent over the corresponding Bolt.diy baseline.

| Test Name | Avg. Run-time (min) | Avg. Cost ($) | Avg. FLOPs (TFLOPs) | Accuracy | Appearance Score |
|---|---|---|---|---|---|
| Claude-3.5-Sonnet + Bolt.diy | 2.1 | 0.0677 | N/A | 26.4 | 3.0 |
| Claude-3.5-Sonnet + WebGen-Agent | 13.1 | 0.2009 | N/A | 51.9$_{(+25.5)}$ | 3.9$_{(+0.9)}$ |
| DeepSeek-R1 + Bolt.diy | 6.3 | 0.0025 | $1.55 \times 10^{16}$ | 27.8 | 2.5 |
| DeepSeek-R1 + WebGen-Agent | 17.5 | 0.0057 | $3.54 \times 10^{16}$ | 46.4$_{(+18.6)}$ | 3.8$_{(+1.3)}$ |
| DeepSeek-V3 + Bolt.diy | 3.2 | 0.0088 | $3.18 \times 10^{16}$ | 20.8 | 2.0 |
| DeepSeek-V3 + WebGen-Agent | 13.3 | 0.0131 | $2.33 \times 10^{16}$ | 52.6$_{(+31.8)}$ | 3.8$_{(+1.8)}$ |
| Qwen2.5-Coder-32B-Inst. + Bolt.diy | 1.4 | N/A | $8.18 \times 10^{14}$ | 9.5 | 1.1 |
| Qwen2.5-Coder-32B-Inst. + WebGen-Agent | 16.7 | N/A | $2.47 \times 10^{15}$ | 32.0$_{(+22.5)}$ | 3.3$_{(+2.2)}$ |
| Qwen2.5-72B-Inst. + Bolt.diy | 2.9 | N/A | $1.90 \times 10^{15}$ | 13.8 | 1.4 |
| Qwen2.5-72B-Inst. + WebGen-Agent | 19.0 | N/A | $6.54 \times 10^{15}$ | 35.9$_{(+22.1)}$ | 3.4$_{(+2.0)}$ |

## N    EFFICIENCY ANALYSIS

We present the average runtime, average cost, and average FLOPs in the table below, as well as the accuracy and appearance scores. As demonstrated in Tab. 14, the average runtimes of the WebGen-Agent pipeline are all between 10 to 20 minutes, and the average costs are all between $0.01 and $0.3, which is within the acceptable range for developing a multi-file codebase of an interactive website from scratch. The time used for Claude-3.5-Sonnet and DeepSeek-V3 is around 13 minutes, as they have strong coding abilities and make fewer mistakes, thus requiring fewer iterations. DeepSeek-R1's average runtime is longer, as it is a reasoning model and needs to generate longer thoughts. Qwen2.5-Coder-32B-Instruct and Qwen2.5-72B-Instruct are weaker models and need more iterations to correct errors and improve the websites.

The average runtimes and average costs of WebGen-Agent are notably higher than those of Bolt.diy, but WebGen-Agent significantly improves the accuracy and appearance scores compared to Bolt.diy, bringing around 20% to 30% rise in accuracy and around a 1.0 to 2.0 rise in appearance scores. Therefore, we believe that this significant increase in performance justifies the increased cost and runtime. We have added the results to Appendix N in the revised paper.

## O    RESULTS OF DIRECT RL TRAINING WITHOUT SFT

We have finished running the experiment of directly training the original model (without special SFT) using our Step-GRPO with Screenshot and GUI-agent Feedback training approach. We per-

Table 15: Result of training directly with RL, without SFT warm-up, on Qwen3-8B. "Accuracy" is a weighted metric where **Yes** answers count 1.0 and **Partial** answers count 0.5, divided by the number of test cases.

| Test Name | Yes | Partial | No | Start Failed | Accuracy | Appearance Score |
|---|---|---|---|---|---|---|
| Qwen3-8B | 29.5 | 9.1 | 61.4 | 0.0 | 34.1 | 3.2 |
| Qwen3-8B + Step-GRPO | 34.2 | 12.7 | 52.4 | 0.8 | 40.5 (+6.4) | 3.5 (+0.3) |
| Qwen3-8B + SFT | 32.8 | 11.6 | 55.6 | 0.0 | 38.6 | 3.4 |
| Qwen3-8B + SFT + Step-GRPO | 37.4 | 12.1 | 50.5 | 0.0 | 43.4 (+4.8) | 3.6 (+0.2) |

formed RL training directly on Qwen3-8B, and the results are presented below. As shown in the table below, applying Step-GRPO with Screenshot and GUI-agent Feedback directly on Qwen3-8B achieves a rise in accuracy of 6.5% and a rise in appearance score of 0.3. This improvement in performance is even more pronounced than training on the SFT model. This demonstrates that **the distillation of DeepSeek-V3 is not absolutely necessary**, and that the same gains could likely be obtained when applied to a larger model.

## P   THE USE OF LARGE LANGUAGE MODELS IN PAPER WRITING

The paper is primarily human-written. However, large language models such as o3 (OpenAI, 2025a) and DeepSeek-V3 (Liu et al., 2024) are used to check for grammar and spelling mistakes. The words and phrases are occasionally polished by LLMs to make the wording more fluent.

**System Prompt:** You are Bolt, an expert AI assistant and exceptional senior software developer with vast knowledge across multiple programming languages, frameworks, and best practices. <system_constraints> You are operating in an environment called WebContainer ......

**User:** Please implement a food distribution website for managing and distributing food. The website should have functionalities for food donation, food distribution, and volunteer management. Users should be able to browse food donation information, apply for food distribution, view volunteer information, and manage their own accounts. Use azure for the background and midnight blue for components.

**Assistant:** I'll create a food distribution website with the requested functionalities and color scheme. Here's the implementation plan: 1. Set up Vite + React with required dependencies 2. Create components for donation, distribution, and volunteer management ......

Installation of dependencies was successful. Starting the service was successful. **The screenshot description:** The webpage has a clean and simple layout with a dark blue header containing navigation links: 'Home', 'Donations', 'Distribution', 'Volunteers', and 'Login' ...... **Suggestions for Improvement:** Consider adding more visual elements such as icons ...... If no further modification is necessary, output <boltAction type="screenshot_validated"/> to signal that the screenshot is satisfactory.

$Score_{shot} = 3, Score_{gui} = 0$

**Assistant:** I'll enhance the design with more visual elements while maintaining the color scheme. Here are the key improvements: 1. Add food-related icons to 2. Add cards for featured ......

Installation of dependencies was successful. Starting the service was successful. **The screenshot description:** The screenshot shows a clean and organized homepage for a website called 'Food Distribution Network.' The overall layout includes a top navigation bar with icons and links labeled 'Home,' 'Donations,' 'Distribution,' 'Volunteers,' and 'Login.' Below the navigation bar, there is ...... If no further modification is necessary, output <boltAction type="screenshot_validated"/> to signal that the screenshot is satisfactory.

<boltAction type="screenshot_validated"/> ✅

**The suggestions based on the GUI-agent testing result:** Address the root cause of the blank page issue, which could involve server-side errors, frontend rendering problems, or resource loading failures. Implement robust error handling and loading indicators to provide feedback to users. Ensure the website is thoroughly tested in staging environments to catch such critical issues before deployment ......

$Score_{shot} = 5, Score_{gui} = 1$

**Assistant:** I'll add error handling and loading states to prevent blank page issues. Here are the key improvements: 1. Add loading states for all pages ......

Installation of dependencies was successful. Starting the service was successful. **The screenshot description:** The screenshot shows a clean and organized homepage for a food distribution network. The overall layout includes a top navigation bar with icons and links labeled 'Home', 'Donations', 'Distribution', 'Volunteers', and 'Login'. Below the navigation bar, there is a prominent heading 'Welcome to...... If no further modification is necessary, output <boltAction type="screenshot_validated"/> to signal that the screenshot is satisfactory.

<boltAction type="screenshot_validated"/> ✅

**The suggestions based on the GUI-agent testing result:** Improve the Food Distribution Application form by: 1) Clearly marking all mandatory fields with an asterisk or placeholder text before submission, 2) Providing a summary of required fields upfront to prevent partial submissions, and 3) Implementing client-side validation to highlight missing fields simultaneously rather than sequentially.

$Score_{shot} = 5, Score_{gui} = 3$

**Assistant:** I'll enhance the distribution form with better validation and field requirements. Here are the improvements: 1. Add asterisks to required fields 2. Implement client-side validation 3. Show all validation errors at once 4. Add a summary of ......

......

......
(More Rounds Omitted)

Figure 8: Example of a WebGen-Agent trajectory.

**System Prompt:** You are Bolt, an expert AI assistant and exceptional senior software developer with vast knowledge across multiple programming languages, frameworks, and best practices. <system_constraints> You are operating in an environment called WebContainer ……

**User:** Please implement a multi-company dashboard for managing and displaying financial data from multiple companies. The dashboard should be able to collect and display financial information from each company, provide consolidated reports, and support cross-company comparisons and reporting. Users should be able to browse financial data from each company, view consolidated reports, and perform financial management and reporting. Apply mint cream as the background; style all components with teal.

**Assistant:** I'll create a multi-company financial dashboard using React with Vite. Here's the implementation plan: 1. Set up Vite + React project 2. Create components for company data display and comparison 3. Implement mock financial data storage 4. Style with mint cream background and teal components……

Installation of dependencies was successful. Starting the service was successful. **The screenshot description:** The screenshot shows a financial dashboard with a clean and organized layout. The top section features a teal header with the title 'Financial Dashboard' and navigation options labeled 'Dashboard' and 'Consolidated Report'. Below the header is a search bar for filtering companies. The main content area is divided into four equal-sized cards …… If no further modification is necessary, output <boltAction type="screenshot_validated"/> to signal that the screenshot is satisfactory.

<boltAction type="screenshot_validated"/> ✅

Click [1]      Click [2]      Scroll WINDOW; down      ……

**The suggestions based on the GUI-agent testing result:** Improve navigation and accessibility of cross-company comparison and financial management functionalities to allow thorough testing. Ensure all critical features are easily reachable and not hidden behind repetitive interactions. Consider adding more interactive elements or drill-down options for detailed financial data exploration ……

$Score_{shot} = 4, Score_{gui} = 2$

**Assistant:** I'll enhance the dashboard with better comparison features and financial management tools. Here are the key improvements: 1. Add comparison mode to select multiple companies 2. Include financial ratios and metrics 3. Add export functionality for reports……

Installation of dependencies was successful. Starting the service was successful. **The screenshot description:** The screenshot shows a financial dashboard with a clean and organized layout. The top section includes a header with the title 'Financial Dashboard' and navigation options labeled 'Dashboard' and 'Consolidated Report'. Below the header is a search bar labeled 'Search companies…'. A button labeled 'Compare Companies' is …… If no further modification is necessary, output <boltAction type="screenshot_validated"/> to signal that the screenshot is satisfactory.

<boltAction type="screenshot_validated"/> ✅

Click [1]      Click [2]      Scroll WINDOW; down      ……

The trajectory shows that the GUI agent successfully verified the mint cream background and teal components, browsed financial data, viewed consolidated reports, and performed cross-company comparisons. The GUI agent testing is successful and no further improvement is necessary. Observe the above feedback and decide whether further modifications to the code are needed based on the GUI-agent testing summary. If no further modification is necessary, output <boltAction type="finish"/>

$Score_{shot} = 4, Score_{gui} = 5$

<boltAction type="finish"/> ✅

Figure 9: Example of a WebGen-Agent trajectory.

**System Prompt:** You are Bolt, an expert AI assistant and exceptional senior software developer with vast knowledge across multiple programming languages, frameworks, and best practices. <system_constraints> You are operating in an environment called WebContainer ......

**User:** Please implement a dashboard website for displaying regulatory policies. The website should have data visualization capabilities to clearly display complex regulatory policies. Users should be able to browse and analyze different policies, view related data and charts, and be able to filter and sort as needed. Assign lavender to the background and indigo to the UI elements.

**Assistant:** I'll create a regulatory policy dashboard with data visualization capabilities using React and Chart.js. Let me think through this step by step: 1. We need a dashboard layout with: - Header with title and navigation ......

Installation of dependencies was successful. Starting the service was successful. **The screenshot contains errors:** This localhost page can't be found. Modify the code to fix the errors in screenshot.

$Score_{shot} = 0, Score_{gui} = 0$ ⊗

**Assistant:** ......

**Feedback:** ......
$Score_{shot} = 0, Score_{gui} = 0$ ⊗

......

Five Consecutive Errors, Backtrack!

**Assistant:** I'll create a regulatory policy dashboard with data visualization capabilities using React and Chart.js. Let me think through this step by step: 1. We need a dashboard layout with navigation and main content area ......

Installation of dependencies was successful. **Starting the service emitted errors:** > regulatory-policy-dashboard@0.0.0 dev > vite   VITE v4.5.14  ready in 3504 ms Local: http://localhost:36593/ Network: use --host to expose Error:  Failed to scan for dependencies from entries: /workspace/index.html ERROR Expected closing "strong" tag to match opening "p" tag   src/pages/PolicyDetailPage.jsx:33:61:...   <p>Effective Date: {policy.date}</p>  The opening "p" tag is here: /PolicyDetailPage.jsx:33:11: Effective Date: ......
$Score_{shot} = 0, Score_{gui} = 0$ ⊗

**Assistant:** I'll fix the JSX syntax errors in the PolicyDetailPage.jsx file. The issue is with mismatched HTML tags in the JSX structure. <boltArtifact id="regulatory-dashboard-fixed" title="Regulatory Policy Dashboard - Fixed"> ......

Installation of dependencies was successful. Starting the service was successful. **The screenshot description:** The screenshot shows a Regulatory Policy Dashboard with a clean and structured layout. The overall design features a purple and white color scheme, giving it a professional and modern look. The dashboard includes a sidebar on the left with navigation options such as Dashboard, Policies, Analytics, Reports, Settings, Filters, Categories, Status, ...... If no further modification is necessary, output <boltAction type="screenshot_validated"/> to signal that the screenshot is satisfactory.
$Score_{shot} = 4, Score_{gui} = 0$

**Assistant:** Looking at the screenshot description, I can see that the dashboard is functioning but could benefit from some improvements to better meet the requirements. The current implementation only shows a basic dashboard view,......

Installation of dependencies was successful. Starting the service was successful. **The screenshot description:** The screenshot shows a Regulatory Policy Dashboard with a clean and organized layout. The overall design features a purple and white color scheme, giving it a professional and modern look. The dashboard includes a sidebar on the left with navigation options such as Dashboard, Policies, ...... If no further modification is necessary, output <boltAction type="screenshot_validated"/> to signal that the screenshot is satisfactory.

<boltAction type="screenshot_validated"/> ✓

Click [1]          Scroll WINDOW; down          ANSWER; YES

Looking at this GUI-agent testing trajectory, I can see that the agent successfully navigated through the application, verified the requested visual elements (lavender background and indigo UI elements), and performed the required actions of browsing policies, using filters, and analyzing data. The GUI agent testing is successful and no further improvement is necessary. If no further modification is necessary, output <boltAction type="finish"/>.
$Score_{shot} = 5, Score_{gui} = 5$

<boltAction type="finish"/>

Figure 10: Example of a WebGen-Agent trajectory.

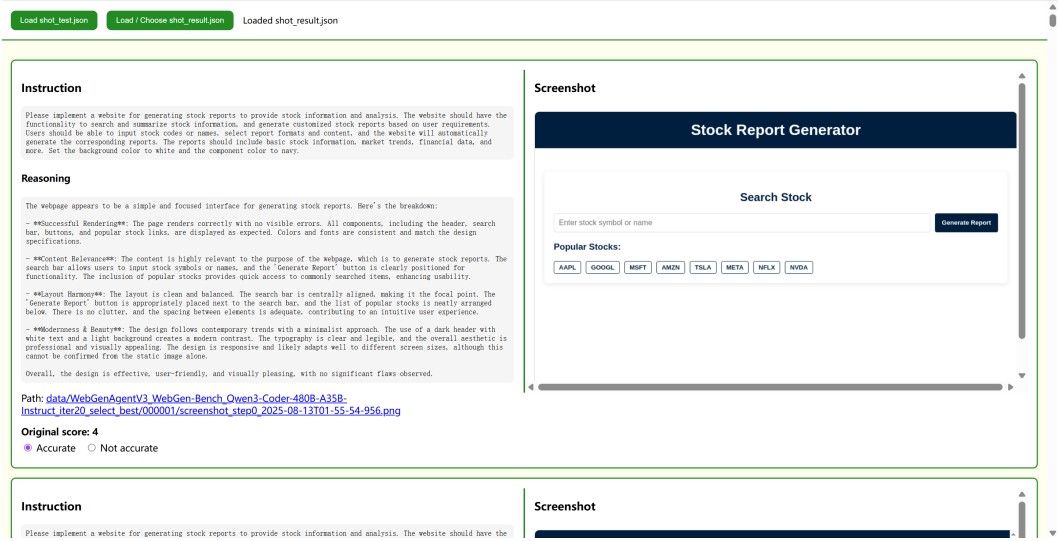

Figure 11: Manual testing interface of the accuracy of the screenshot scores.

**Screenshot Score Accuracy Evaluation Guidelines:**
Check the VLM-generated screenshot scores at each step in the WebGen-Agent trajectories. If the score match the appearance of the screenshot, select "Accurate". If the score does not match the appearance of the screenshot, select "Not accurate", and a number input box would appear next to the "Not accurate" choice. You should enter the accurate score in it.

**Grading Scale:**

- **0 (Blank Page):** The screenshot is completely blank or does not contain any visible content. It may only have a background color or display an error message.

- **1 (Poor):** Major rendering issues (e.g., broken layouts, incorrect colors). Content is irrelevant or missing. Layout is chaotic. Design is outdated or visually unappealing.

- **2 (Below Average):** Partial rendering with noticeable errors. Content is partially relevant but poorly organized. Layout lacks consistency. Design is basic or uninspired.

- **3 (Average):** Mostly rendered correctly with minor flaws. Content is relevant but lacks polish. Layout is functional but unremarkable. Design is clean but lacks modern flair.

- **4 (Good):** Rendered well with no major errors. Content is relevant and logically organized. Layout is harmonious and user-friendly. Design is modern and visually appealing.

- **5 (Excellent):** Flawless rendering. Content is highly relevant, intuitive, and tailored to user needs. Layout is polished, responsive, and innovative. Design is cutting-edge, beautiful, and memorable.

Figure 12: Grading guidelines for manual evaluation of the accuracy of the screenshot scores.

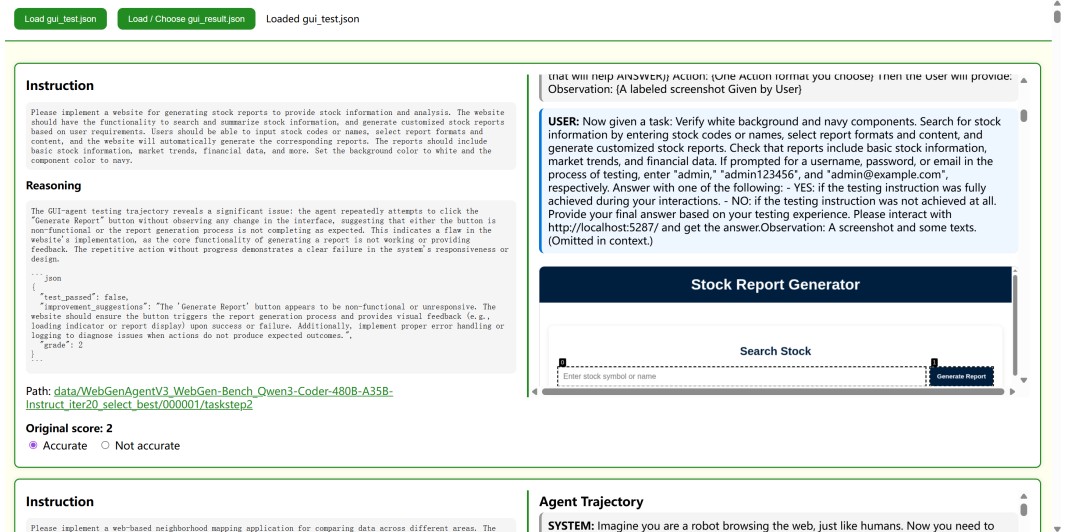

Figure 13: Manual testing interface of the accuracy of the GUI-agent scores.

**GUI-agent Score Accuracy Evaluation Guidelines:**
Check the VLM-generated GUI-agent scores at each step in the WebGen-Agent trajectories. The GUI agent trajectories are to the right, and you can scroll to inspect them. If the score match the GUI agent trajectory, select "Accurate". If the score does not match the GUI agent trajectory, select "Not accurate", and a number input box would appear next to the "Not accurate" choice. You should enter the accurate score in it.

**Grading Scale:**

- **1:** The vast majority of tested functions fail or behave incorrectly.
- **2:** Many functions fail; only a few behave as expected.
- **3:** About half of the functions work as expected; success is mixed.
- **4:** Most functions work as expected; only minor issues remain.
- **5:** All tested functions work exactly as expected; no issues observed.

Figure 14: Grading guidelines for manual evaluation of the accuracy of the GUI-agent scores.

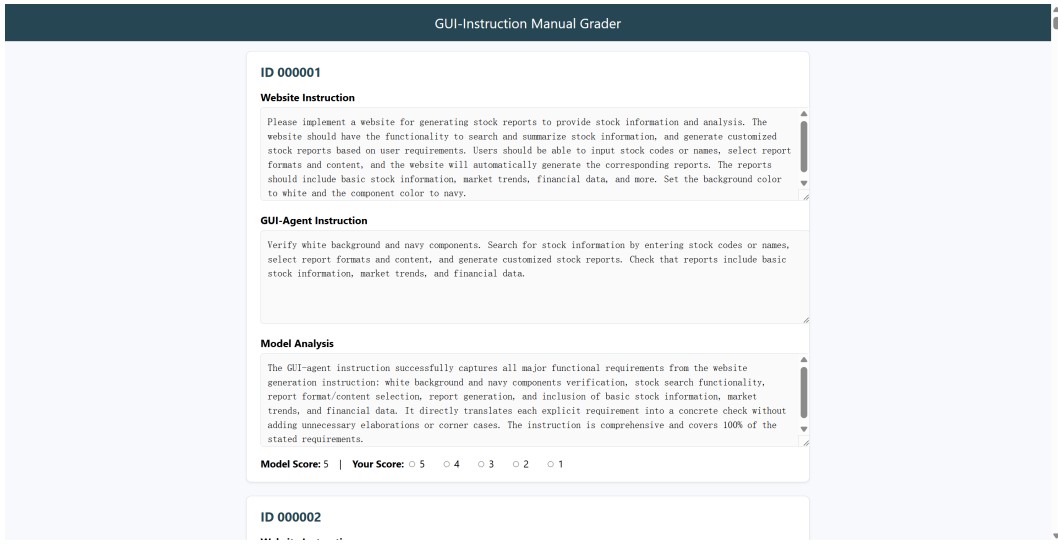

Figure 15: Manual testing interface of the comprehensiveness of the GUI-agent testing instructions.

---

**GUI-agent Instruction Evaluation Guidelines:**
Score the instruction 1 – 5, where 5 = best. The dominant criterion is comprehensiveness: how completely the instruction translates every website requirement into concrete GUI-agent checks.

**Grading Scale:**

- **1 (Minimal, < 25 %):** The instruction overlooks most of the stated requirements.
- **2 (Low, 25 – 50 %):** Only some primary requirements are mentioned; many important items are absent.
- **3 (Moderate, 50 – 75 %):** Core functionalities are covered, but several secondary features or style rules are skipped.
- **4 (High, 75 – 90 %):** All major functional requirements plus most visual or secondary ones are included; only a few minor details are missing.
- **5 (Complete, ≈ 100 % of requirements):** Every requirement is turned into checks. Nothing significant is left out.

Figure 16: Grading guidelines for manually evaluating GUI-agent testing instructions

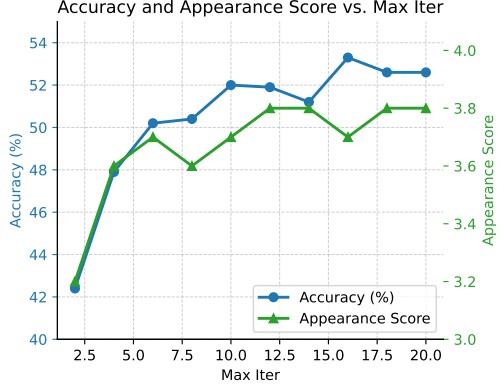 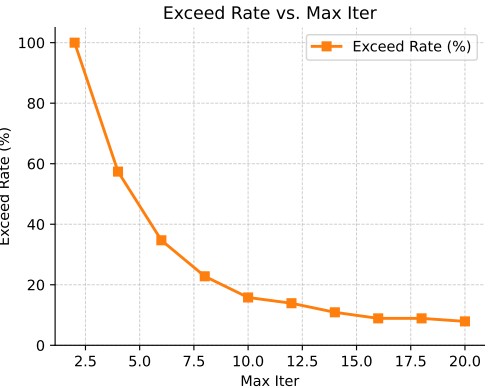

(a) Accuracy (%) and Appearance Score as a function of the maximum number of iterations.

(b) Exceed Rate (%) versus the maximum number of iterations.

Figure 17: Effect of the maximum iteration number hyper-parameter on different performance metrics. The coding LLM used is DeepSeek-V3.

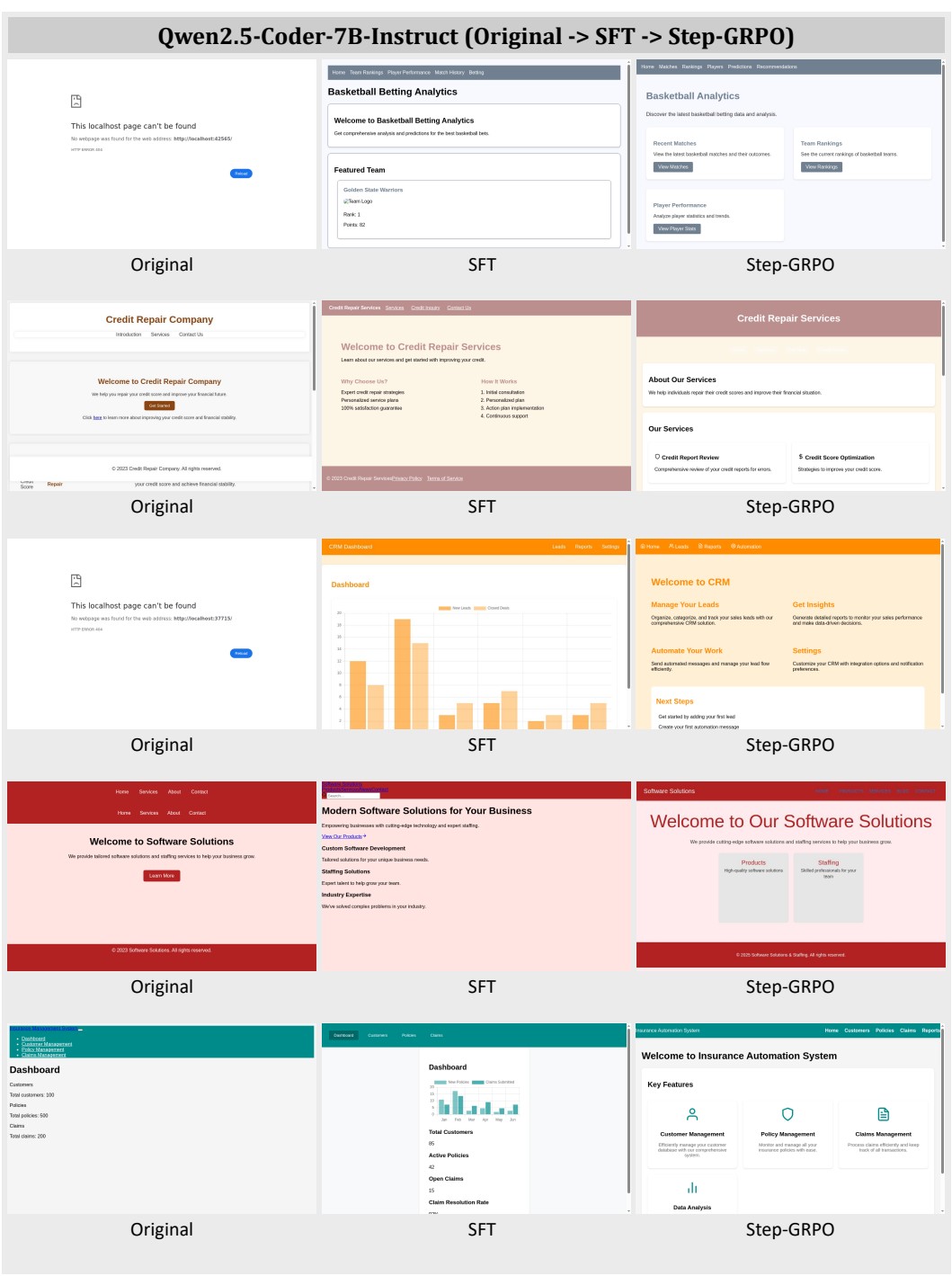

Figure 18: Screenshots of websites created by Qwen2.5-Coder-7B-Instruct, WebGenAgent-LM-7B-SFT, and WebGenAgent-LM-7B-Step-GRPO.

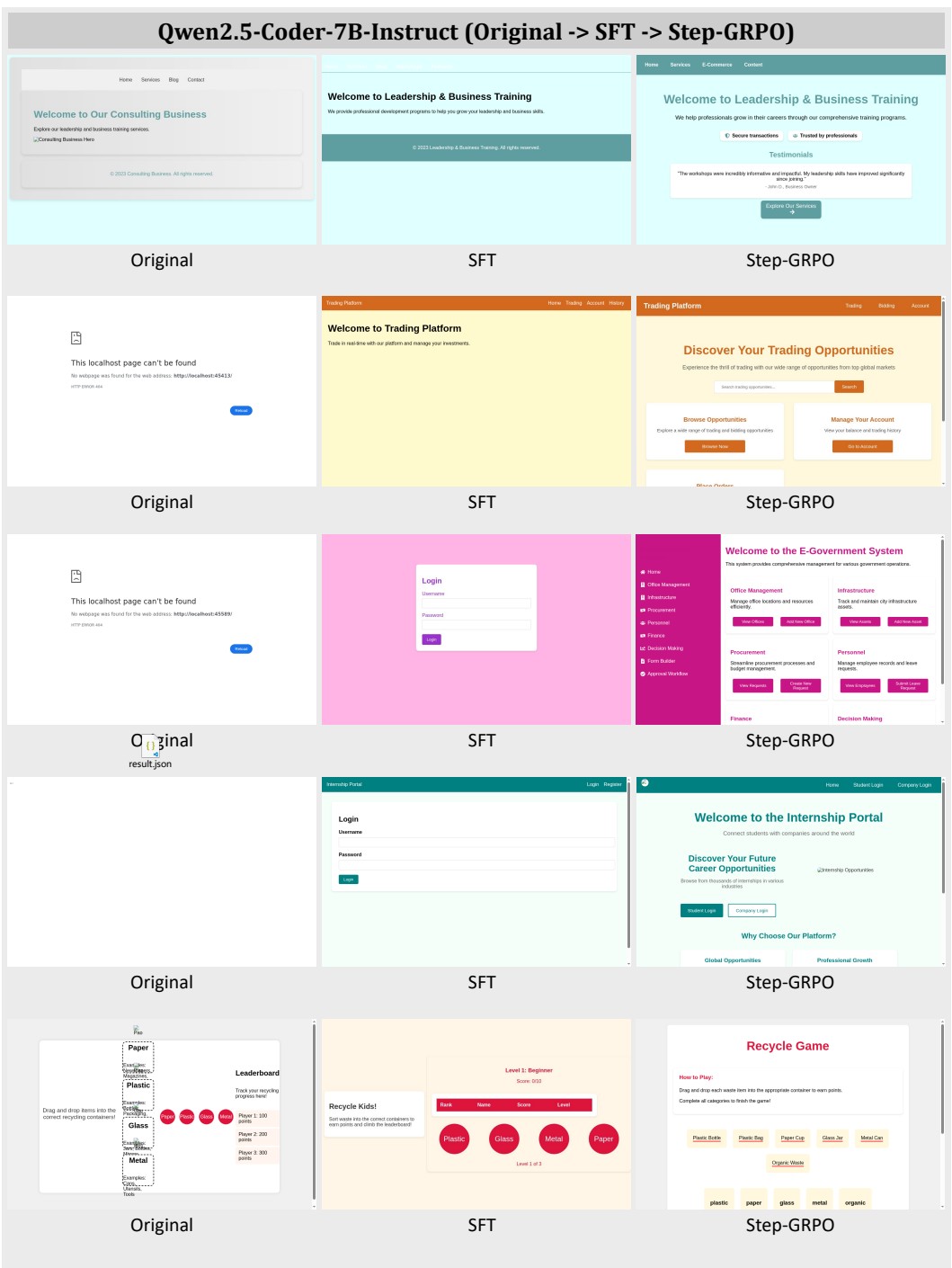

Figure 19: Screenshots of websites created by Qwen2.5-Coder-7B-Instruct, WebGenAgent-LM-7B-SFT, and WebGenAgent-LM-7B-Step-GRPO.

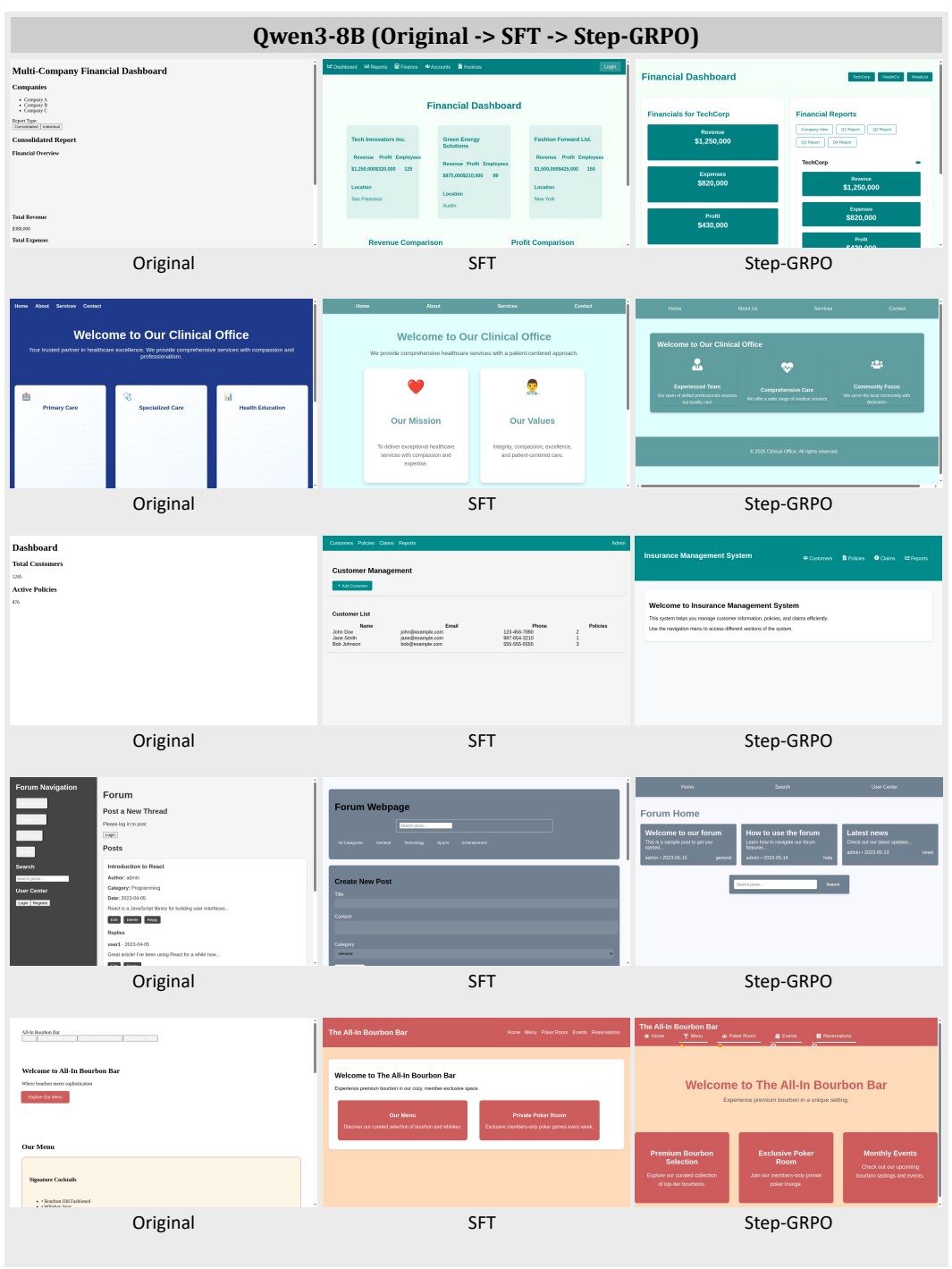

Figure 20: Screenshots of websites created by Qwen3-8B, WebGenAgent-LM-8B-SFT, and WebGenAgent-LM-8B-Step-GRPO.

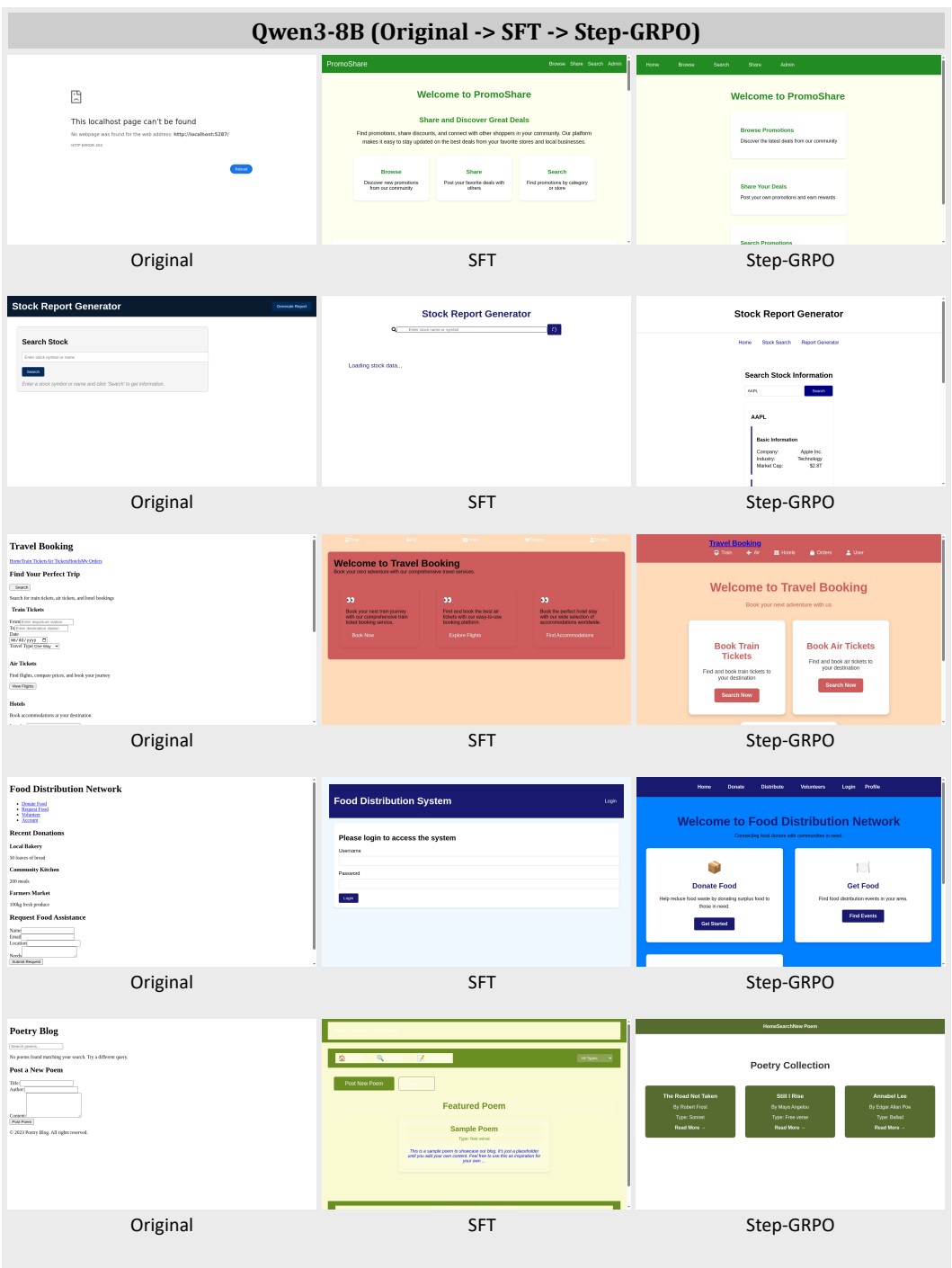

Figure 21: Screenshots of websites created by Qwen3-8B, WebGenAgent-LM-8B-SFT, and WebGenAgent-LM-8B-Step-GRPO.

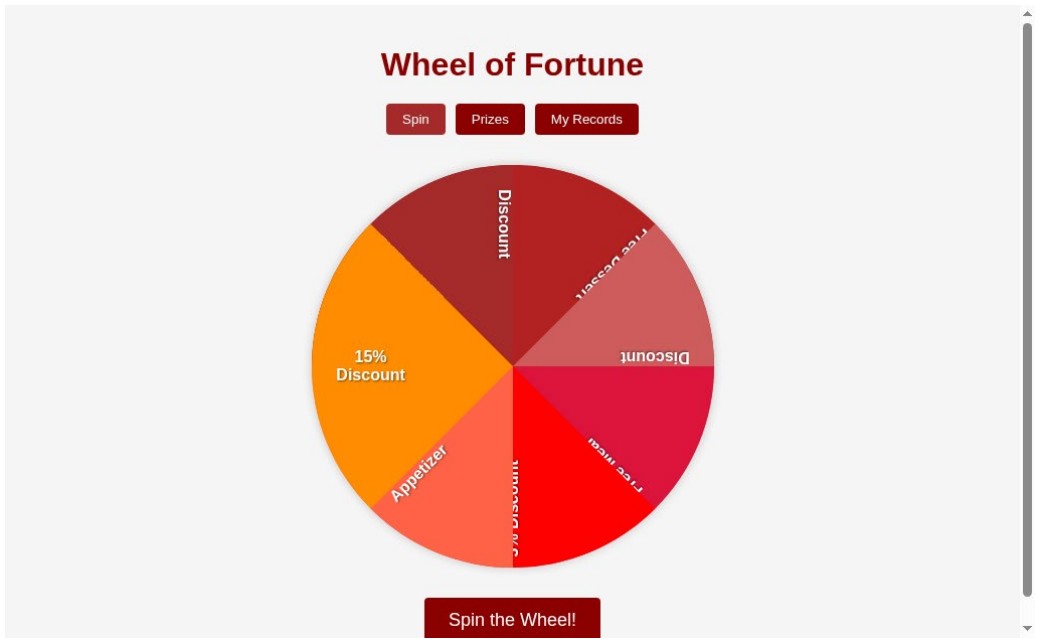

Suggestions: 1. Consider adding hover effects to the buttons to provide visual feedback when users interact with them. 2. Ensure consistent spacing around the 'Spin the Wheel!' button to maintain alignment with the top navigation buttons......

I'll enhance the UI based on the feedback while maintaining all the required functionality. Here are the improvements: <boltArtifact id="wheel-of-fortune-enhanced" title="Enhanced Wheel of Fortune Website">\n <boltAction type="file" filePath="src/App.css">.app {\n max-width: 800px;\n margin: 0 auto;\n padding: 20px;\n text-align: center;\n ......

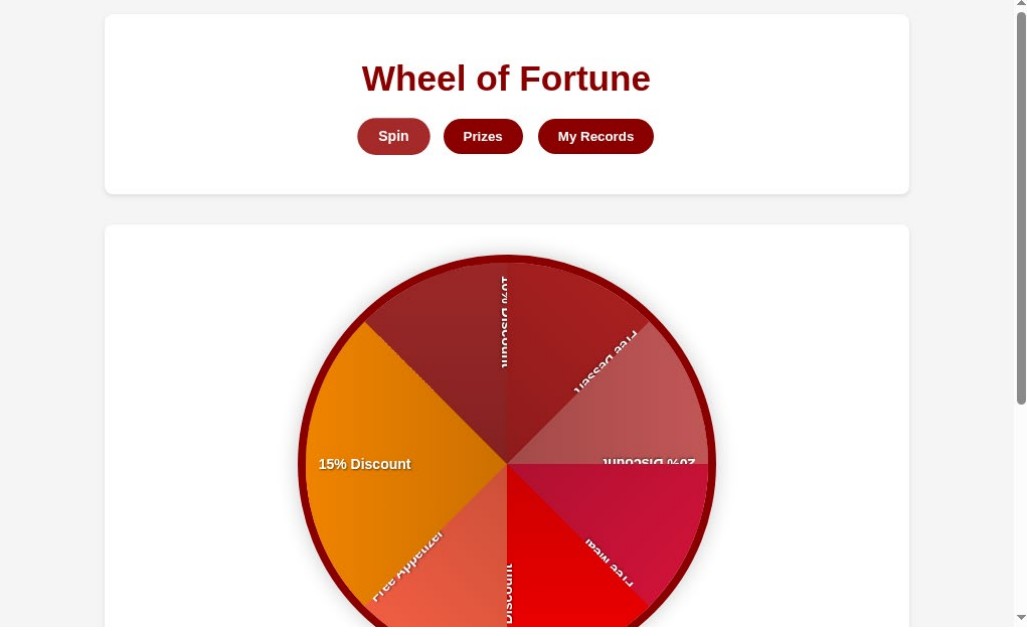

Figure 22: Example of a step in a WebGen-Agent trajectory where the agent improves the website's appearance based on screenshot feedback. The step is simplified due to space constraints.

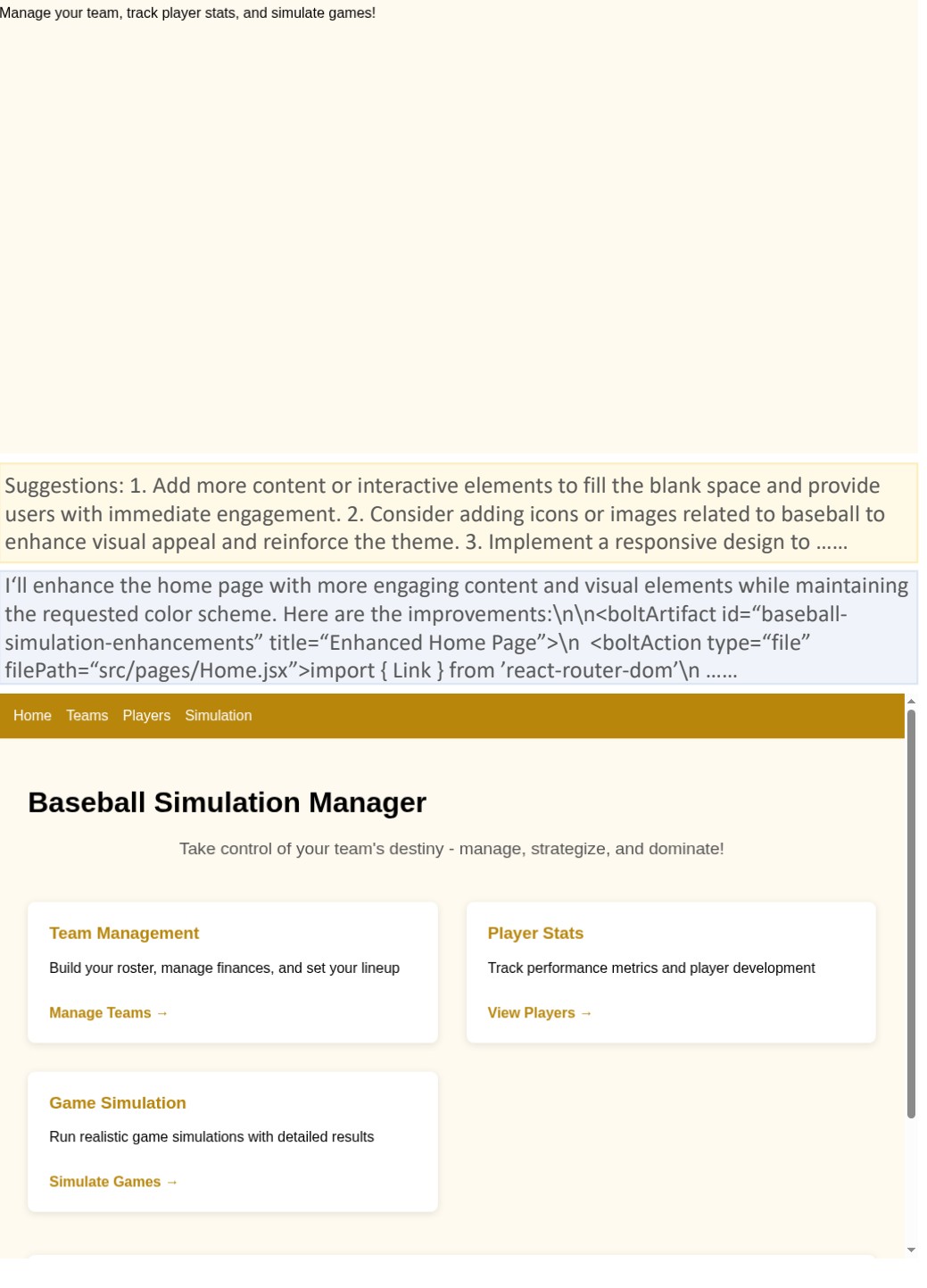

Figure 23: Example of a step in a WebGen-Agent trajectory where the agent improves the website's appearance based on screenshot feedback. The step is simplified due to space constraints.

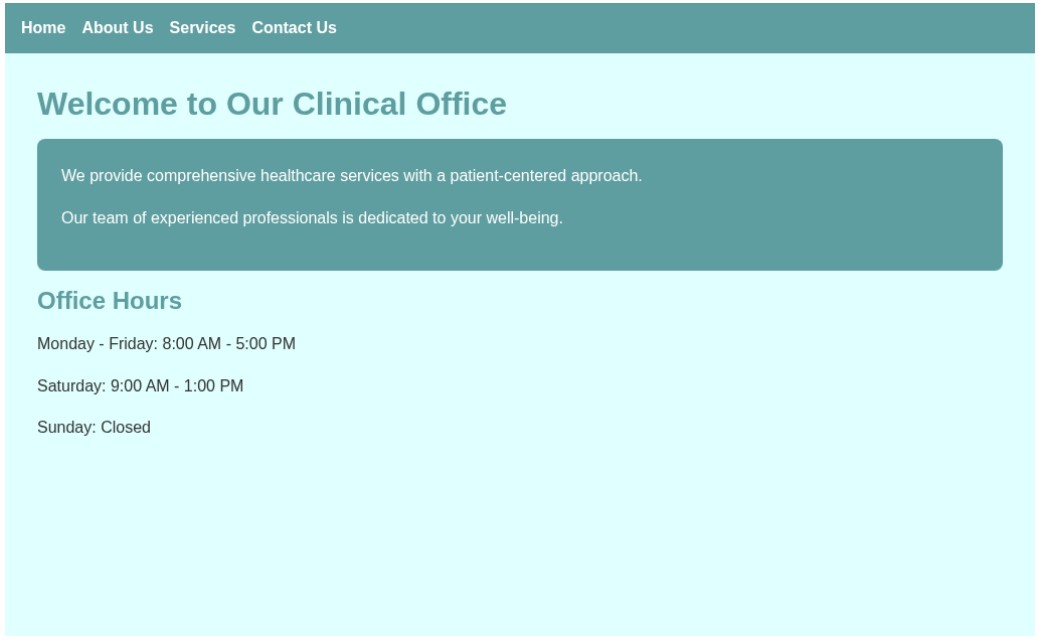

Suggestions: Consider adding some visual elements such as icons or images related to healthcare to enhance engagement. Additionally, introducing a footer with contact information or social media links could improve user experience.......

I'll enhance the website by adding a footer with contact info and some healthcare icons to improve visual appeal. Here are the modifications:\n\n<boltArtifact id=\"clinical-office-website-enhanced\" title=\"Enhanced Clinical Office Website\">\n  <boltAction type=\"file\" filePath=\"src/index.css\">* {\n  margin: 0;\n\n......

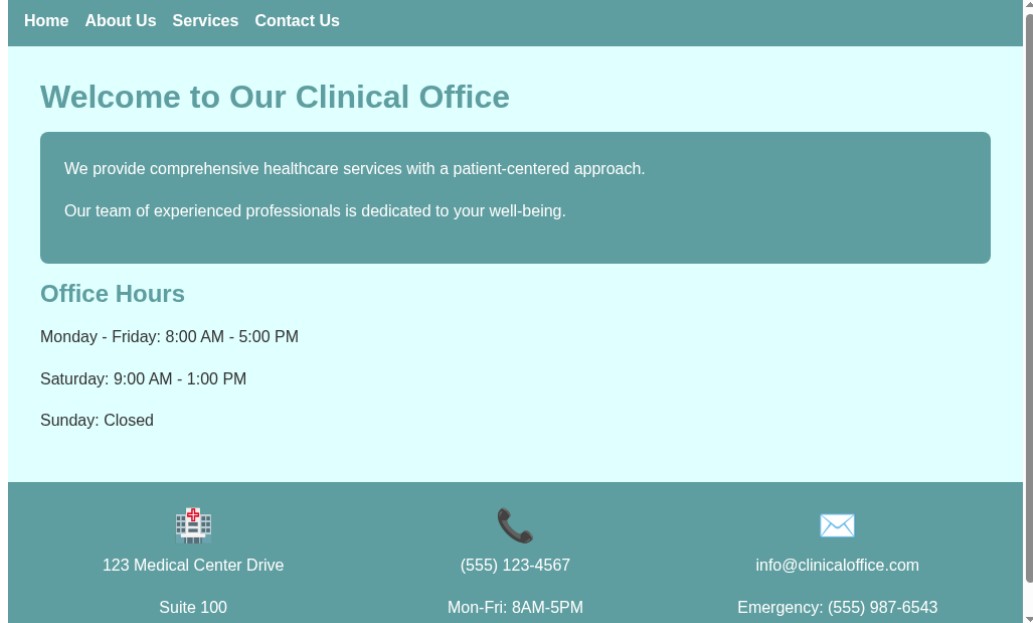

Figure 24: Example of a step in a WebGen-Agent trajectory where the agent improves the website's appearance based on screenshot feedback. The step is simplified due to space constraints.

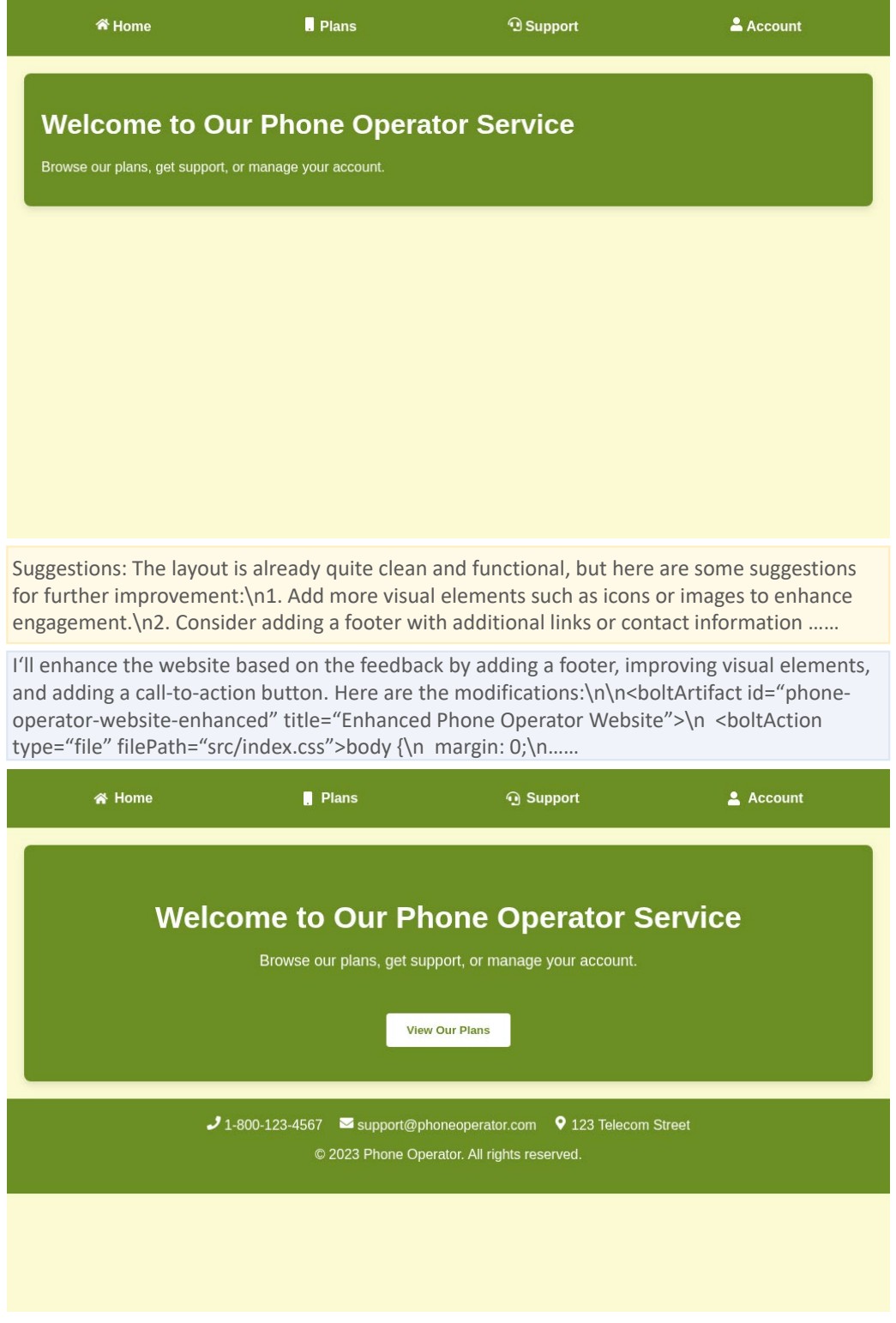

Figure 25: Example of a step in a WebGen-Agent trajectory where the agent improves the website's appearance based on screenshot feedback. The step is simplified due to space constraints.

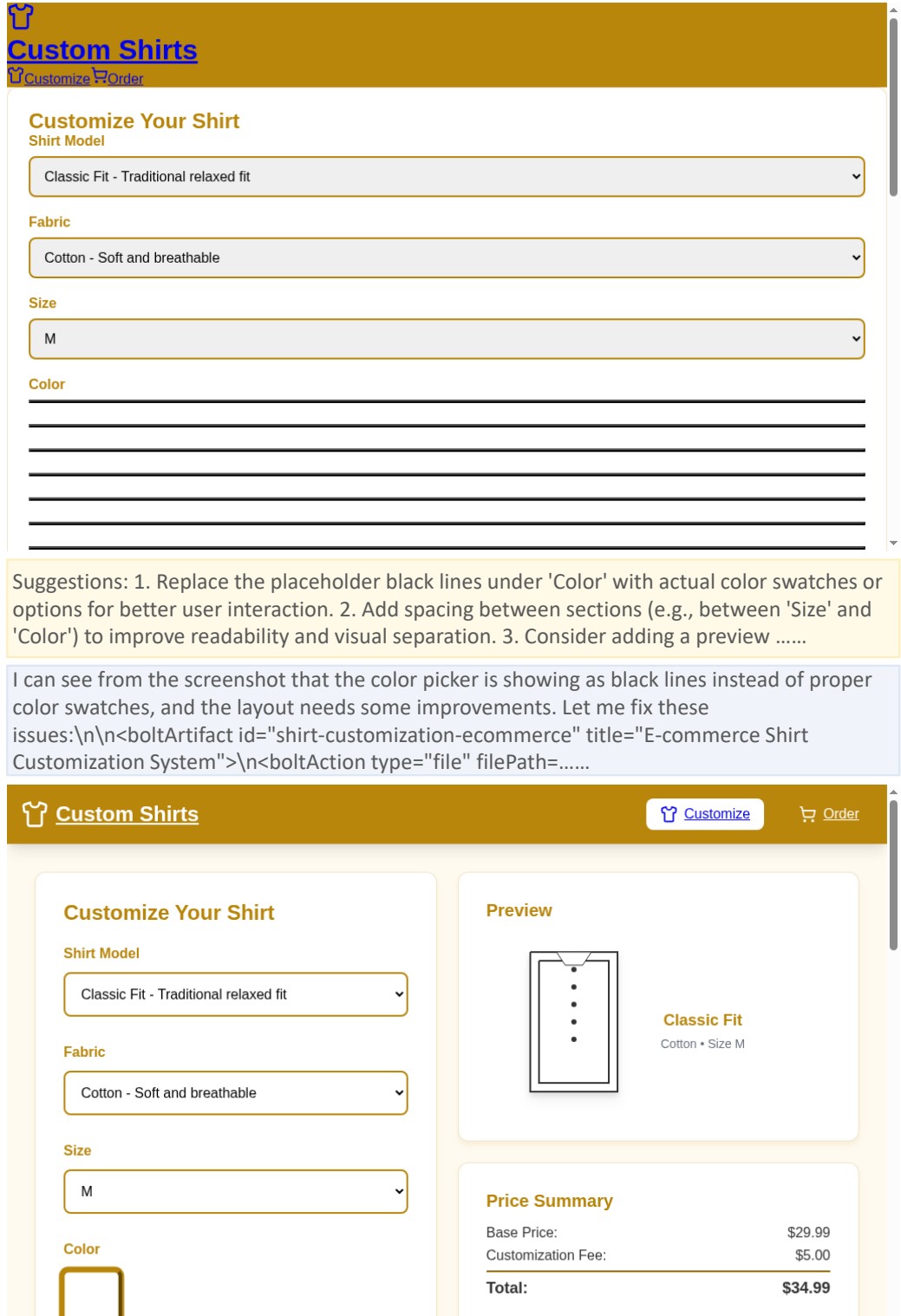

Figure 26: Example of a step in a WebGen-Agent trajectory where the agent improves the website's appearance based on screenshot feedback. The step is simplified due to space constraints.

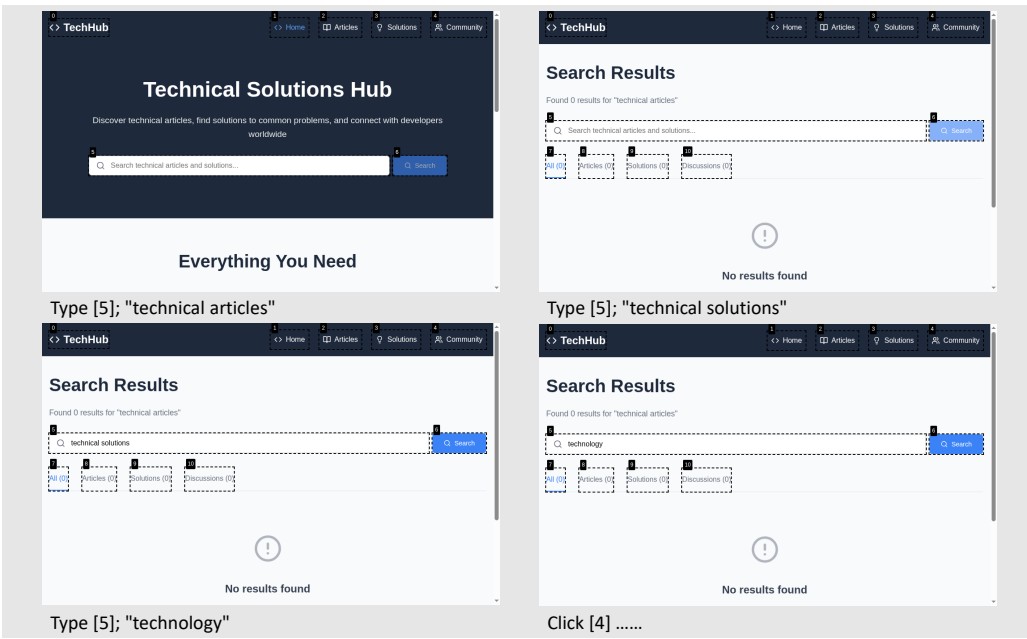

Suggestions: 1. Fix the search functionality to properly index and return relevant technical content, or populate the database with searchable articles and solutions. 2. Implement proper error messaging for failed searches and form submissions to provide user feedback ......

I need to address the specific issues mentioned in the feedback. Let me fix the search functionality, form submission issues, and add proper error messaging.\n\n<boltArtifact id="technical-website" title="Technical Information and Solutions Website">\n<boltAction type="file" ......

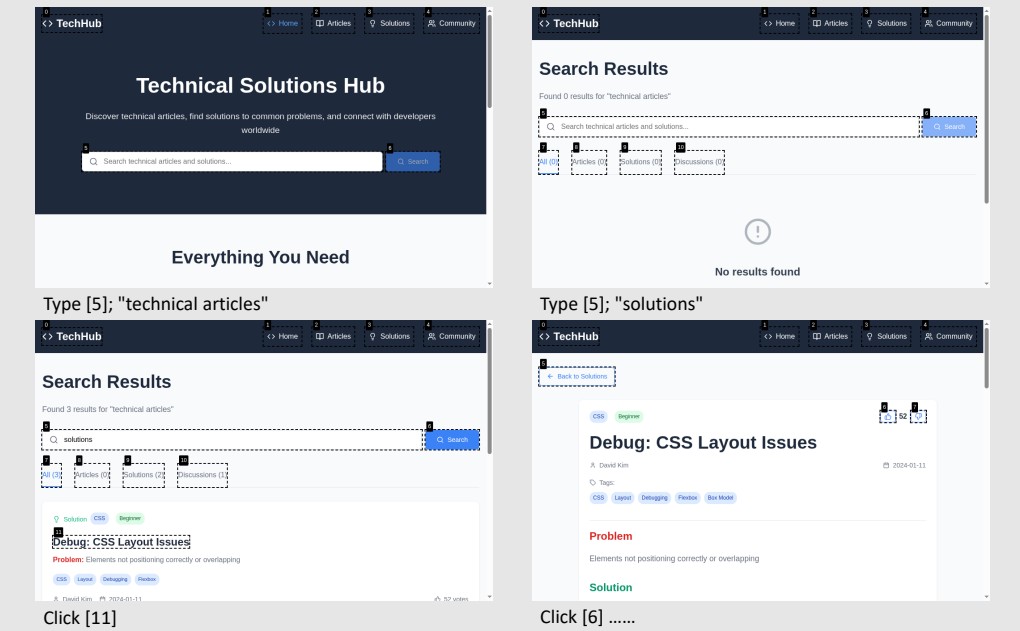

Figure 27: Example of a step in a WebGen-Agent trajectory where the agent improves the website's appearance based on GUI-agent testing feedback. The step is simplified due to space constraints.

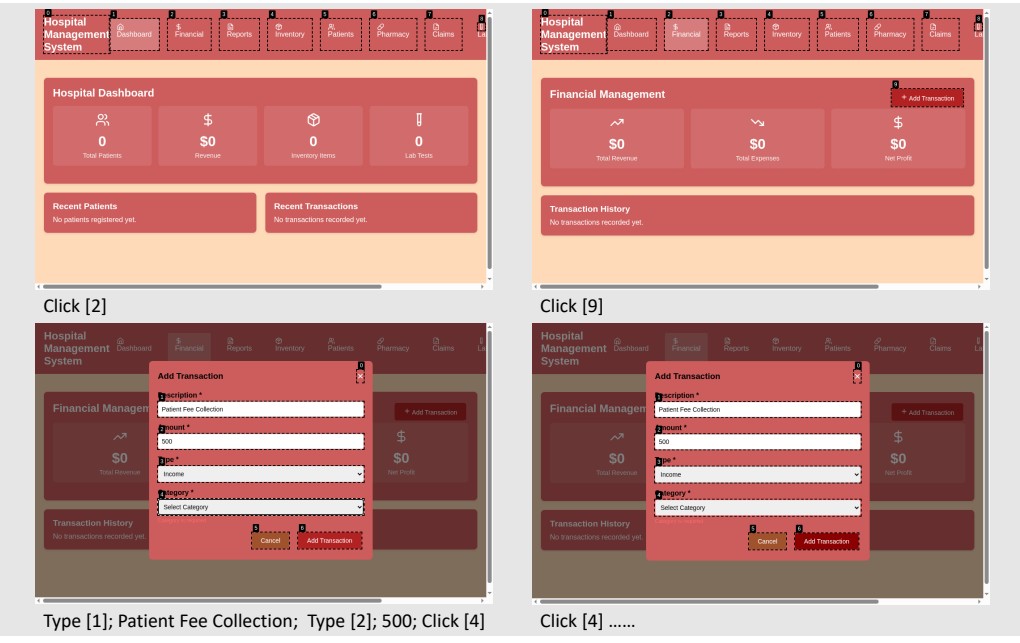

Click [2]    Click [9]

Type [1]; Patient Fee Collection;  Type [2]; 500; Click [4]    Click [4] ......

Suggestions: The transaction submission functionality is not working properly. This prevents users from recording financial data and blocks testing of other dependent features. The system should ensure that the transaction is properly .......

I need to debug and fix the transaction submission issue. Let me add better error handling and ensure the transaction data is properly saved and displayed.\n\n<boltArtifact id="hospital-management-system" title="Hospital Management System">\n<boltAction type="file" filePath="src/components/FinancialManagement.jsx">import React ......

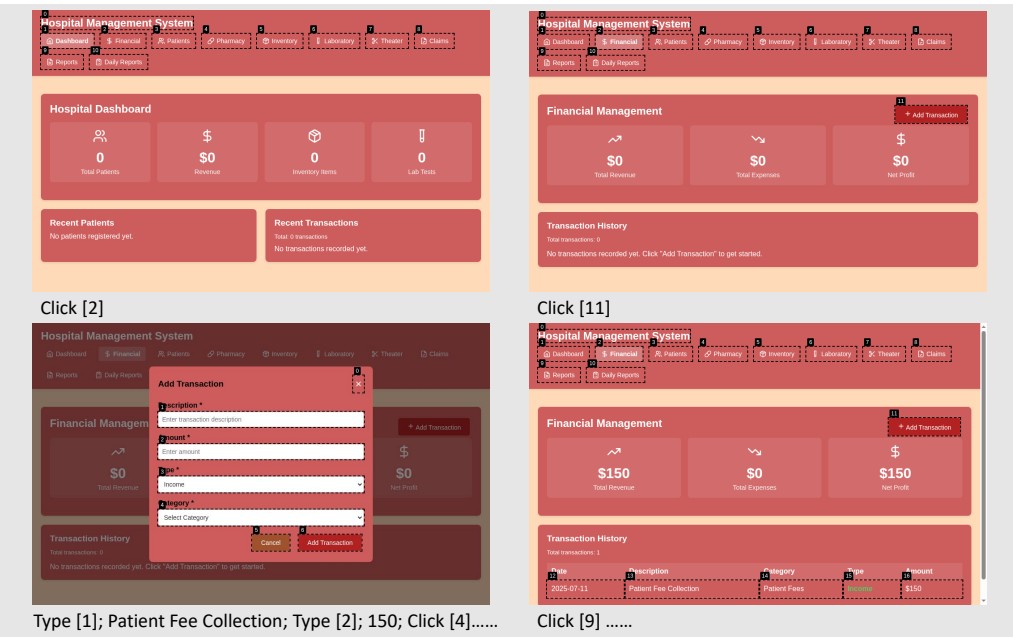

Click [2]    Click [11]

Type [1]; Patient Fee Collection; Type [2]; 150; Click [4]......    Click [9] ......

Figure 28: Example of a step in a WebGen-Agent trajectory where the agent improves the website's appearance based on GUI-agent testing feedback. The step is simplified due to space constraints.

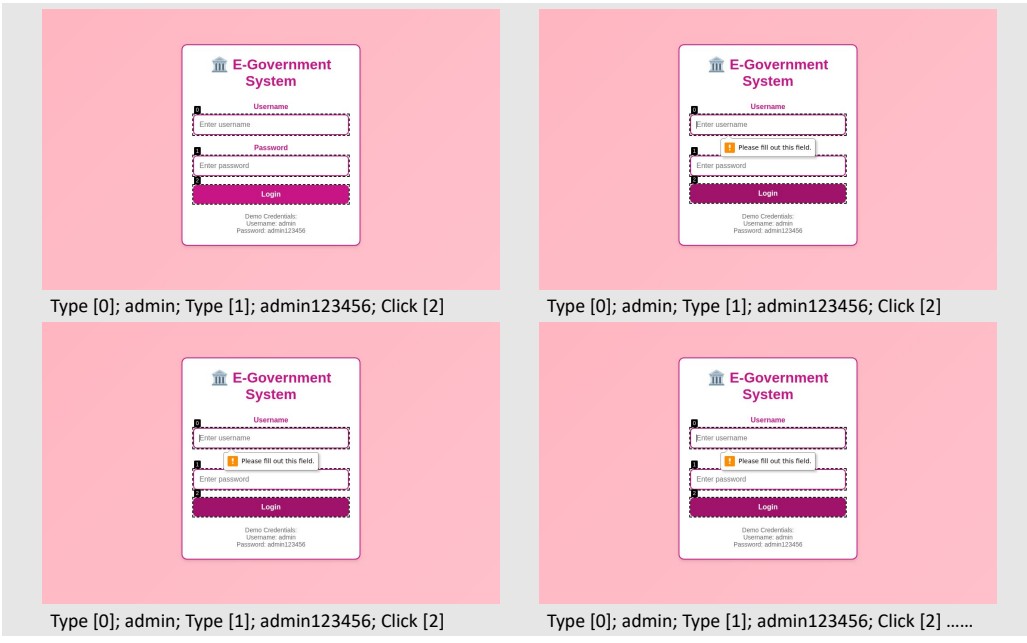

Suggestions: The login form has a critical issue where the password field fails to register or retain entered values, consistently showing 'password field is empty' errors despite multiple input attempts. This suggests problems with form input handling, data binding ……

I need to fix the login form issue where the password field is not properly registering input values. Let me update the Login component to ensure proper form handling and validation.\n\n<boltArtifact id="e-government-system" title="E-Government Office Management System">\n<boltAction type="file" filePath= ……

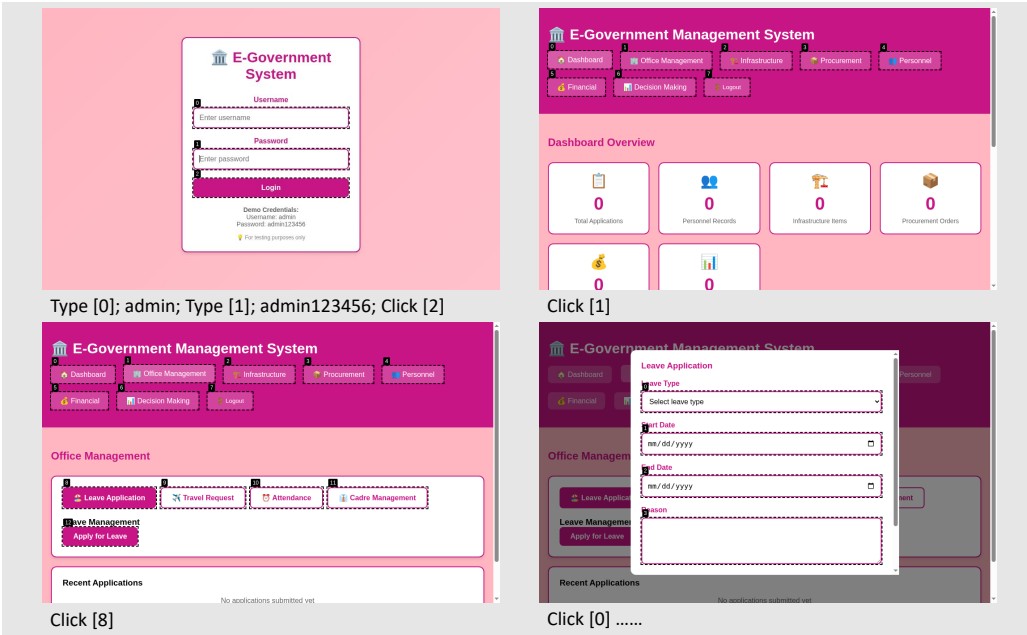

Figure 29: Example of a step in a WebGen-Agent trajectory where the agent improves the website's appearance based on GUI-agent testing feedback. The step is simplified due to space constraints.

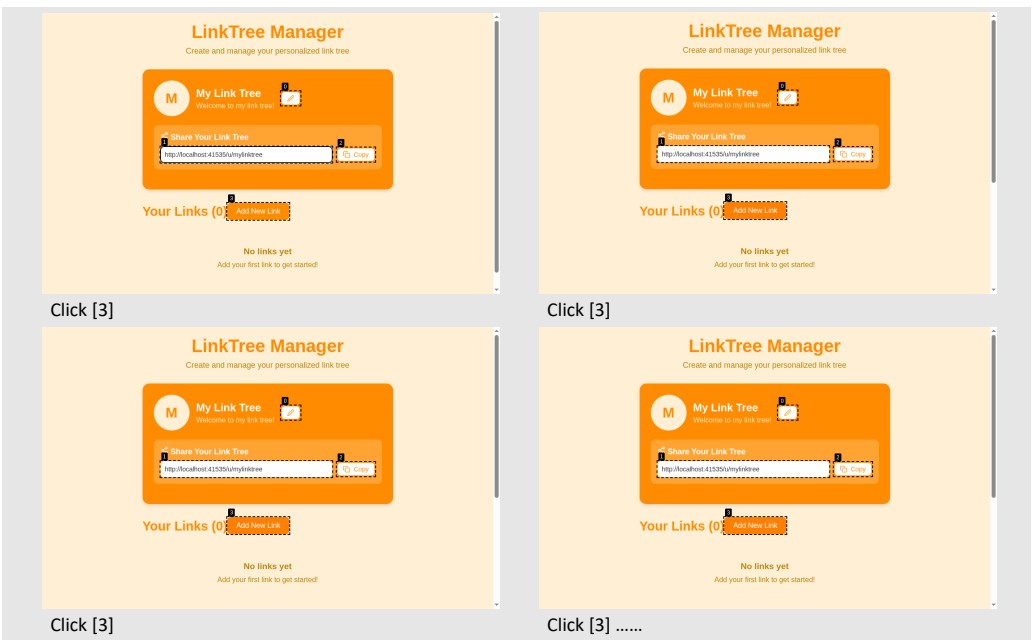

Suggestions: The 'Add New Link' button appears to be completely non-functional, as evidenced by 13 consecutive clicks with no response or state change. This is a critical issue that prevents users from accessing the core functionality of the LinkTree Manager ……

I need to fix the non-functional "Add New Link" button. Let me debug and update the code to ensure proper functionality.\n\n<boltArtifact id="linktree-website" title="Link Tree Website with CRUD Operations">\n<boltAction type="file" filePath="src/components/Dashboard.jsx">import React, { useState } from 'react' ……

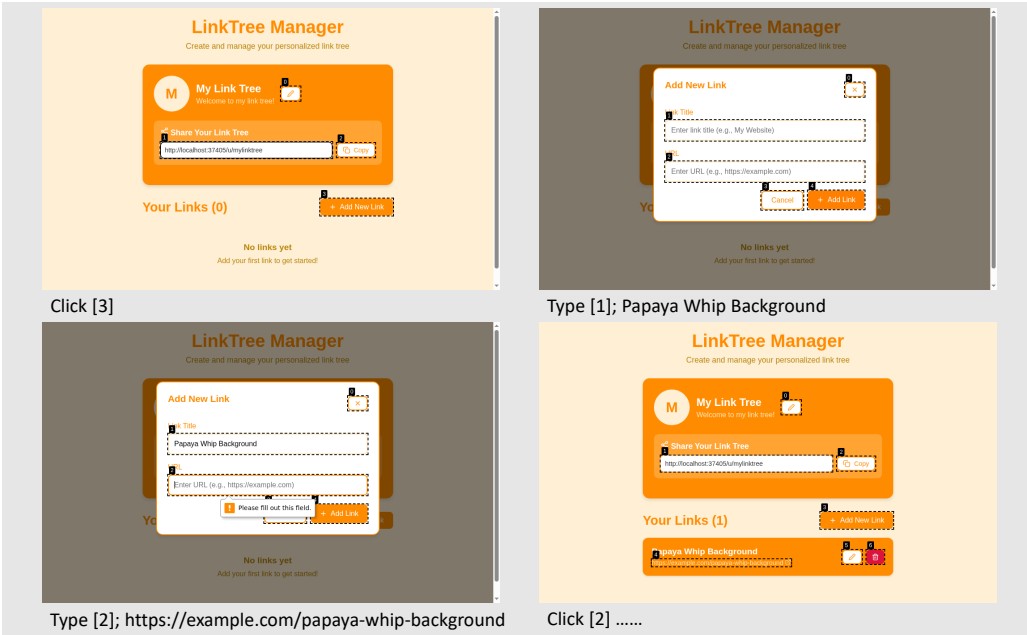

Figure 30: Example of a step in a WebGen-Agent trajectory where the agent improves the website's appearance based on GUI-agent testing feedback. The step is simplified due to space constraints.

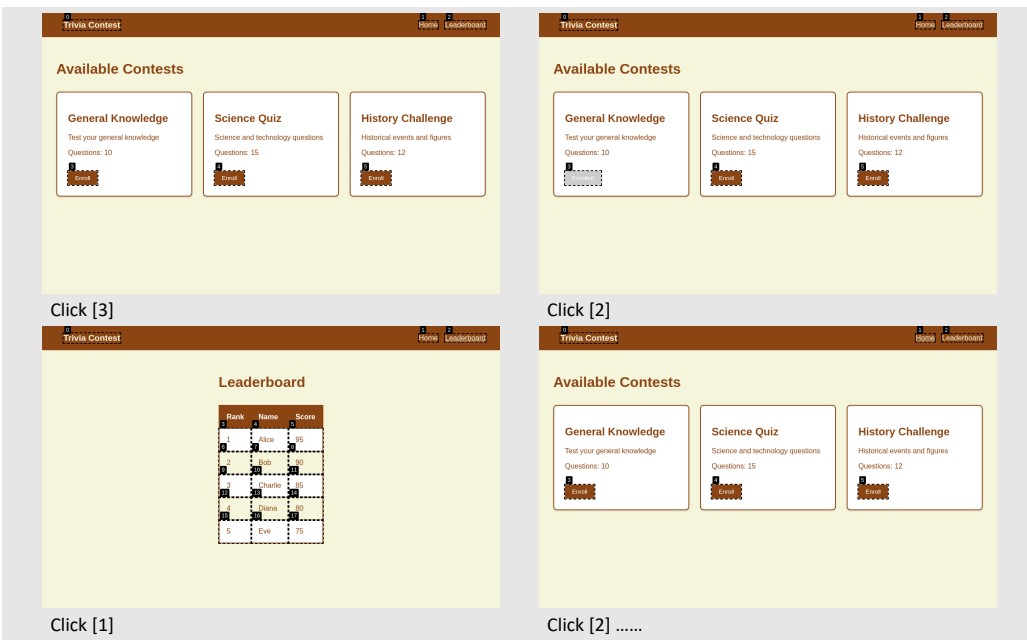

Click [3]

Click [2]

Click [1]

Click [2] ......

Suggestions: 1. Ensure contest questions are accessible after enrollment, either by redirecting users to the questions automatically or providing a clear navigation path (e.g., a 'Start Contest' button). 2. Add a dedicated section or page for answering questions and submitting ......

I'll enhance the app by adding contest access after enrollment and improving the question flow. Here are the modifications:\n\n<boltArtifact id="trivia-contest-enhancements" title="Trivia Contest Enhancements">\n  <boltAction type="file" filePath="src/pages/Home.jsx">import { useState } from 'react' ......

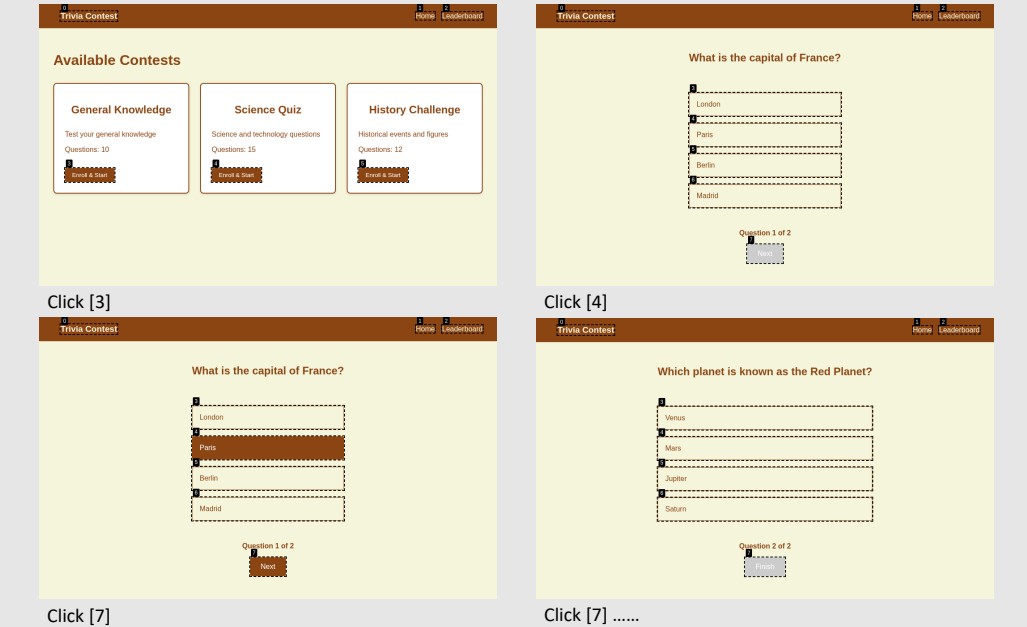

Click [3]

Click [4]

Click [7]

Click [7] ......

Figure 31: Example of a step in a WebGen-Agent trajectory where the agent improves the website's appearance based on GUI-agent testing feedback. The step is simplified due to space constraints.

