# OpenReview forum: "WebGen-Agent: Enhancing Interactive Website Generation with Multi-Level Feedback and Step-Level Reinforcement Learning"
_ICLR.cc/2026/Conference — ICLR 2026 Poster_

### Official Review · Reviewer_Cbyh · 2025-10-28

**Soundness:** 3
**Presentation:** 2
**Contribution:** 4
**Rating:** 4
**Confidence:** 3

**Summary:**

This paper introduces WebGen-Agent, a framework for website code generation that jointly leverages execution-based functional feedback and visual feedback to iteratively refine website code.
In addition, the authors propose Step-GRPO, which uses automated functional and visual feedback from WebGen-Agent as step-wise reward signals.
Experimental results on WebGen-Bench validate that both WebGen-Agent and Step-GRPO meaningfully improve functional accuracy and appearance scores.

**Strengths:**

- Extensive experimental evaluation.

- Proposes both training-free (WebGen-Agent) and training-based (Step-GRPO) approaches.

**Weaknesses:**

- Human annotation transparency (Appendix E, Tables 5 and 6): Details of human evaluation are missing, including the number of annotators, recruitment method, compensation, annotation interface, and inter-annotator agreement (e.g., Cohen's kappa).

- Execution-only already beats Bolt.diy: The execution-only baseline in the table already surpasses Bolt.diy’s reported accuracy, yet the authors do not clarify why their setting performs better even before applying the proposed method. There might be potential mismatches in iteration counts or evaluation conditions (e.g., 20 iterations for Bolt.diy).

- Relation to prior work (Bolt.diy): The difference is only briefly mentioned ("similar to our work" in line 452), but the precise factors behind the performance gap (e.g., reward type, iteration loop design, or evaluation feedback fidelity) are not explained.

- Ambiguity in backbone settings: Some reports (e.g., Table 2, Figure 12) omit which LLM backbone was used, complicating interpretation.

**Questions:**

Some of suggestions:
- Please include the missing details of the human evaluation process.

- Please clarify why the execution-only baseline already outperforms Bolt.diy.

- Please provide more detailed distinctions between WebGen-Agent and related works such as Aider and Bolt.diy.

- Please clarify which backbone is used in every Figure and Table.

- Since Step-GRPO uses training-time feedback, the reward quality could vary depending on whether the test cases are gold or model-generated. If gold test cases exist in WebGen-Instruct, it would be insightful to compare benchmark performance when using gold vs. generated test cases as reward signals.

- A recent work, namely TDDev [1], achieves 78.2 / 70.2 accuracy with GPT-4.1 / Claude-4 backbones on WebGen-Bench. It would be interesting to check whether WebGen-Agent scales with TDDev or to compare WebGen-Agent with TDDev under the same backbone (e.g., Qwen3-Coder-30B-A3B-Inst.).

[1] Wan, Yuxuan, et al. "Automatically Generating Web Applications from Requirements via Multi-Agent Test-Driven Development." arXiv preprint arXiv:2509.25297 (2025).

---

> ### Author Response · Authors · 2025-11-18
> **Response to Official Review by Reviewer Cbyh (1/3)**
>
> Thank you very much for the time and effort you have given to reviewing our work. We deeply appreciate your valuable suggestions, which are invaluable in improving our project.
>
> **Q1.** Human annotation transparency (Appendix E, Tables 5 and 6): Details of human evaluation are missing, including the number of annotators, recruitment method, compensation, annotation interface, and inter-annotator agreement (e.g., Cohen's kappa). Please include the missing details of the human evaluation process.
>
> **A1.** Thank you for your thoughtful suggestion. The manual testing results serve as the ground truth and require precision; therefore, **three human testers** independently annotated the results and we assessed the consistency of their annotations. If the annotations of a test case are inconsistent, a fourth human tester is tasked with re-examining the test case and the inconsistent annotations to decide on a final annotation. The testers are **student volunteers with bachelor’s degrees** in computer science related majors who are studying for a master’s degree. We also compute **Cohen’s κ** between the three human annotators to evaluate inter-annotator agreement, as shown in Table A and Table B below (Cohen’s κ 1–2 denotes Cohen's κ between annotator 1 and annotator 2, the others are similar). All of Cohen's κ values are above 0.8, with most above 0.9, demonstrating strong inter-annotator agreement. Additionally, the **annotation interface screenshots** and **grading guidelines** together with all the information above have been added to Appendix E and Appendix G in the revised paper.
>
> Table A:
> | Score Type | Coding LLM | Feedback VLM | Accuracy (%) | Std. Error | Cohen’s κ 1–2 | Cohen’s κ 1–3 | Cohen’s κ 2–3 | Avg. Cohen’s κ |
> |----|----|---|---|--|---|--|--|--|
> | Screenshot | Claude-4-Sonnet | Qwen2.5-VL-32B-Inst. | 93.6 | 0.25 | 0.959 | 0.964 | 0.933 | 0.952 |
> | Screenshot | Qwen3-Coder-30B-A3B-Inst. | Qwen2.5-VL-32B-Inst. | 93.9 | 0.26 | 0.964 | 0.984 | 0.978 | 0.975 |
> | Screenshot | Qwen3-Coder-480B-A35B-Inst. | Qwen2.5-VL-32B-Inst. | 95.6 | 0.20 | 0.948 | 0.953 | 0.912 | 0.938 |
> | Screenshot | DeepSeek-V3 | Qwen2.5-VL-32B-Inst. | 94.8 | 0.22 | 0.968 | 0.982 | 0.985 | 0.979 |
> | Screenshot | DeepSeek-V3 | GPT-4o | 95.5 | 0.20 | 0.980 | 0.986 | 0.970 | 0.979 |
> | GUI agent | Claude-4-Sonnet | Qwen2.5-VL-32B-Inst. | 90.1 | 0.31 | 0.973 | 0.946 | 0.932 | 0.950 |
> | GUI agent | Qwen3-Coder-30B-A3B-Inst. | Qwen2.5-VL-32B-Inst. | 91.4 | 0.44 | 0.933 | 0.885 | 0.941 | 0.920 |
> | GUI agent | Qwen3-Coder-480B-A35B-Inst. | Qwen2.5-VL-32B-Inst. | 89.6 | 0.41 | 0.967 | 0.971 | 0.943 | 0.960 |
> | GUI agent | DeepSeek-V3 | Qwen2.5-VL-32B-Inst. | 91.2 | 0.36 | 0.947 | 0.960 | 0.911 | 0.939 |
> | GUI agent | DeepSeek-V3 | GPT-4o | 92.2 | 0.33 | 0.928 | 0.977 | 0.948 | 0.951 |
>
> Table B:
> | Model | 5 | 4 | 3 | 2 | 1 | Avg. Score | Cohen’s κ 1–2 | Cohen’s κ 1–3 | Cohen’s κ 2–3 | Avg. Cohen’s κ |
> |-------|----|----|----|----|----|-----------|---------------|---------------|---------------|----------------|
> | Claude-4-Sonnet | 84.2 | 13.9 | 2 | 0 | 0 | 4.82 | 0.873 | 0.807 | 0.930 | 0.870 |
> | DeepSeek-V3 | 73.3 | 24.8 | 2 | 0 | 0 | 4.71 | 0.940 | 0.922 | 0.864 | 0.908 |
> | Qwen3-Coder-30B-A3B-Inst. | 75.2 | 23.8 | 1 | 0 | 0 | 4.74 | 0.914 | 0.914 | 0.880 | 0.903 |
> | Qwen3-Coder-480B-A35B-Inst. | 76.2 | 21.8 | 2 | 0 | 0 | 4.74 | 0.868 | 0.822 | 0.860 | 0.850 |
> | **Total** | **77.2** | **21.0** | **1.7** | **0** | **0** | **4.75** | **0.898** | **0.881** | **0.871** | **0.883** |
>
> **Q2.** Execution-only already beats Bolt.diy: The execution-only baseline in the table already surpasses Bolt.diy’s reported accuracy, yet the authors do not clarify why their setting performs better even before applying the proposed method. There might be potential mismatches in iteration counts or evaluation conditions (e.g., 20 iterations for Bolt.diy). Please clarify why the execution-only baseline already outperforms Bolt.diy.
>
> **A2.** Thank you for your insightful question. Bolt.diy is a website generation system that, upon receiving a user instruction, first imports a template, then generates updated content of various files for the template, and finally installs the dependencies and starts the service in a web container to provide a preview to the user. While the generated codebase is executed, the executed results are, in fact, not returned to the model. **This often results in blank templates, start-up failures, and erroneous webpages.** In contrast, our execution-only baseline returns any error messages and console output generated during the dependency installation and service initialization back to the coding LLM, thus enabling the model to iteratively improve the generated codebase, greatly improving the accuracy of the results. This explains why the execution-only baseline already outperforms Bolt.diy. We have added the explanation about Bolt.diy to the Related Work section in the revised paper.

---

> ### Author Response · Authors · 2025-11-18
> **Response to Official Review by Reviewer Cbyh (2/3)**
>
> **Q3.** Relation to prior work (Bolt.diy): The difference is only briefly mentioned ("similar to our work" in line 452), but the precise factors behind the performance gap (e.g., reward type, iteration loop design, or evaluation feedback fidelity) are not explained. Please provide more detailed distinctions between WebGen-Agent and related works such as Aider and Bolt.diy.
>
> **A3.** Thank you for your valuable suggestion. OpenHands and Aider are general-purpose code agents that are not limited to a single field, though their website development abilities do not match that of specialist code agents such as Bolt.diy. Their relatively low performance in website development might be due to the fact that their tools are not suited to website development (inaccurate code localization, inability to write multiple files in a single response), and their general system prompts.
>
> Similar to our work, Bolt.diy specializes in multi-file website generation. Upon receiving a user instruction, it first imports a template, then generates updated content of various files for the template, and finally installs the dependencies and starts the service in a web container to provide a preview to the user. While the generated codebase is executed, the executed results are, in fact, not returned to the model. This often results in blank templates, start-up failures, and erroneous webpages. In contrast, our WebGen-Agent is a code agent specializing in end-to-end website generation, with screenshot feedback and GUI-agent testing features specifically designed for this task, achieving state-of-the-art performance. We have added this to the Related Work section in the revised paper.
>
> **Q4.** Ambiguity in backbone settings: Some reports (e.g., Table 2, Figure 12) omit which LLM backbone was used, complicating interpretation. Please clarify which backbone is used in every Figure and Table.
>
> **A4.** Thank you for your valuable comment. The LLM backbones used in Table 2 and Figure 12 are both DeepSeek-V3. We have added this to the captions of Table 2, Table 11, and Figure 17 in the revised paper.
>
> **Q5.** Since Step-GRPO uses training-time feedback, the reward quality could vary depending on whether the test cases are gold or model-generated. If gold test cases exist in WebGen-Instruct, it would be insightful to compare benchmark performance when using gold vs. generated test cases as reward signals.
>
> **A5.** Thank you for the insightful suggestion. However, the training samples in WebGen-Instruct **do not contain gold test cases**. Creating gold test cases for large amounts of training website-generation instructions would require costly manual annotation and is difficult to scale. In comparison, our *Step-GRPO with Screenshot and GUI-agent Feedback* use the appearance and functional scores inherently generated in the WebGen-Agent pipeline to provide accurate and effective supervision for reinforcement learning. It is both cost-effective and good at improving the website-generation ability of the coding LLM, increasing the accuracy by 6.5% and 4.8% when applied to Qwen2.5-Coder-7B-Instruct and Qwen3-8B, respectively.

---

> ### Author Response · Authors · 2025-11-18
> **Response to Official Review by Reviewer Cbyh (3/3)**
>
> **Q6.** A recent work, namely TDDev [1], achieves 78.2 / 70.2 accuracy with GPT-4.1 / Claude-4 backbones on WebGen-Bench. It would be interesting to check whether WebGen-Agent scales with TDDev or to compare WebGen-Agent with TDDev under the same backbone (e.g., Qwen3-Coder-30B-A3B-Inst.).
>
> **A6.** Thank you for your valuable suggestion. First of all, it is important to note that TDDev [1] was first made **publicly available on arXiv on 29 Sep 2025, which is later than the ICLR paper submission deadline (Sep 24 2025)**. Therefore, we could not have compared our work to TDDev in our submitted paper. As mentioned in the ICLR 2026 guidelines [2]: *“We consider papers contemporaneous if they are published within the last two months. That means, since our full paper deadline is September 24, if a paper was published (i.e., at a peer-reviewed venue) on or after July 24, 2025, authors are not required to compare their own work to that paper.”* A work that was released on arXiv **after** the submission deadline certainly falls into that category.
>
> Regarding the reported 78.2 / 70.2 accuracy with GPT-4.1 / Claude-4 in TDDev, we would like to draw attention to the fact that these results were acquired by testing on **only 10 instructions** sampled from the 101 testing instructions in WebGen-Bench, as mentioned in Page 11, Section 4.6.1 of the TDDev paper [1]: *“To perform the evaluation, we  randomly select 10 websites while preserving the category distribution of the test set.”* This introduces potential **bias** to the testing process, as the sampled test set could be simpler than the whole test set. Also, testing on only 10 instructions greatly increases the instability and inaccuracy of the evaluation. In contrast, our experiments are all conducted on the full WebGen-Bench test set of 101 instructions.
>
> Following your request, we have evaluated TDDev on the full WebGen-Bench test set of 101 instructions, as shown in the table below. We use Qwen3-Coder-480B-A35B-Instruct as the coding LLM and Qwen2.5-VL-32B-Instruct as the VLM for fair comparison. Our evaluation only yields an accuracy of 44.0% and an appearance score of 3.7 on TDDev, which are lower than WebGen-Agent by 14.2% and 0.6, respectively. There are several possible reasons for this: (1) the accuracy in the TDDev paper [1] is inflated due to only testing on a small subset of WebGen-Bench. (2) TDDev added a golden “Design Image” generated by Gemini-2.5-Flash-Image to each of the samples they tested, thus adding additional hints and simplifying the task. When tested without this golden Design Image, the accuracy is further reduced.
>
> Compared to TDDev, our *WebGen-Agent* pipeline introduces explicit screenshot and GUI-agent testing feedback, including text feedback and quantitative scores, iteratively improving both appearance and functionality of the generated website. Additionally, the integration of backtracking and select-best mechanisms further improves the performance. We also propose *Step-GRPO with Screenshot and GUI-agent Feedback*, a highly effective reinforcement learning method that leverages the screenshot and GUI-agent scores inherently generated in the WebGen-Agent pipeline to provide accurate and reliable supervision, greatly improving the coding LLMs’ ability to generate high-quality websites.
>
> | Tool           | Yes | Partial | No  | Start Failed | Accuracy | Appearance Score |
> |-----------------|----:|--------:|----:|-------------:|---------:|-----------------:|
> | TDDev           | 36.9 | 14.1 | 49.0 | 0 | 44.0 | 3.7 |
> | WebGen-Agent    | 50.5 | 15.3 | 34.2 | 0 | 58.2 | 4.3 |
>
> [1] Wan, Yuxuan, et al. "Automatically Generating Web Applications from Requirements via Multi-Agent Test-Driven Development." arXiv preprint arXiv:2509.25297 (2025).
>
> [2] https://iclr.cc/Conferences/2026/ReviewerGuide

---

> > ### Comment · Reviewer_Cbyh · 2025-11-25
> > **Official Comment by Reviewer Cbyh**
> >
> > Thank you for the thorough response. Most of my concerns have been addressed, and I have one small follow-up question related to **A5**.
> >
> > Since there is no gold test cases and GRPO training relies solely on generated ones, a brief analysis of the accuracy of these generated test cases could further strengthen the work, as it may offer insights into noise filtering strategies that could improve reward signals in future studies.

---

> > > ### Author Response · Authors · 2025-11-25
> > > **Thank you for acknowledging our rebuttal efforts**
> > >
> > > Thank you very much for acknowledging our rebuttal efforts. We are grateful for your insightful comments and suggestions, which have been invaluable in improving our project. We would like to address your follow-up question below.
> > >
> > > **Analysis of the accuracy of generated test cases**
> > >
> > > To analyze the accuracy of the generated test cases, we randomly sampled 100 test cases and inspected them manually. We judge the test case of a website development instruction to be accurate if it fulfills the following two conditions:
> > > 1. The test case covers all the requirements in the instruction.
> > > 2. Each part of the test case corresponds to a requirement in the instruction.
> > >
> > > The accuracy of the 100 test cases is 87%. Among the 100 test cases, only 13 show slight mismatches with the website development instructions, demonstrating the effectiveness of the generated test cases. As shown in the experiments in our paper, the accuracy of the test cases is already enough to bring significant improvements to the website generation process. Future work might further benefit from noise filtering strategies that could improve reward signals, which is an interesting future direction but is beyond the scope of this paper. The analysis has been added to Appendix G of the revised paper. We have also added the test results in `data/test_case_accuracy_100.json` in the supplementary materials.
> > >
> > > Thank you once again for the time and effort you have dedicated to reviewing our work.

---

> > > > ### Comment · Reviewer_Cbyh · 2025-11-26
> > > > **Official Comment by Reviewer Cbyh**
> > > >
> > > > Thank you for the analysis. It is interesting to see that the accuracy of the generated test cases is quite high; this may be inherent to the task itself, since website generation naturally involves minor edge cases unlike standard code generation.
> > > >
> > > > I have no further concerns, and I have updated my score accordingly.

---

> > > > > ### Author Response · Authors · 2025-11-26
> > > > > **Thank you very much for acknowledging our rebuttal efforts**
> > > > >
> > > > > Thank you for acknowledging our rebuttal efforts. We truly appreciate the time and effort you dedicated to reviewing our paper. Your comments and suggestions were extremely helpful in improving our work.
> > > > >
> > > > > Best regards,
> > > > > The Authors

---

### Official Review · Reviewer_jD87 · 2025-10-29

**Soundness:** 3
**Presentation:** 3
**Contribution:** 3
**Rating:** 6
**Confidence:** 3

**Summary:**

This work proposes WebGen-Agent, which augments a website coding agent with two automatic evaluators: a VLM that scores a landing‑page screenshot for visual quality, and a GUI agent that exercises the site and grades functionality. These signals are used in two modes: (i) inference‑time self‑revision for closed‑source LLMs (iteratively editing the codebase), and (ii) training for open models GRPO at every editing iteration of the codebase. The author compares WebGen-Agent with existing general-purpose coding agents on the WebGen-Bench dataset. The results suggest that the proposed method outperforms competing methods.

**Strengths:**

1. It is a novel and interesting idea that combines VLM‑based visual feedback with a GUI agent for functional feedback, moving beyond execution‑only signals.

2. On WebGen‑Bench, the method significantly outperforms general‑purpose coding agents that lack VLM/GUI feedback, indicating the value of integrating visual and functional signals.

3. The author conducts experiments with various open and proprietary LLMs, as well as thorough ablation studies.

**Weaknesses:**

1. The VLM feedback is based on a single-page screenshot, which ignores the coherence and consistency across multiple pages.
2. The reliability of the functionality reward is bottlenecked by the performance of the GUI agent. However, this remains a challenging and open problem, especially when the website is complex.
3. Some experiment details are missing and confusing. For instance, the author doesn't mention how the human annotation is conducted. Also, OpenHands is an open platform; it's unclear how it is included as a baseline.
4. While the proposed method is novel in the area of web coding agents, similar ideas have been well explored in a broader field: using external LLM-as-judge and/or agent-as-judge to provide reward or natural language feedback, fine-tuning the LLM with an existing RL algorithm.

**Questions:**

Questions:

* How did you implement the OpenHands baseline?
* What is the exact definition of Accuracy in Table 3? Is it computed from Yes/Partial/No/Start Failed?
* Why does the experiment only adopt one dataset? There are other existing website coding benchmark such as Web2Code and Design2Code.

Typo:
1. L235: Wrong inline citation format
2. L134: is used to reflect**s**
3. L1044: The definition of the scores are presented (subject-verb agreement)

---

> ### Author Response · Authors · 2025-11-18
> **Response to Official Review by Reviewer jD87 (1/4)**
>
> Thank you for the time and effort you have dedicated to reviewing our project. Your insightful comments and suggestions are invaluable to us for improving our work.
>
> **Q1.** The VLM feedback is based on a single-page screenshot, which ignores the coherence and consistency across multiple pages.
>
> **A1.** Thank you for raising the question of cross-page coherence. This is an important aspect of web design, and we appreciate the opportunity to clarify our design choice. Our evaluation focuses on page-level appearance, which is where VLMs currently demonstrate the highest reliability. Extending the task to multi-page inputs would introduce a qualitatively different problem formulation, as cross-page coherence is inherently subjective and highly context-dependent (e.g., narrative flow, functional layout transitions, or multi-team design).
>
> Importantly, strict cross-page coherence is not a universal property of real-world websites. Large platforms and government/university websites frequently contain pages with noticeably different visual styles, layouts, and interaction patterns, often due to legacy modules, diverse content types, or independent feature development. This means imposing a coherence requirement may conflate model performance with assumptions that do not hold across real web ecosystems.
>
> For these reasons, we adopt a single-page evaluation paradigm consistent with prior VLM benchmarks on screen/page-level understanding. We have added this clarification to Section 2.1 of the revised paper.
>
> **Q2.** The reliability of the functionality reward is bottlenecked by the performance of the GUI agent. However, this remains a challenging and open problem, especially when the website is complex.
>
> **A2.** Thank you for the thoughtful comment. We agree that the functionality reward depends on the capability of the GUI agent. However, this dependency is not a limitation specific to our approach—it reflects a broader, well-recognized challenge in GUI automation.
>
> Importantly, our experiments show that even with the current WebVoyager agent, the functionality reward provides meaningful and consistent feedback and learning signals. Using no agent at all would remove an essential source of behavioral feedback. As shown in Table 3 in the paper and the table below, adding the GUI-agent score in the GRPO training significantly improves the performance of Qwen2.5-Coder-7B-Instruct, increasing the accuracy by 5.2% and the appearance score by 0.2. This demonstrates the effectiveness and necessity of the GUI agent feedback.
>
> | Test Name                                   | Yes | Partial |  No  | Start Failed | Accuracy      | Appearance Score |
> |---------------------------------------------|-----:|--------:|-----:|-------------:|---------------|-----------------|
> | Step-GRPO w/ Screenshot Reward Only         | 34.9 | 10.5    | 53.9 | 0.6          | 40.2          | 3.5             |
> | Step-GRPO w/ Screenshot + GUI-agent | 40.2 | 10.5    | 49.3 | 0            | 45.4 (+5.2)   | 3.7 (+0.2)      |
>
> Moreover, our design is agent-agnostic: as GUI agents improve, the functionality reward naturally becomes more accurate and more informative. Thus, our method is expected to benefit directly from advances in GUI interaction models. We have clarified this connection and emphasized that our method works effectively with current agents while remaining compatible with stronger agents in Appendix A.

---

> ### Author Response · Authors · 2025-11-18
> **Response to Official Review by Reviewer jD87 (2/4)**
>
> **Q3.** The author doesn't mention how the human annotation is conducted.
>
> **A3.** Thank you for your valuable comment. The manual testing results serve as the ground truth and require precision; therefore, **three human testers** independently annotated the results and we assessed the consistency of their annotations. If the annotations of a test case are inconsistent, a fourth human tester is tasked with re-examining the test case and the inconsistent annotations to decide on a final annotation. The testers are **student volunteers with bachelor’s degrees** in computer science related majors who are studying for a master’s degree. We also compute **Cohen’s κ** between the three human annotators to evaluate inter-annotator agreement, as shown in Table A and Table B below (Cohen’s κ 1–2 denotes Cohen's κ between annotator 1 and annotator 2, the others are similar). All of Cohen's κ values are above 0.8, with most above 0.9, demonstrating strong inter-annotator agreement. Additionally, the **annotation interface screenshots** and **grading guidelines** together with all the information above have been added to Appendix E and Appendix G in the revised paper.
>
> Table A:
> | Score Type | Coding LLM | Feedback VLM | Accuracy (%) | Std. Error | Cohen’s κ 1–2 | Cohen’s κ 1–3 | Cohen’s κ 2–3 | Avg. Cohen’s κ |
> |------------|------------|--------------|--------------|------------|---------------|---------------|---------------|----------------|
> | Screenshot | Claude-4-Sonnet | Qwen2.5-VL-32B-Inst. | 93.6 | 0.25 | 0.959 | 0.964 | 0.933 | 0.952 |
> | Screenshot | Qwen3-Coder-30B-A3B-Inst. | Qwen2.5-VL-32B-Inst. | 93.9 | 0.26 | 0.964 | 0.984 | 0.978 | 0.975 |
> | Screenshot | Qwen3-Coder-480B-A35B-Inst. | Qwen2.5-VL-32B-Inst. | 95.6 | 0.20 | 0.948 | 0.953 | 0.912 | 0.938 |
> | Screenshot | DeepSeek-V3 | Qwen2.5-VL-32B-Inst. | 94.8 | 0.22 | 0.968 | 0.982 | 0.985 | 0.979 |
> | Screenshot | DeepSeek-V3 | GPT-4o | 95.5 | 0.20 | 0.980 | 0.986 | 0.970 | 0.979 |
> | GUI agent | Claude-4-Sonnet | Qwen2.5-VL-32B-Inst. | 90.1 | 0.31 | 0.973 | 0.946 | 0.932 | 0.950 |
> | GUI agent | Qwen3-Coder-30B-A3B-Inst. | Qwen2.5-VL-32B-Inst. | 91.4 | 0.44 | 0.933 | 0.885 | 0.941 | 0.920 |
> | GUI agent | Qwen3-Coder-480B-A35B-Inst. | Qwen2.5-VL-32B-Inst. | 89.6 | 0.41 | 0.967 | 0.971 | 0.943 | 0.960 |
> | GUI agent | DeepSeek-V3 | Qwen2.5-VL-32B-Inst. | 91.2 | 0.36 | 0.947 | 0.960 | 0.911 | 0.939 |
> | GUI agent | DeepSeek-V3 | GPT-4o | 92.2 | 0.33 | 0.928 | 0.977 | 0.948 | 0.951 |
>
> Table B:
> | Model | 5 | 4 | 3 | 2 | 1 | Avg. Score | Cohen’s κ 1–2 | Cohen’s κ 1–3 | Cohen’s κ 2–3 | Avg. Cohen’s κ |
> |-------|----|----|----|----|----|-----------|---------------|---------------|---------------|----------------|
> | Claude-4-Sonnet | 84.2 | 13.9 | 2 | 0 | 0 | 4.82 | 0.873 | 0.807 | 0.930 | 0.870 |
> | DeepSeek-V3 | 73.3 | 24.8 | 2 | 0 | 0 | 4.71 | 0.940 | 0.922 | 0.864 | 0.908 |
> | Qwen3-Coder-30B-A3B-Inst. | 75.2 | 23.8 | 1 | 0 | 0 | 4.74 | 0.914 | 0.914 | 0.880 | 0.903 |
> | Qwen3-Coder-480B-A35B-Inst. | 76.2 | 21.8 | 2 | 0 | 0 | 4.74 | 0.868 | 0.822 | 0.860 | 0.850 |
> | **Total** | **77.2** | **21.0** | **1.7** | **0** | **0** | **4.75** | **0.898** | **0.881** | **0.871** | **0.883** |
>
> **Q4.** OpenHands is an open platform; it's unclear how it is included as a baseline.  How did you implement the OpenHands baseline?
>
> **A4.** Thank you for the thoughtful question. The OpenHands results are taken from the WebGen-Bench paper [1]. They have explained the implementation details in their paper and open-sourced the evaluation code [2]. We have added an explanation about the source of the OpenHands results in Table 1 in the revised paper.
>
> [1] Lu, Zimu, et al. "WebGen-Bench: Evaluating LLMs on Generating Interactive and Functional Websites from Scratch." NeurIPS 2025.
>
> [2] https://github.com/mnluzimu/OpenHands-WebGen-Fork

---

> ### Author Response · Authors · 2025-11-18
> **Response to Official Review by Reviewer jD87 (3/4)**
>
> **Q5.** While the proposed method is novel in the area of web coding agents, similar ideas have been well explored in a broader field: using external LLM-as-judge and/or agent-as-judge to provide reward or natural language feedback, fine-tuning the LLM with an existing RL algorithm.
>
> **A5.** Thank you for pointing out the connection to prior work on LLM-as-judge and agent-as-judge reward modeling. However, directly applying generic LLM-as-judge approaches to the web coding setting is not feasible. General-purpose judges are known to struggle with evaluating the correctness of complex, multimodal, and highly structured outputs such as dynamic interactions and GUI rendering results. These evaluations require domain-specific grounding in visual appearance, functionality, and user interaction flows that generic judges are not designed to capture. Our method addresses precisely this gap by introducing domain-aligned reward signals tailored to web coding agents. Unlike standard language-only reward models, our evaluation integrates (1) rendered webpages, (2) visual and structural consistency, and (3) interaction-based functionality checks. These components go significantly beyond the scope of existing LLM-as-judge work and cannot be obtained by simply reusing a generic LLM-as-judge or agent-as-judge pipeline.
>
> A key contribution of our work is precisely the introduction of *WebGen-Agent*, a domain-aligned workflow for web coding agents, which defines how the agent should generate, render, inspect, and validate webpages. This workflow creates the necessary scaffolding for meaningful supervision in this domain. Another key contribution is *Step-GRPO with Screenshot and GUI-agent Feedback*, which is built on top of this workflow. It leverages the appearance and functional scores inherently generated by the WebGen-Agent pipeline to provide accurate and reliable step-level supervision for the RL algorithm. Without such a workflow, standard LLM-as-judge reward pipelines simply do not provide actionable or reliable signals for web coding tasks.
>
> **Q6.** What is the exact definition of Accuracy in Table 3? Is it computed from Yes/Partial/No/Start Failed?
>
> **A6.** Thank you for your valuable question. The definition of Accuracy in Table 3 follows the WebGen-Bench paper [1]. It is computed using a weighted score, where YES samples are weighted by 1 and PARTIAL samples are weighted by 0.5; the total score is then divided by the number of test cases. We have added this to the captions of Table 1, Table 2, and Table 3 in the revised paper.
>
> [1] Lu, Zimu, et al. "WebGen-Bench: Evaluating LLMs on Generating Interactive and Functional Websites from Scratch." NeurIPS 2025.

---

> ### Author Response · Authors · 2025-11-18
> **Response to Official Review by Reviewer jD87 (4/4)**
>
> **Q7.** Why does the experiment only adopt one dataset? There are other existing website coding benchmarks such as Web2Code and Design2Code.
>
> **A7.** Thank you for your insightful question. However, Web2Code [1] and Design2Code [2] do not match the use case of our WebGen-Agent pipeline, as they represent the task of taking in a **webpage screenshot** and generating its corresponding HTML file. In contrast, our WebGen-Agent aims to assist non-expert users to build **complicated, interactive websites** by creating multi-file codebases based on **natural language website-development instructions**. Our problem setting requires an agentic workflow, high-level planning, organizing complex multi-file codebases, and implementing nuanced user requirements in both functionality and design, while Web2Code and Design2Code tasks can be achieved by an MLLM itself and are better suited to testing basic MLLM abilities. Therefore, we did not test WebGen-Agent on Web2Code and Design2Code.
>
> To demonstrate the generalizability of our method, we test WebGen-Agent on the Web Application subset of the ArtifactsBench [3] dataset, which also requires generating a website based on user instructions, though their requirements are less complicated than the instructions in WebGen-Bench. The baseline results of the closed-source models and DeepSeek-V3 are taken from [3], while the others are tested using their open-sourced codebase. As shown in the table below, WebGen-Agent with Qwen3-Coder-480B-A35B-Instruct as the coding LLM achieves the best performance of 63.86, outperforming the baseline results of various closed-source models and the baseline result of Qwen3-Coder-480B-A35B-Instruct. WebGen-Agent with DeepSeek-V3 also outperforms the baseline result of DeepSeek-V3. This demonstrates the effectiveness and generalizability of WebGen-Agent. We have added this to Appendix M of the revised paper.
>
> | Test Name                                                | Average MLLM score |
> |----------------------------------------------------------|--------------------|
> | GPT-4o                                                   | 39.27 |
> | o1                                                       | 39.01 |
> | Claude 3.7-Sonnet                                        | 53.64 |
> | Claude 4.0-Sonnet                                        | 55.79 |
> | Gemini-2.5-Pro                                           | 58.30 |
> | DeepSeek-V3                                              | 47.47 |
> | **DeepSeek-V3 + WebGen-Agent**                               | 55.74 |
> | Qwen3-Coder-480B-A35B-Instruct                           | 55.42 |
> | **Qwen3-Coder-480B-A35B-Instruct + WebGen-Agent**            | **63.86** |
>
> [1] Yun, Sukmin, et al. "Web2code: A large-scale webpage-to-code dataset and evaluation framework for multimodal llms." Advances in neural information processing systems 37 (2024): 112134-112157.
>
> [2] Si, Chenglei, et al. "Design2code: Benchmarking multimodal code generation for automated front-end engineering." Proceedings of the 2025 Conference of the Nations of the Americas Chapter of the Association for Computational Linguistics: Human Language Technologies (Volume 1: Long Papers). 2025.
>
> [3] Zhang, Chenchen, et al. "Artifactsbench: Bridging the visual-interactive gap in llm code generation evaluation." arXiv preprint arXiv:2507.04952 (2025).
>
> **Q8.** L235: Wrong inline citation format; L134: is used to reflects; L1044: The definition of the scores are presented (subject-verb agreement)
>
> **A8.** Thank you for pointing out the typos. We have corrected them in the revised paper.

---

### Official Review · Reviewer_z3Vc · 2025-10-30

**Soundness:** 3
**Presentation:** 3
**Contribution:** 2
**Rating:** 4
**Confidence:** 3

**Summary:**

This paper focuses on the creation of webpages with LLM. In particular, it uses a recent benchmark  (`WebGen-Bench`, from [Lu et al 2025](https://arxiv.org/abs/2505.03733)) to develop an agentic scaffold tailored to it, and benchmarks a variety of models with that scaffolding.

On top of that, the paper proposes to finetune smaller (7B) models with data distillation through SFT and a GRPO-like loss for RL showing significant gains

**Strengths:**

* The resulting benchmark numbers are impressive. The evaluation benchmark is quite novel, but the gains are still significant

* The use of the reward signal in an RL setting is interesting by itself, and a good use-case of using VLM-guided models-as-a-judge evaluation as reward

**Weaknesses:**

* Most of the paper is dedicated to the agentic scaffolding, which I consider to be a minor contribution and that by itself alone is below the significance criteria for an ICLR paper

* Only one benchmark is used, and therefore it is not clear if the scaffolding or the general proposal generalizes to other datasets

* The final numbers of the 7B models are impressive, however most of the gains are obtained from the SFT step which consists in data-distilling Deepseek. The use of a large VLM model as judge also raises the question if the gains are not "just" distillation from a larger models, or if there is some learning that could generalize to larger models

**Questions:**

* From Table 4 it seems that the VLM used as judge does not necessarily have a major impact. Did you benchmark this behaviour when finetuning the smaller models with GRPO?

---

> ### Author Response · Authors · 2025-11-18
> **Response to Official Review by Reviewer z3Vc (1/4)**
>
> Thank you for your valuable questions and suggestions. They have been greatly helpful to us in improving our project.
>
> **Q1.** Most of the paper is dedicated to the agentic scaffolding, which I consider to be a minor contribution and that by itself alone is below the significance criteria for an ICLR paper.
>
> **A1.** Thank you for the comment. The main contribution of our paper is the overall agentic workflow, rather than the scaffolding module in isolation. In our paper, we introduce a novel website development agent, WebGen-Agent, which is **the first code agent to leverage visual feedback from screenshots and GUI agent trajectories** to iteratively improve the generated codebase, increasing the accuracy of Claude-3.5-Sonnet from 26.4% to 51.9% and its appearance score from 3.0 to 3.9, outperforming the previous state-of-the-art agent system. We further introduce *Step-GRPO with Screenshot and GUI-agent Feedback*, an innovative training method that leverages the **step-level appearance and functional scores** inherently generated in the WebGen-Agent pipeline as supervision signals for GRPO training, increasing the accuracy of Qwen2.5-Coder-7B-Instruct from **38.9% to 45.4%** and raises the appearance score from **3.4 to 3.7**.
>
> Notably, agentic workflows themselves have become an important research direction, with many recent works focusing on designing end-to-end agentic systems [1, 2, 3, 4]. In this context, our work introduces a workflow that substantially extends the capabilities demonstrated in prior papers on webpage generation or repository-level code manipulation. These prior systems often lack mechanisms for effective self-analysis, especially when visual content is involved in the code generation process. In contrast, by integrating a VLM and a GUI agent, our workflow enables the model to inspect the execution result of its previously generated code, analyze failures or mismatches, and refine its code based on this analysis. This structured, iterative improvement is not supported by existing agentic frameworks and represents a central, novel contribution. **In addition to the workflow design**, we propose a GRPO-based method that directly encourages improvement across iterations. This reward formulation is tailored for agentic refinement and significantly contributes to performance, as demonstrated in our ablation studies (Table 3). Finally, our experiments demonstrate that both the workflow design and the reward computation method are essential for achieving the results reported. Removing the workflow components or replacing the reward leads to substantial drops in performance. These findings indicate that the contribution of our system is not limited to a minor scaffolding mechanism, but lies in a coherent workflow that enables capabilities beyond those of existing approaches.
>
> [1] Yang, John, et al. "Swe-agent: Agent-computer interfaces enable automated software engineering." NeurIPS 2024.
>
> [2] Venkatesh, Kavana, Connor Dunlop, and Pinar Yanardag. "Crea: A collaborative multi-agent framework for creative content generation with diffusion models." NeurIPS 2025.
>
> [3] Wang, Junlin, et al. "Mixture-of-agents enhances large language model capabilities." ICLR 2025
>
> [4] Wang, Xingyao, et al. "Openhands: An open platform for ai software developers as generalist agents." ICLR 2025.

---

> > ### Comment · Reviewer_z3Vc · 2025-11-24
> > **I disagree**
> >
> > Thank you for your reply, and re-situating your contributions
> > This might be one of these subjective areas where we will have to disagree. I do think that proposing only a Agentic scaffolding for a rather new task is not a substantive contribution to the research community. From the examples you provided, two tackle multi-agent environments (a field which has a long and rich history... although some of the recent contributions are rather limiting as well from what I have seen), while the others describe scaffolding which have seen huge community uptake with which it is hard to argue (and well, targeting SWEBench used to be a good path for impact anyway, at least last year)
> >
> > Now, I do acknowledge that your paper goes beyond that. This comment was just focusing on this one contribution, which is highlighted as one of the 3 by the paper (the third one being "extensive experiments")

---

> ### Author Response · Authors · 2025-11-18
> **Response to Official Review by Reviewer z3Vc (2/4)**
>
> **Q2.** Only one benchmark is used, and therefore it is not clear if the scaffolding or the general proposal generalizes to other datasets
>
> **A2.** Thank you for your insightful question. Our WebGen-Agent aims to assist non-expert users to build **complicated, interactive websites** by creating multi-file codebases based on **natural language website-development instructions**. Our problem setting requires an agentic workflow, high-level planning, organizing complex multi-file codebases, and implementing nuanced user requirements in both functionality and design, while most of the other benchmarks such as Web2Code [1] and Design2Code [2] represent the task of taking in a **webpage screenshot** and generating its corresponding HTML file, which can be achieved by an MLLM itself and is better suited to testing basic MLLM abilities.
>
> To demonstrate the generalizability of our method, we test WebGen-Agent on the Web Application subset of the ArtifactsBench [3] dataset, which also requires generating a website based on user instructions, though their requirements are less complicated than the instructions in WebGen-Bench. The baseline results of the closed-source models and DeepSeek-V3 are taken from [3], while the others are tested using their open-sourced codebase. As shown in the table below, WebGen-Agent with Qwen3-Coder-480B-A35B-Instruct as the coding LLM achieves the best performance of 63.86, outperforming the baseline results of various closed-source models and the baseline result of Qwen3-Coder-480B-A35B-Instruct. WebGen-Agent with DeepSeek-V3 also outperforms the baseline result of DeepSeek-V3. This demonstrates the effectiveness and generalizability of WebGen-Agent. We have added this to Appendix M of the revised paper.
>
> | Test Name                                                | Average MLLM score |
> |----------------------------------------------------------|--------------------|
> | GPT-4o                                                   | 39.27 |
> | o1                                                       | 39.01 |
> | Claude 3.7-Sonnet                                        | 53.64 |
> | Claude 4.0-Sonnet                                        | 55.79 |
> | Gemini-2.5-Pro                                           | 58.30 |
> | DeepSeek-V3                                              | 47.47 |
> | **DeepSeek-V3 + WebGen-Agent**                               | 55.74 |
> | Qwen3-Coder-480B-A35B-Instruct                           | 55.42 |
> | **Qwen3-Coder-480B-A35B-Instruct + WebGen-Agent**            | **63.86** |
>
> [1] Yun, Sukmin, et al. "Web2code: A large-scale webpage-to-code dataset and evaluation framework for multimodal llms." Advances in neural information processing systems 37 (2024): 112134-112157.
>
> [2] Si, Chenglei, et al. "Design2code: Benchmarking multimodal code generation for automated front-end engineering." Proceedings of the 2025 Conference of the Nations of the Americas Chapter of the Association for Computational Linguistics: Human Language Technologies (Volume 1: Long Papers). 2025.
>
> [3] Zhang, Chenchen, et al. "Artifactsbench: Bridging the visual-interactive gap in llm code generation evaluation." arXiv preprint arXiv:2507.04952 (2025).

---

> > ### Comment · Reviewer_z3Vc · 2025-11-24
> > **kudos for extra experiments**
> >
> > Thanks for those additional experiments, this is nice to see, quite reassuring and gives me enough confidence to raise my rating

---

> ### Author Response · Authors · 2025-11-18
> **Response to Official Review by Reviewer z3Vc (3/4)**
>
> **Q3.** The final numbers of the 7B models are impressive, however most of the gains are obtained from the SFT step which consists in data-distilling Deepseek. The use of a large VLM model as judge also raises the question if the gains are not "just" distillation from a larger model, or if there is some learning that could generalize to larger models.
>
> **A3.** Thank you for the insightful comment. However, we would like the clarify that the performance of our *Step-GRPO with Screenshot and GUI-agent Feedback* training method cannot be achieved by just distilling from a larger model. As shown in Table 4 and 5 in the paper, as well as the table below, *Step-GRPO with Screenshot and GUI-agent Feedback* significantly improves the performance of Qwen2.5-Coder-7B-Instruct (SFT for 1 Epoch), increasing the accuracy by 6.5% and the appearance score by 0.3. **In contrast**, extending the SFT training to 2 epochs only yields an improvement in accuracy of 0.4% and no improvement in appearance score at all. This demonstrates the effectiveness and necessity of our *Step-GRPO with Screenshot and GUI-agent Feedback* approach.
>
> As for the concern that *“the use of a large VLM model as judge raises the question if the gains are not ‘just’ distillation from a larger models”*, we would like to point out that as shown in Table 4 in the paper and the last line of the table below, Qwen2.5-VL-32B-Instruct performs poorly as the coding LLM on WebGen-Bench, producing an accuracy of only 5.6% and an appearance score of only 1.3, which is **much lower than even the original Qwen2.5-Coder-7B-Instruct model**. This is due to the fact that while Qwen2.5-VL-32B-Instruct excels at providing visual feedback, its coding ability is much weaker than the LLMs that specialize in code generation. Therefore, the gains in *Step-GRPO with Screenshot and GUI-agent Feedback* does not come from distillation from a larger model, as Qwen2.5-VL-32B-Instruct is not stronger than Qwen2.5-Coder-7B-Instruct in code generation. This also strongly suggests that our method can generalize to larger models, as a strong coding LLMs can **benefit from the visual supervision** provided by a VLM with strong visual abilities and **little coding skills**.
>
> | Test Name                                                                      | Yes | Partial | No Start | Failed | Accuracy      | Appearance Score |
> |--------------------------------------------------------------------------------|----:|--------:|---------:|-------:|--------------:|-----------------:|
> | Qwen2.5-Coder-7B-Instruct                                                      | 10   | 4.8     | 60.9     | 24.3  | 12.4          | 1.6              |
> | Qwen2.5-Coder-7B-Instruct (SFT for 1 Epoch, **starting point of RL**)             | 33.8 | 10.2    | 56       | 0     | 38.9          | 3.4              |
> | Qwen2.5-Coder-7B-Instruct (SFT for 2 Epochs)                                   | 32.1 | 14.2    | 53.5     | 0.2   | 39.3 (+0.4)          | 3.4 (+0.0)             |
> | **Qwen2.5-Coder-7B-Instruct (Step-GRPO w/ Screenshot + GUI-agent)**                | 40.2 | 10.5    | 49.3     | 0     | 45.4 (+6.5)   | 3.7 (+0.3)       |
> | Qwen2.5-VL-32B-Instruct                                                        | 4.5  | 2.2     | 78.8     | 14.5  | 5.6           | 1.3              |

---

> > ### Comment · Reviewer_z3Vc · 2025-11-24
> > **2 epochs != more distillation**
> >
> > I am not sure what the experiment with 2 epochs is telling us, beyond that we see overfitting. This does not contradict my statement that `most of the gains are obtained from the SFT step which consists in data-distilling Deepseek`. It is still not clear to me if the same gains could be obtained when applied to a larger model (which would give more credit to your RL set-up). If that is unfeasible to run computational-wise, the next best thing would be run the RL step directly on Qwen2.5-Coder-7B-Instruct

---

> ### Author Response · Authors · 2025-11-18
> **Response to Official Review by Reviewer z3Vc (4/4)**
>
> **Q4.** From Table 4 it seems that the VLM used as a judge does not necessarily have a major impact. Did you benchmark this behaviour when finetuning the smaller models with GRPO?
>
> **A4.** Thank you for your valuable question. To show the impact of VLM as a judge, we replace Qwen2.5-VL-32B-Instruct with a much weaker model, Qwen2.5-VL-7B-Instruct, in GRPO training. As shown in Table A below, using **a weaker VLM greatly reduces** the effectiveness of the GRPO training, as it introduces noise and inaccuracies in the RL supervision signal. Using Qwen2.5-VL-7B-Instruct as the feedback VLM provides almost no gain in the GRPO training process, demonstrating the major role the VLM plays in our WebGen-Agent pipeline and Step-GRPO with Screenshot and GUI-agent Feedback training approach.
>
> As for Table 4, the reason that using GPT-4o does not make a large difference compared to Qwen2.5-VL-32B-Instruct is that Qwen2.5-VL-32B-Instruct can already provide highly accurate visual feedback. **There is little gap between Qwen2.5-VL-32B-Instruct and GPT-4o in providing appearance and functional judgements.** To demonstrate this, we test Qwen2.5-VL-32B-Instruct on two VLM benchmarks, PairBench [1] and AgentRewardbench [2]. PairBench tests whether VLMs are reliable at comparing what they see, while AgentRewardBench tests VLMs’ ability to judge the success of GUI agent trajectories. The two benchmarks can accurately reflect the ability of a VLM to provide the screenshot and GUI agent feedback.
>
> As shown in Table B,  Qwen2.5-VL-32B-Instruct outperforms closed-source models such as GPT-4o and Gemini-1.5-Pro on three out of four metrics. Notably, on MMScore, which is the main metric in the PairBench paper [1], Qwen2.5-VL-32B-Instruct outperforms GPT-4o by 20.03%. This shows that Qwen2.5-VL-32B-Instruct is good at comparing similar images, making it a suitable choice for providing screenshot feedback and appearance scores in the WebGen-Agent pipeline.
>
> Additionally, as demonstrated in Table C, Qwen2.5-VL-32B-Instruct outperforms GPT-4o, GPT-4o-mini, and Qwen2.5-VL-72B-Instruct in Overall Precision, which is the main metric used in [2] to judge a model’s ability to predict the success of web agent trajectories. Its performance is only 1.1% lower than Claude-3.7-Sonnet, while being much more cost-effective than the closed-source models. On the five data splits, Qwen2.5-VL-32B-Instruct achieves the highest performance on VWA (VisualWebArena) and Wk++ (WorkArena++), and is only lower than the highest precision on Work (WorkArena) by 0.5%. This demonstrates that Qwen2.5-VL-32B-Instruct is a reliable judge of the GUI agent trajectories and can provide accurate functional feedback for our WebGen-Agent pipeline. Together, Table B and Table C explain why using GPT-4o does not make a large difference compared to Qwen2.5-VL-32B-Instruct in Table 4, and that Table 4 does not indicate that “VLM used as a judge does not necessarily have a major impact”. We have added these new results and their analysis to Section 3.2 and Appendix F of the revised paper.
>
> Table A:
> | Training Method | Feedback VLM| Yes | Partial | No   | Start Failed | Accuracy | Appearance Score |
> |---|--|-|-|--|-|-|--|
> | SFT for 1 Epoch (RL starting point)| –  | 33.8 | 10.2 | 56   | 0 | 38.9 | 3.4 |
> | Step-GRPO w/ Screenshot + GUI-agent| Qwen2.5-VL-32B-Instruct| 40.2 | 10.5| 49.3 | 0  | 45.4 (+6.5) | 3.7 (+0.3)  |
> | Step-GRPO w/ Screenshot + GUI-agent | Qwen2.5-VL-7B-Instruct| 33.7 | 10.7 | 54.7 | 0.9| 39 (+0.1) | 3.3 (-0.1) |
>
> Table B: PairBench
> | Model | MMScore (%) | 1-RelaxSym (%) | Smoothness | Controllability (%) |
> |-|-|-|-|-|
> | GPT-4o-mini | 48.28 | 89.07 | 1.59 | 85.48|
> | GPT-4o | 53.95 | 91.53| 1.54 | 72.77 |
> | Gemini-1.5-Flash| 56.55 | **93.19**  | 1.34 | 74.54|
> | Gemini-1.5-Pro| 52.60 | 88.72 | 1.17| 88.09 |
> | Qwen2.5-VL-32B-Instruct  | **73.98**| 91.30| **2.87**| **93.37**|
>
> Table C: AgentRewardBench
> | Model | Overall Precision | Overall Recall | Overall F1 | AB Precision | VWA Precision | WA Precision | Work Precision | Wk++ Precision |
> |----|-|--|-|-|-|--|-|-|
> | Claude-3.7-Sonnet| **69.4**| 76.3| 72.7| 71.4         | 64.8          | 69.3         | 85.3           | **66.7**           |
> | GPT-4o| 68.1| 80.3| **73.7**| 77.8| 60.7| **69.9**         | **93.8**           | 59.6           |
> | GPT-4o-mini | 64.5| 78.3| 70.8| **80.0**| 57.4          | 66.9         | 90.3           | 54.8           |
> | Qwen2.5-VL-72B-Instruct| 64.5| **86.1**| 73.7       | 70.0         | 58.5          | 62.9         | **93.8**           | 64.4           |
> | Qwen2.5-VL-32B-Instruct| 68.3| 79.7| 73.5       | 70.0         | **68.2**          | 61.8         | 93.3           | **66.7**           |
>
> [1] Feizi, Aarash, et al. "Pairbench: A systematic framework for selecting reliable judge vlms." arXiv preprint arXiv:2502.15210 (2025).
>
> [2] Lù, Xing Han, et al. "Agentrewardbench: Evaluating automatic evaluations of web agent trajectories." arXiv preprint arXiv:2504.08942 (2025).

---

> > ### Comment · Reviewer_z3Vc · 2025-11-24
> > **interesting general point**
> >
> > It seems to me that this experiment is making my point. If a 32B judge model gives you better result on a 7B policy, than using a 7B judge model then what the pipeline seems to be doing is to knowledge-distill the info from the 32B model
> >
> > There seems to be however a more general argument here, which is that your set-up allows you to combine a strong code model (policy) with a strong VLM (judge model) _without the need of pretraining for this specifically_ . This could well be the main contribution of this work IMO (would require more experiments, and a dedicated narrative): right now it is somehow hidden and became apparent to me only during the rebuttal. Anyway, this is partly responsible for my increase in score

---

> ### Author Response · Authors · 2025-11-25
> **Thank you for acknowledging our rebuttal efforts**
>
> Thank you very much for acknowledging our rebuttal efforts and for taking the time to provide further insightful comments and suggestions. We would like to address your concerns below:
>
> **1. Contribution.**
>
> Thank you for acknowledging our contributions. In our paper, we innovatively introduced WebGen-Agent, a code agent that leverages visual feedback from screenshots and GUI agent trajectories to iteratively improve the generated codebase, greatly improving the ability of various LLMs to generate complex, interactive websites from scratch. Additionally, we introduce *Step-GRPO with Screenshot and GUI-agent Feedback*, an RL training method that leverages the appearance and functionality scores inherently produced in the WebGen-Agent pipeline, greatly improving the accuracy of the coding LLM. While we agree that a single agentic scaffolding might seem weak for a conference paper, our contributions go beyond that, providing a unified framework of agentic inference and RL training with accurate visual feedback.
>
> **2. Most of the gains are obtained from the SFT step, which consists of data-distilling Deepseek.**
>
> Thank you for your insightful comment and valuable suggestion. The result of SFT training for 2 epochs is to demonstrate that continued SFT training does not bring significant improvement compared to training for only 1 epoch. This is a necessary experiment, as most of the previous SFT distillation works train for 2 to 3 epochs ([1], [2]).
>
> However, we do realize that it might not be completely fair, as it only scales computation and does not represent the result of increasing the amount of training data. Therefore, we have generated 700 new trajectories from instructions in WebGen-Instruct with DeepSeek-V3. Combining them with the original 700 trajectories, we created a scaled training set of approximately 1400 trajectories. As shown in the table below, increasing the SFT data to twice the original size does not significantly improve the accuracy and appearance score of Qwen2.5-Coder-7B-Instruct. This demonstrates that simple distillation has a clear upper bound, and could not achieve the same gain in accuracy and appearance score as our *Step-GRPO with Screenshot and GUI-agent Feedback* training method. Therefore, our proposed RL training approach is a necessary stage that brings the performance of the model already trained with SFT to a higher level. We have added the new result to Table 3 and the analysis to Section 3.2 of the revised paper. The scaled training data has been added to `WebGen-RL/data/webgen-agent_train_1432.jsonl` in the supplementary material for reproducibility.
>
> | Test Name | Yes | Partial |  No  | Start Failed | Accuracy | Appearance Score |
> |---|-|-|-|-|-|--|
> | Qwen2.5-Coder-7B-Instruct| 10  | 4.8     | 60.9 | 24.3         | 12.4     | 1.6 |
> | Qwen2.5-Coder-7B-Instruct (SFT with 700 samples for 1 epoch)  | 33.8| 10.2    | 56.0   | 0.0 | 38.9 | 3.4  |
> | Qwen2.5-Coder-7B-Instruct (SFT with 1400 samples for 1 epoch) | 33.5| 11.6    | 54.9 | 0.0 | 39.3 (+0.4)| 3.4 (+0.0) |
> | **Qwen2.5-Coder-7B-Instruct (Step-GRPO w/ Screenshot + GUI-agent)**  | 40.2 | 10.5| 49.3 | 0.0| 45.4 (+6.5)   | 3.7 (+0.3) |
>
> Regarding the suggestion of running the RL step directly on the model without the SFT stage, we agree that it is an interesting and insightful experiment. Currently, the experiment is still running, and we are expecting to obtain the result before the end of the rebuttal period.
>
> [1] Yu, Longhui, et al. "Metamath: Bootstrap your own mathematical questions for large language models." arXiv preprint arXiv:2309.12284 (2023).
>
> [2] Yang, John, et al. "Swe-smith: Scaling data for software engineering agents." arXiv preprint arXiv:2504.21798 (2025).
>
> **3. Combining a strong code model (policy) with a strong VLM (judge model) without the need of pretraining for this specifically.**
>
> Thank you for the insightful comment. Indeed, our *Step-GRPO with Screenshot and GUI-agent Feedback* training method leverages the accurate step-level screenshot and GUI-agent testing scores generated by the VLM to improve the coding ability of the strong code model. This is a non-trivial process, transforming information obtained from the VLM’s visual ability into the coding LLM’s programming skills. We have added this clarification to the Introduction section in the revised paper.
>
> Thank you once again for the time and effort you have dedicated to reviewing our work. We are grateful for all your insightful comments and suggestions, which have been invaluable in improving our project.

---

> > ### Author Response · Authors · 2025-11-28
> > **Follow-up results of direct RL training without SFT**
> >
> > We have finished running the experiment of directly training the original model (without special SFT) using our *Step-GRPO with Screenshot and GUI-agent Feedback* training approach. We performed RL training directly on Qwen3-8B, and the results are presented below. As shown in the table below, applying *Step-GRPO with Screenshot and GUI-agent Feedback* directly on Qwen3-8B achieves a rise in accuracy of 6.5% and a rise in appearance score of 0.3. This improvement in performance is even more pronounced than training on the SFT model. This demonstrates that **the distillation of DeepSeek-V3 is not absolutely necessary**, and that the same gains could likely be obtained when applied to a larger model. We have added the new results to the revised paper in Appendix O in the revised paper.
> >
> > | Test Name                         | Yes | Partial | No   | Start Failed | Accuracy    | Appearance Score |
> > |-----------------------------------|-----|---------|------|--------------|-------------|------------------|
> > | Qwen3-8B                          | 29.5 | 9.1     | 61.4 | 0            | 34.1        | 3.2              |
> > | Qwen3-8B + Step-GRPO              | 34.2 | 12.7    | 52.4 | 0.8          | 40.5 (+6.4) | 3.5 (+0.3)       |
> > | Qwen3-8B + SFT                    | 32.8 | 11.6    | 55.6 | 0            | 38.6        | 3.4              |
> > | Qwen3-8B + SFT + Step-GRPO        | 37.4 | 12.1    | 50.5 | 0            | 43.4 (+4.8) | 3.6 (+0.2)       |

---

### Official Review · Reviewer_mw4M · 2025-10-31

**Soundness:** 3
**Presentation:** 3
**Contribution:** 3
**Rating:** 4
**Confidence:** 4

**Summary:**

This paper presents WebGen-Agent, a test-time inference framework that leverages screenshots and GUI-agent feedback signals to iteratively improve the appearance and performance of web generation from scratch. The authors demonstrate that using WebGen-Agent leads to substantial gains for both proprietary and open-source models compared to prior frameworks such as Bolt.diy. They also introduce Step-GRPO, which employs rewards derived from feedback based on screenshots and GUI-agent evaluations. The models Qwen2.5-Coder-7B-Instruct and Qwen3-8B are trained with SFT and Step-GRPO, achieving notable improvements over their base models.

**Strengths:**

- Strong ablation studies demonstrating the contribution of each component, including different rewards, backtracking, and the impact of SFT.

- Significant improvements shown with both WebGen-Agent and step-GRPO.

- Excellent reproducibility, with detailed appendices that include all prompts, sample outputs, and qualitative examples.

**Weaknesses:**

- The source of instructions for the 700 trajectories used in SFT training (generated with DeepSeek-V3 using WebGen-Agent) is not specified. The authors should clarify whether the data originate from WebGen-Instruct or WebGen-Bench. This is a critical point not clarified in the paper to ensure there is no data leakage.

- The paper does not provide a cost or time comparison between WebGen-Agent and Bolt.diy. For proprietary models, it would be helpful to include runtime or cost data, while for open-source models, a FLOPs or time comparison would clarify efficiency.

- The results do not include performance for WebGen-LM-3B, 7B, and 32B with WebGen-Agent. Including these models would provide a more comprehensive comparison, especially since WebGenAgent-7B and WebGen-LM-7B share the same base model (Qwen2.5-Coder-Instruct-7B).

- It would be beneficial to determine which model acts as the most reliable judge or produces the most aligned outputs. Benchmarks such as PairBench [1] or AgentRewardBench [2] could help evaluate this aspect.

---
Minor Comments

- The formula fonts in Figure 2 should be larger for improved readability.

[1] PairBench: Are Vision-Language Models Reliable at Comparing What They See?

[2] AgentRewardBench: Evaluating Automatic Evaluations of Web Agent Trajectories

**Questions:**

Please address the issues raised in weaknesses section.

---

> ### Author Response · Authors · 2025-11-18
> **Response to Official Review by Reviewer mw4M (1/3)**
>
> Thank you for the time and effort you have dedicated to reviewing our paper. Your comments have been invaluable in improving our work.
>
> **Q1.** The source of instructions for the 700 trajectories used in SFT training (generated with DeepSeek-V3 using WebGen-Agent) is not specified. The authors should clarify whether the data originate from WebGen-Instruct or WebGen-Bench. This is a critical point that is not clarified in the paper to ensure there is no data leakage.
>
> **A1.** Thank you for your valuable question. The 700 trajectories used in SFT training are generated from website generation instructions **taken from WebGen-Instruct**, which is introduced in the WebGen-Bench [1] paper. As explained in [1], meticulous decontamination steps were taken in the construction of WebGen-Instruct. They first employ a 5-gram Jaccard similarity, removing instructions from the training set with a similarity score higher than 0.6 to one of the instructions in the testing set. Then, to remove instructions that are semantically similar to those in the testing set, they compute the sentence embeddings of the instructions using the all-MiniLM-L6-v2 model of Sentence-Transformer [2], removing the training instructions with a cosine similarity greater than 0.55.
>
> To further ensure there is no data leakage, we also apply the n-gram testing, as demonstrated in the table below. The overlap percentages for various n-grams are quite low, and the overlap becomes 0.00% when n-grams are 13. We have added this analysis to Appendix L and clarified the source of the instructions used in SFT training in the revised paper.
>
> | n-grams | 7 | 9 | 11 | 13 |
> |---------|---|---|----|----|
> | Overlap Ratio (%) | 5.59 % | 1.47 % | 0.22 % | 0.00 % |
>
> [1] Lu, Zimu, et al. "WebGen-Bench: Evaluating LLMs on Generating Interactive and Functional Websites from Scratch." NeurIPS 2025.
>
> [2] Reimers, Nils, and Iryna Gurevych. "Sentence-bert: Sentence embeddings using siamese bert-networks." EMNLP 2019.
>
> **Q2.** The paper does not provide a cost or time comparison between WebGen-Agent and Bolt.diy. For proprietary models, it would be helpful to include runtime or cost data, while for open-source models, a FLOPs or time comparison would clarify efficiency.
>
> **A2.** Thank you for your insightful suggestion. We present the average runtime, average cost, and average FLOPs in the table below, as well as the accuracy and appearance scores. As demonstrated in the table, the average runtimes of the WebGen-Agent pipeline are all between 10 to 20 minutes and the average costs are all between 0.01 and 0.3 dollars, which is **within the acceptable range for developing a multi-file codebase of an interactive website from scratch**. The time used for Claude-3.5-Sonnet and DeepSeek-V3 is around 13 minutes, as they have strong coding abilities and make fewer mistakes, thus requiring fewer iterations. DeepSeek-R1’s average runtime is longer as it is a reasoning model and needs to generate longer thoughts. Qwen2.5-Coder-32B-Instruct and Qwen2.5-72B-Instruct are weaker models and need more iterations to correct errors and improve the websites.
>
> The average runtimes and average costs of WebGen-Agent are notably higher than those of Bolt.diy, but WebGen-Agent significantly improves the accuracy and appearance scores compared to Bolt.diy, bringing around 20% to 30% rise in accuracy and around 1.0 to 2.0 rise in appearance scores. Therefore, we believe that **this significant increase in performance justifies the increased cost and runtime**. We have added the results to Appendix N in the revised paper.
>
> | Test Name| Average Runtime (min) | Average Cost ($) | Average FLOPs (TFLOPs) | Accuracy | Appearance Score |
> |-|-|-|-|-|-|
> | Claude-3.5-Sonnet + Bolt.diy| 2.1| 0.0677| N/A | 26.4     | 3.0 |
> | Claude-3.5-Sonnet + WebGen-Agent | 13.1                      | 0.2009                | N/A                         | 51.9 (+25.5) | 3.9 (+0.9)          |
> | DeepSeek-R1 + Bolt.diy| 6.3                       | 0.0025                | $1.55 \times 10^{16}$     | 27.8     | 2.50                  |
> | DeepSeek-R1 + WebGen-Agent| 17.5                      | 0.0057                | $3.54 \times 10^{16}$     | 46.4 (+18.6) | 3.8 (+1.3)          |
> | DeepSeek-V3 + Bolt.diy | 3.2                       | 0.0088                | $3.18 \times 10^{16}$     | 20.8     | 2.00                  |
> | DeepSeek-V3 + WebGen-Agent| 13.3                      | 0.0131                | $2.33 \times 10^{16}$     | 52.6 (+31.8) | 3.8 (+1.8)          |
> | Qwen2.5-Coder-32B-Inst. + Bolt.diy| 1.4| N/A  | $8.18 \times 10^{14}$     | 9.5      | 1.10                  |
> | Qwen2.5-Coder-32B-Inst. + WebGen-Agent| 16.7| N/A                   | $2.47 \times 10^{15}$     | 32.0 (+22.5) | 3.3 (+2.2)          |
> | Qwen2.5-72B-Inst. + Bolt.diy| 2.9| N/A| $1.90 \times 10^{15}$     | 13.8     | 1.40                  |
> | Qwen2.5-72B-Inst. + WebGen-Agent| 19.0| N/A| $6.54 \times 10^{15}$     | 35.9 (+22.1) | 3.4 (+2.0) |

---

> ### Author Response · Authors · 2025-11-18
> **Response to Official Review by Reviewer mw4M (2/3)**
>
> **Q3.** The results do not include performance for WebGen-LM-7B, 14B, and 32B with WebGen-Agent. Including these models would provide a more comprehensive comparison, especially since WebGenAgent-7B and WebGen-LM-7B share the same base model (Qwen2.5-Coder-Instruct-7B).
>
> **A3.** Thank you for your valuable suggestion. We test the performance of WebGen-LM-7B, 14B, and 32B with WebGen-Agent and present the results in Table A and Table B below. The highest accuracy and appearance scores at each model size (7B, 14B, and 32B) are marked in **bold**. As shown in Table A, when using the WebGen-LM models, our WebGen-Agent pipeline consistently outperforms the previous SOTA, Bolt.diy, across all three model sizes.
>
> Additionally, when we compare WebGenAgent-7B with WebGen-LM-7B, which shares the same base model, our WebGenAgent-7B model outperforms WebGen-LM-7B by 12.6% and 0.4 in accuracy and appearance score, respectively, under the WebGen-Agent pipeline. This demonstrates the effectiveness of our *Step-GRPO with Screenshot and GUI-agent Feedback* training method, which leverages step-level appearance and functional scores to provide accurate supervision for the reinforcement-learning process. We have added these new results and their analysis to Table 1, Table 10, and Section 3.1 in the revised paper.
>
> Table A:
> | Test Name| Yes  | Partial | No   | Start Failed | Accuracy | Appearance Score |
> |-|-|-|-|-|-|-|
> | WebGen-LM-32B + Bolt.diy| 34.2 | 8.0| 57.8 | 0.0| 38.2     | 2.8 |
> | WebGen-LM-32B + WebGen-Agent | 35.4 | 9.9     | 54.6 | 0.2          | **40.3**     | **3.6**              |
> | WebGen-LM-14B + Bolt.diy| 25.0 | 8.7     | 66.3 | 0.0          | 29.4     | 2.5              |
> | WebGen-LM-14B + WebGen-Agent| 32.9 | 10.2    | 38.9 | 17.9         | **38.0**     | **3.4**              |
> | WebGen-LM-7B + Bolt.diy| 24.9 | 7.1     | 68.0 | 0.0          | 28.4     | 2.5              |
> | WebGen-LM-7B + WebGen-Agent| 27.8 | 9.9     | 51.2 | 11.1         | 32.8     | 3.3              |
> | WebGenAgent-LM-7B-Step-GRPO + WebGen-Agent | 40.2 | 10.5    | 49.3 | 0.0          | **45.4** (+12.6)     | **3.7** (+0.4)              |
> | WebGenAgent-LM-8B-Step-GRPO + WebGen-Agent| 37.4 | 12.1    | 50.5 | 0.0          | 43.4     | 3.6              |
>
> Table B:
> | Test Name| Content Presentation | User Interaction | Data Management | Functional Testing | Data Display Testing | Design Validation Testing |
> |-|-|-|-|-|-|-|
> | WebGen-LM-32B + Bolt.diy                            | 46.6 | 33.2 | 38.8 | **29.1** | 43.0 | 56.1 |
> | WebGen-LM-32B + WebGen-Agent                        | **47.1** | **37.2** | **39.1** | 24.6 | **45.7** | **75.8** |
> | WebGen-LM-14B + Bolt.diy                            | 30.2 | 27.8 | 31.6 | 23.6 | 26.9 | 49.2 |
> | WebGen-LM-14B + WebGen-Agent                        | **44.3** | **31.3** | **44.4** | **28.6** | **40.1** | **61.1** |
> | WebGen-LM-7B + Bolt.diy                             | 27.9 | 23.8 | 38.1 | 22.0 | 27.7 | 47.5 |
> | WebGen-LM-7B + WebGen-Agent                         | 33.9 | 27.5 | 41.9 | 22.0 | 34.7 | 59.8 |
> | WebGenAgent-LM-7B-Step-GRPO + WebGen-Agent          | **51.1** | **41.1** | **47.8** | **30.7** | **56.7** | **69.3** |
> | WebGenAgent-LM-8B-Step-GRPO + WebGen-Agen           | 52.0 | 38.8 | 43.1 | 30.2 | 51.1 | 68.4 |

---

> ### Author Response · Authors · 2025-11-18
> **Response to Official Review by Reviewer mw4M (3/3)**
>
> **Q4.** It would be beneficial to determine which model acts as the most reliable judge or produces the most aligned outputs. Benchmarks such as PairBench [1] or AgentRewardBench [2] could help evaluate this aspect.
>
> **A4.** Thank you for the insightful suggestion. We test the performance of our chosen VLM, Qwen2.5-VL-32B-Instruct, on PairBench and AgentRewardBench, and present the results in Table A and Table B below. Results of the closed-source models are taken from [1] and [2]. Table A shows the aggregated MMScore, 1-RelaxSym, Smoothness, and Controllability over all four data splits in PairBench [1]. Table B shows the VLM judge performance for predicting success, mainly measured with precision [2].
>
> As shown in Table A,  Qwen2.5-VL-32B-Instruct outperforms closed-source models such as GPT-4o and Gemini-1.5-Pro on three out of four metrics. Notably, on MMScore, which is the main metric in the PairBench paper [1], Qwen2.5-VL-32B-Instruct outperforms GPT-4o by 20.03%. This shows that Qwen2.5-VL-32B-Instruct is good at comparing similar images, making it a suitable choice for providing screenshot feedback and appearance scores in the WebGen-Agent pipeline.
>
> Additionally, as demonstrated in Table B, Qwen2.5-VL-32B-Instruct outperforms GPT-4o, GPT-4o-mini, and Qwen2.5-VL-72B-Instruct in Overall Precision, which is the main metric used in [2] to judge a model’s ability to predict the success of web agent trajectories. Its performance is only 1.1% lower than Claude-3.7-Sonnet, while being much more cost-effective than the closed-source models. On the five data splits, Qwen2.5-VL-32B-Instruct achieves the highest performance on VWA (VisualWebArena) and Wk++ (WorkArena++), and is only lower than the highest precision on Work (WorkArena) by 0.5%. This demonstrates that Qwen2.5-VL-32B-Instruct is a reliable judge of the GUI agent trajectories and can provide accurate functional feedback for our WebGen-Agent pipeline. We have added these new results and their analysis to Appendix F of the revised paper.
>
> Table A: PairBench
> | Model                         | MMScore (%) | 1-RelaxSym (%) | Smoothness | Controllability (%) |
> |-------------------------------|-------------|----------------|------------|---------------------|
> | GPT-4o-mini                   | 48.28       | 89.07          | 1.59       | 85.48               |
> | GPT-4o                        | 53.95       | 91.53          | 1.54       | 72.77               |
> | Gemini-1.5-Flash              | 56.55       | **93.19**          | 1.34       | 74.54               |
> | Gemini-1.5-Pro                | 52.60       | 88.72          | 1.17       | 88.09               |
> | Qwen2.5-VL-32B-Instruct       | **73.98**       | 91.30          | **2.87**       | **93.37**               |
>
> Table B: AgentRewardBench
> | Model                          | Overall Precision | Overall Recall | Overall F1 | AB Precision | VWA Precision | WA Precision | Work Precision | Wk++ Precision |
> |--------------------------------|-------------------|----------------|------------|--------------|---------------|--------------|----------------|----------------|
> | Claude-3.7-Sonnet              | **69.4**              | 76.3           | 72.7       | 71.4         | 64.8          | 69.3         | 85.3           | **66.7**           |
> | GPT-4o                          | 68.1              | 80.3           | **73.7**       | 77.8         | 60.7          | **69.9**         | **93.8**           | 59.6           |
> | GPT-4o-mini                     | 64.5              | 78.3           | 70.8       | **80.0**         | 57.4          | 66.9         | 90.3           | 54.8           |
> | Qwen2.5-VL-72B-Instruct         | 64.5              | **86.1**           | 73.7       | 70.0         | 58.5          | 62.9         | **93.8**           | 64.4           |
> | Qwen2.5-VL-32B-Instruct         | 68.3              | 79.7           | 73.5       | 70.0         | **68.2**          | 61.8         | 93.3           | **66.7**           |
>
> [1] Feizi, Aarash, et al. "Pairbench: A systematic framework for selecting reliable judge vlms." arXiv preprint arXiv:2502.15210 (2025).
>
> [2] Lù, Xing Han, et al. "Agentrewardbench: Evaluating automatic evaluations of web agent trajectories." arXiv preprint arXiv:2504.08942 (2025).
>
> **Q5.** The formula fonts in Figure 2 should be larger for improved readability.
>
> **A5.** Thank you for your valuable suggestion. We have increased the formula fonts in Figure 2 in the revised paper.

---

### Author Response · Authors · 2025-11-19
**Paper Update Summary**

Thanks to all the chairs and reviewers for their constructive suggestions and comments. We summarize the revised paper details.

1. Analysis of potential data leakage is added to Appendix L,  and clarification of the source of the SFT instructions is added to the third paragraph of Section 3.1. (**Reviewer mw4M**)
2. Cost and time comparison between WebGen-Agent and Bolt.diy is added to Appendix N. (**Reviewer mw4M**)
3. Performance for WebGen-LM-7B, 14B, and 32B with WebGen-Agent is added to Table 1, Table 10, and the fourth paragraph of Section 3.1. (**Reviewer mw4M**)
4. Comparison of the ability to judge the visual output between various VLMs is added to Appendix F. (**Reviewer mw4M**, **Reviewer z3Vc**)
5. The performance of WebGen-Agent on the Web Application subset of the ArtifactsBench dataset is added to Appendix M. (**Reviewer z3Vc**, **Reviewer jD87**)
6. The result of replacing Qwen2.5-VL-32B-Instruct with Qwen2.5-VL-7B-Instruct in the GRPO training is added to Table 5 in Section 3.2. (**Reviewer z3Vc**)
7. Clarification of the single-page screenshot design choice is added to Section 2.1. (**Reviewer jD87**)
8. Clarification of the connection with the performance of developing GUI agents is added to Appendix A. (**Reviewer jD87**)
9. Human annotation details are added to Appendix E and Appendix G. (**Reviewer jD87**, **Reviewer Cbyh**)
10. Clarification of the source of the OpenHands baseline results is added to Table 1. (**Reviewer jD87**)
11. Definition of Accuracy is added to Table 1, Table 2, and Table 3. (**Reviewer jD87**)
12. Typos of wrong inline citation format in L235, incorrect phrasing in L134, and subject-verb agreement error in L1044 are corrected. (**Reviewer jD87**)
13. Relation between WebGen-Agent and previous works is added to Section 4. (**Reviewer Cbyh**)
14. LLM backbone setting is added to Table 2, Table 11, and Figure 17. (**Reviewer Cbyh**)
15. Clarification of the relationship between the coding LLM and the VLM in our training approach is added to Section 1. (**Reviewer z3Vc**)
16. Result of direct RL training without SFT is added to Appendix O. (**Reviewer z3Vc**)

---

### Author Response · Authors · 2025-11-29
**Summary of the discussion period**

We would like to express our gratitude to AC, PC, and all the reviewers for the time and effort they have dedicated to reviewing our work. Their insightful comments and suggestions have been invaluable in improving our project.

In their comments about our work, the reviewers described our method as “interesting” (**Reviewer z3Vc**, **Reviewer jD87**) and “novel” (**Reviewer jD87**). They also described our experimental results to be “strong” (**Reviewer mw4M**), “impressive” (**Reviewer z3Vc**), “thorough” (**Reviewer jD87**), and “extensive” (**Reviewer Cbyh**), showing “significant improvements” (**Reviewer mw4M**) and “excellent reproducibility” (**Reviewer mw4M**). Additionally, our writing is described as “detailed” (**Reviewer mw4M**).

Of the four reviewers, **Reviewer jD87** gave a positive score in the initial review. **Reviewer z3Vc** and **Reviewer Cbyh** both agreed to **raise their scores** and vote in favor of accepting our paper after reading our rebuttal, describing our rebuttal to be “quite reassuring and gives me enough confidence to raise my rating” (**Reviewer z3Vc**) and stating that they “have no further concerns, and I have updated my score accordingly” (**Reviewer Cbyh**). **Reviewer mw4M** did not have the opportunity to respond to our rebuttal efforts. However, we would like to point out that though his/her original score was 4, he/she had **not** raised any concerns regarding the novelty of our method. All of his/her comments are *constructive suggestions or requests for clarification*, which we have fully addressed in our response and in our revised paper. Therefore, we have grounds to believe that **Reviewer mw4M** would have raised his/her score and voted for the acceptance of our paper had he/she been given the opportunity to do so.

We present the main concerns and suggestions from the reviewers and our responses to them below:

**1. Contribution (Reviewer z3Vc):** In our paper, we introduce a novel website development agent, WebGen-Agent, which is **the first code agent to leverage visual feedback from screenshots and GUI agent trajectories** to iteratively improve the generated codebase, significantly outperforming the previous SOTA agentic framework. We further introduce *Step-GRPO with Screenshot and GUI-agent Feedback*, an innovative training method that leverages the step-level appearance and functional scores inherently generated in the WebGen-Agent pipeline as process supervision signals for GRPO training, greatly improving the ability of the coding LLMs to generate complex, interactive websites.

**2. Concern that most of the gains are obtained from the SFT step, which consists of data-distilling Deepseek. (Reviewer z3Vc):** To prove that the SFT stage of data-distilling DeepSeek-V3 is not absolutely necessary, we train the original model (without special SFT) using our *Step-GRPO with Screenshot and GUI-agent Feedback* training approach. Applying the RL training approach directly on Qwen3-8B achieves an improvement **even more pronounced than training on the SFT model**, demonstrating that our training method does not rely on distilling a stronger model and can make the model self-improve.

**3. Generalization to other benchmarks (Reviewer z3Vc, Reviewer jD87)**: To demonstrate the generalizability of our method, we tested WebGen-Agent on the Web Application subset of the ArtifactsBench [1] dataset. WebGen-Agent with Qwen3-Coder-480B-A35B-Instruct as the coding LLM achieves the best performance of 63.86, outperforming the baseline results of various closed-source models and the baseline result of Qwen3-Coder-480B-A35B-Instruct, demonstrating the effectiveness and generalizability of WebGen-Agent.

[1] Zhang, Chenchen, et al. "Artifactsbench: Bridging the visual-interactive gap in llm code generation evaluation." arXiv preprint arXiv:2507.04952 (2025).

Responses to other comments and suggestions are presented in our responses to the reviewers and in the revised paper.

In summary, up to 25 Nov 2025, 13:50 AOE (which is the last time a reviewer posted a response), three (**Reviewer jD87**, **Reviewer z3Vc**, and **Reviewer Cbyh**) out of four reviewers have voted for acceptance of our paper. **Reviewer mw4M** did not have the opportunity to respond, but since all of his/her comments are **constructive suggestions and requests for clarification**, which we have fully addressed in our rebuttal efforts, we have grounds to believe that he/she would have voted for acceptance for our paper as well.

We would like to thank AC, PC, and all the reviewers again for the time and effort they have dedicated to reviewing our paper. We are deeply grateful for all the valuable comments and suggestions, which have been fundamental in improving our project.

---

### Meta-Review · Area_Chair_3HwT · 2026-01-07

**Summary:**

The paper introduces a web code-generation agent that iteratively refines a multi-file codebase using a VLM-based visual scoring from screenshots and GUI-agent interaction-based functional evaluation. These signals also serve as step-level rewards in Step-GRPO to provide dense process supervision for training.

Across reviewers, strengths include strong empirical gains over prior systems, broad model coverage, and solid ablations; concerns around novelty and judge reliability. Overall, the experiments are thorough and cover the right comparisons across several backbones and baselines. The ablations are clear. The approach looks useful for web coding agents, especially when both visual and interaction feedback matter.

**Reviewer Concerns:**

- Addressed concerns: (1) Data provenance/leakage concerns were addressed. (2) Efficiency/compute transparency concerns were addressed. (3) Contribution attribution (SFT vs RL/judge signal) was largely addressed with added evidence.

- Remaining concerns are mainly about broader significance/generalization beyond the benchmark and reliance on the chosen judge/GUI-agent setup.

**Reviewer Scores:**

- Reviewer mw4M: likely would have raised their score (and voted accept) since all concerns were addressed and no novelty issues remained.
- Reviewer z3Vc: did raise their score after the additional experiments and clarifications.
- Reviewer jD87: likely would have kept a positive score (minor concerns addressed; already above threshold).
- Reviewer Cbyh: did raise their score after the follow-up analysis and stated no further concerns.

---

### Decision · Program_Chairs · 2026-01-26

Accept (Poster)